# Global distribution and decline of mangrove coastal protection extends far beyond area loss

Xichen Xu [1,2], Dongjie Fu [1,2] ✉, Fenzhen Su [1,2,3] ✉, Vincent Lyne [1,4], Hao Yu [1,2], Jiasheng Tang[1,2], Xiaorun Hong [1,2] & Juan Wang[5]

Mangroves protect coasts from extreme weather and erosion but can be destroyed by climate change and harvesting. However, there is no consistent formulation of protective capacity that integrates key factors such as area, width, height, and health. Here, we quantified and analyzed a process-based measure of mangrove coastal protection index (MCPI) incorporating cross-shore width, canopy height, and the normalized difference vegetation index (health index). Width/area generally declined at low rates but width increases in some regions reduced MCPI. Cluster exchange network analysis from 2007 to 2019 showed an 800% increase in mangrove forests with characteristically low height, width, and MCPI. Globally, this suggests a 25% decrease in MCPI from 2007 to 2019, primarily from height/biomass change, compared to a 2% decrease in area. Relatively sheltered low-latitude high mangrove strands (>20 m) of high-MCPI appear to be resilient to destruction from cyclones. In contrast, our results highlight an alarming, widespread decline in low MCPI, particularly along coasts exposed to deep water, possibly in concert with human destruction, cyclones, and intensifying oceanic boundary currents.

Global climate change has intensified coastal erosion, flooding, and storm surges, driven by rising sea levels, intensifying boundary currents, and more frequent extreme weather events, amongst other aspects[1–3]. Waves generated by wind and ships have impacted coastal habitats and caused widespread erosion along coasts[4]. Meanwhile, storm surges and hurricanes cause abrupt and catastrophic impacts further inland, posing threats to lives and properties[5–7]. Considering these challenges, mangroves may offer economic, socially beneficial, environmentally friendly, and sustainable coastal protection, compared to engineered coastal defense structures[8,9]. However, questions still remain on how the degree of protection varies with intrinsic structural and health properties of mangroves, how they should be quantified, and how their distributions have changed across space and time[10].

Mangroves intrinsic coastal defense capabilities are due to unique environmental adaptability in otherwise erodible soft substrates that support their unique ecological functions[11–13]. First, coastal mangroves mitigate erosion and stabilize soil, thereby providing long-term coastal protection[8,13]. Their intricate and expansive crown-root systems entrap sediments and dissipate turbulent kinetic energy which also enhances sediment deposition[14,15]. Second, mangroves are crucial in mitigating the impact of tropical cyclones and storm surges[16]. Their complex array of aerial roots, trunks, and leaves can attenuate surge heights and slow the velocity of water flow by impeding the free movement of water through the forest[17–19]. Furthermore, these resilient trees offer substantial protection against wind damage during storms, as their foliage and branches buffer wind-flow velocity[20,21]. Therefore, mangroves provide important long-term coastal defenses to offenses from

[1]State Key Laboratory of Resources and Environmental Information System, Institute of Geographic Sciences and Natural Resources Research, Chinese Academy of Sciences, Beijing, China. [2]College of Resources and Environment, University of Chinese Academy of Sciences, Beijing, China. [3]Collaborative Innovation Center for the South China Sea Studies, Nanjing University, Nanjing, China. [4]IMAS-Hobart, University of Tasmania, Tasmania, Australia. [5]College of Applied Arts and Sciences, Beijing Union University, Beijing, China. ✉e-mail: fudj@lreis.ac.cn; sufz@lreis.ac.cn

extreme weather and ocean-related events. But the question is, which areas can continue to defend coasts against intensifying climate-change forces, which may have already succumbed, and how these changes are distributed in space and time. This study synthesizes aspects of mangroves intrinsic coastal protection processes in order to broadly parametrize the protection ability of coastal mangroves. Such studies are necessary to monitor and prioritize global, regional, and local conservation-management plans and actions.

Multiple evaluations of mangrove coastal protection capacity, mostly through laboratory experiments and numerical simulations, indicate that key factors include: forest width, crown diameter, root density, tree density, forest vertical structure, species, and ecological conditions[13,17,22,23]. One critical factor is distance extending inland from the coastline, which provides inland protection against tsunamis and hurricanes[14,24]. A wider mangrove belt acts as a buffer zone, absorbing and dissipating the energy of incoming waves and storm surges before they reach populated coastal areas[17]. Additionally, vertical and horizontal extents are important factors[8], as studies show that the height of pneumatophores or vegetation could affect their coastal protection capacity[25,26]. However, the variability and complexity of mangrove ecosystems have led to limited research on protection capacity being scattered across regions of significant disparity. Consequently, existing interpretations of protection capacity are inconsistent or missing. For instance, mangroves in Latin America and Africa hold crucial ecological value, yet relevant research in these regions is scarce. Therefore, there is a lack of consistent, systematic, and comprehensive studies of changes in coastal protection ability of mangrove forests that can be applied at a global scale.

Here, using Global Mangrove Watch (GMW) 3.0[27] and calculations detailed in Methods, a Mangrove Coastal-Protection Index (MCPI) was calculated from factors derived from transects perpendicular to the shoreline. Indicator factors used included: Average cross-shore Mangrove Width (AMW) to represent horizontal structure; Average Canopy Height (ACH) to represent vertical structure and biomass; and Normalized Difference Vegetation Index (NDVI) to represent vegetation growth condition[28]. These variables were combined in MCPI (multiplicatively as described in Methods) to quantify the intrinsic ability of mangroves to protect coastal area and to quantify protection/damage status. We used a process-based multiplicative formulation that included Zhang et al.[17] current attenuation data for the width factor. The Cintrón and Schaeffer-Novell's[29] classic mangrove structural relationships were used to formulate a biomass factor that varied with height. For the latter factor, we assumed that biomass characterized drag and turbulence resistance more accurately than height alone. Min-max scaled NDVI was used to reflect overall health through greenness.

To summarize and comprehend changes from 2007 to 2019, we used a Cluster Exchange Network Analysis (CENA) technique adapted from Lyne et al.[30] for studying changes in socio-economic geography. Application of CENA involved clustering normalized ACH, AMW, and NDVI for both years and then analyzing changes in cluster profiles and allegiances between the 2 years. Cluster changes, and corresponding global geo-temporal changes of MCPI, were mapped to analyze distributional changes in factors and mangrove coastal protection capacity. Analyzes also included regression and hypothesis testing on AMW data at different scales to calculate the annual rate of change in width. Significant changes in cluster allegiances and spatial patterns are described along with a discussion of height data-related issues, process dynamics, policy implications, unforeseen implications of the current state, and prognosis for global future coastal destruction of, and protection from, mangroves.

## Results and discussion
### Indicators and mangrove coastal protection index
The spatial patterns of the three indicator MCPI variables for 2007 and 2019 were extracted using a 1° × 1° grid, with non-mangrove pixels excluded during the sampling process, ensuring that width-AMW reflected actual effective width of mangrove portions (Fig. 1). Height-ACH reflected "averaged height" along the transect, hence both actual height reduction and possible changes in tree density contributed to calculated changes in ACH. Average NDVI was used as a scaled min-max variable.

Distribution properties of the variables, in 2007 and 2019, shown as "violin" plots in Fig. 1, reveal that NDVI is top heavy and similar between years, AMW is the reverse and dominated by low widths, while ACH shows a clear flattening-out in 2019. Hence, the clearest variation between the years is declining ACH by 2019, which dominates MCPI variations.

Spatial distribution and key statistics of the indicators and MCPI in 2019 are shown in Fig. 2 and Table 1. Globally, the width-based wave Attenuation factor ($A$) did not exhibit widespread spatial aggregation (Fig. 2a). Unscaled-AMW however showed an extensive range, with Bangladesh recording the highest value of 25.86 km due to its unique geographical and environmental conditions, while the global average was only 1.54 km. The scaled $A$, based on an exponential-protection relationship (see Methods), reduced the actual width disparity. Its distribution globally showed a less pronounced pattern with latitude (Fig. 2e). Approximately half of the grids had $A$ values below 0.2, while values exceeding 0.8 were primarily observed in regions with extensive mangrove cultivation, such as the Gulf of Mexico, Northeastern Brazil, Gulf of Guinea, Bay of Bengal, Borneo Island, and New Guinea.

The biomass-based frontal Barrier protection factor ($B$) exhibited a pronounced latitudinal pattern compared to $A$, decreasing with latitude, with values exceeding 0.8 concentrated along the equator, such as New Guinea, Borneo Island, Sumatra, northwestern and northeastern South America (Fig. 2b, f). The average global ACH was 6 m, with New Guinea recording the highest value of 28.36 m. About one-fifth of mangroves exceeded an ACH of 10 m and had $B$ values recorded as 1.0; meanwhile, more than half of mangroves had relatively small ACH, with $B$ values less than 0.2.

Global average NDVI was 0.593 (Fig. 2c, g). Higher NDVI was along the equator, but slightly higher latitudes also had widespread elevated NDVI values (0.6–0.8), such as the Greater Antilles and eastern South America. Highest NDVI was in the Solomon Islands (0.935), followed by Borneo, New Guinea, and the Caribbean coast (>0.8). Gulf of California and western Asia had lower NDVI (<0.4).

The spatial distribution of the MCPI (scaled by 1000) was computed using formulas detailed in Methods (Fig. 2d). Global average MCPI in 2019 was a very low 80.4, with the highest value (743.48) in New Guinea, showing significant latitude differentiation (Fig. 2h). Latitudinal peaks of MCPI appeared at the equator, related more closely to $B$ than the more-broadly distributed $A$ and flatter NDVI. Larger MCPI were primarily along eastern Venezuela, northeastern Brazil, western Colombia, Gulf of Guinea, eastern Andaman Sea, northwestern Borneo, and western New Guinea. Regions with low MCPI were primarily along Gulf of California, Caribbean, Red Sea, Gulf of Oman, western India, southern China, and Australia.

Thus, high MCPI mangroves are primarily distributed within latitudes of 5° N and 5° S, resulting from the combined influence of natural conditions and human factors. Natural factors such as higher rainfall, warmer temperatures, and lower tropical cyclone frequency provide better ecological foundations for mangroves[2,31]. Additionally, the relatively minor impact of human activities in these regions and favorable tidal hydrodynamic conditions help maintain their wider coastal width[12]. Higher latitudes, like Northern Bay of Bengal, have shorter mangrove heights and less favorable growth conditions. Still, their extensive distributions afford decent overall inland protection. However, notable MCPI variations go against conventional understanding, even in consistent climates. For instance, in New Guinea Island, differing widths reduce protection capacity for eastern mangroves, next to deeper ocean, compared to the inshore western part.

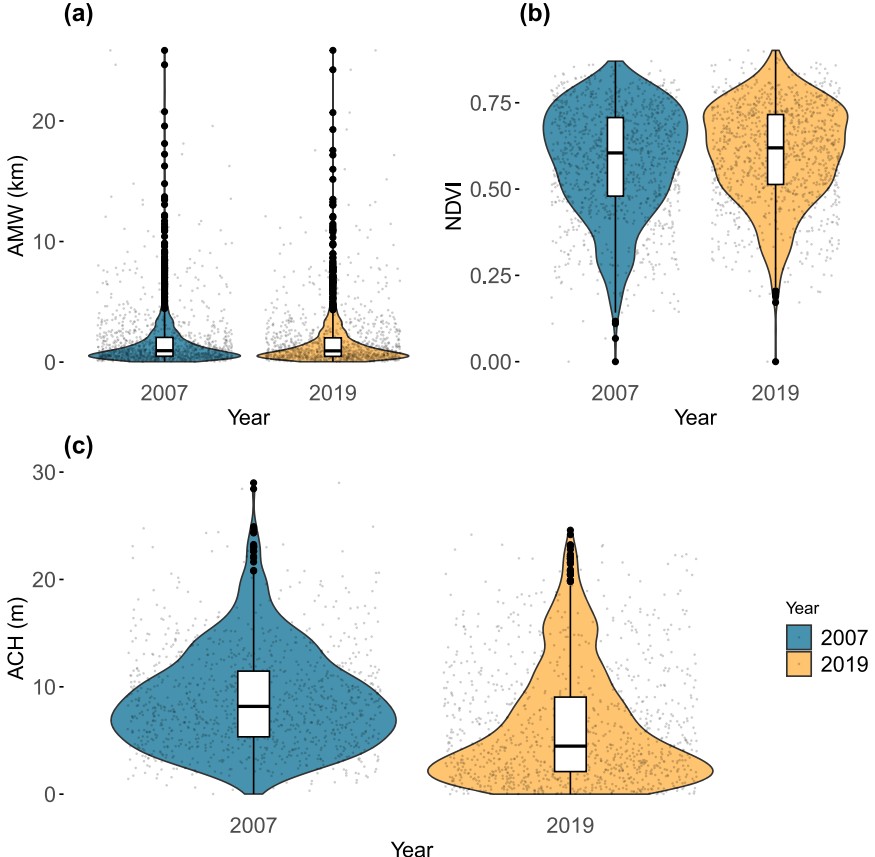

**Fig. 1 | Statistical "violin" with included Box-and-Whisker plot.** The violin width indicates the frequency distribution of data points at that variable value—indicated by the y-axis. Data points are plotted with horizontal "jitter" to prevent overlap. The box represents the median (Q2) and interquartile range (IQR) between the 25th (Q1) and 75th (Q3) percentiles. Variables are: (**a**) Average cross-shore Mangrove Width (AMW), (**b**) NDVI, and (**c**) Average Canopy Height (ACH), plotted for the 2 years.

This suggests that coasts next to deep water may be more exposed, than sheltered coasts, to threats from both ocean storm waves and cyclones making landfall from the deep sea. Also, coasts with wide/sheltered continental shelves able to attenuate ocean storms include: the north-eastern coasts of South America of relatively wide shelves; substantial parts of Australia's wide North-West Shelf; and the sheltered seas between North and South America (Caribbean Sea and Gulf of Mexico).

## Width trends

Global AMW were extracted for 1996, 2007–2010, and 2015–2020, and regressions conducted for 1996–2020 (Fig. 3) to analyze global and regional trends. Key results listed in Table 2, include:

1. Average AMW declined over the 24 years from 2122 m in 1996 to 2059 m in 2020 (Fig. 3c and Table 2) at −2.34 m/year, representing a rate of only -0.124 %/year.
2. For a finer grid scale, overall loss in width (−59 m, −3.84%) was relatively greater, compared to loss in average area (−365 ha, −3.44%).
3. Worldwide, significant regional linear declines occurred in approximately half of 34 countries (Fig. 3a, b). Declines >10 m/year occurred across Asia and Africa, including Myanmar, Nigeria, Guinea, Guinea-Bissau, and Sri Lanka. Pakistan declined the most at >20 m/year.
4. At continental scales, significant linear loss in AMW was observed in Asia (−88 m), Africa (−80 m), and America (−33 m), and less so in Oceania (−11 m) (Fig. 3d, e, f, g).
5. AMW increased across Asia and the Americas in only 10 countries, led by Bangladesh at >40 m/year, with French Guiana, Suriname, and Guyana at >10 m/year.

Grid-scale results were different from averaging over larger scales (Fig. 3a and Table 2):

1. Fastest reduction was in Borneo, >150 m/year, and fastest increase in the Brazilian Amazon basin, >50 m/year.
2. South-eastern Borneo and Sulawesi Island mangrove loss were severe and continuous, and rapid declines in Indonesia were probably due to conversion for agriculture, aquaculture, and oil palm expansion[32,33]. By comparison, Bangladesh's mangrove conservation policies demonstrated more favorable outcomes.
3. For Brazil, despite some regions showing high annual mangrove width change rates (such as the Amazon Basin), many mangroves along its eastern coastline did not exhibit a sustained trend[34].

To summarize, declines are more globally prevalent than increases. But substantial regional disparities exist between regions (e.g., Borneo and Brazil). Relative loss in width was slightly higher than area, but still relatively low at under 4% across the 24 years.

## Height trends

For height analyzes, 2007 forest canopy height data were missing, hence we used available coarser-resolution data for 2005. Globally, mean (median) ACH across the 1°-grid cells declined from 8.8 m (8.18 m) in 2005 to 6.2 m (4.48 m) in 2019, representing a (mean) decline of 2.1%/year or 29.4% (~41% biomass decline) across the 14 years. The median decline rate was higher at 3.2%/year, suggesting that distributional properties were also changing inconsistently by becoming more positively skewed, with more patches of lower height (see Fig. 1c). This represents almost a 20-fold (28-fold biomass) greater decline compared to the global width change noted in Table 2.

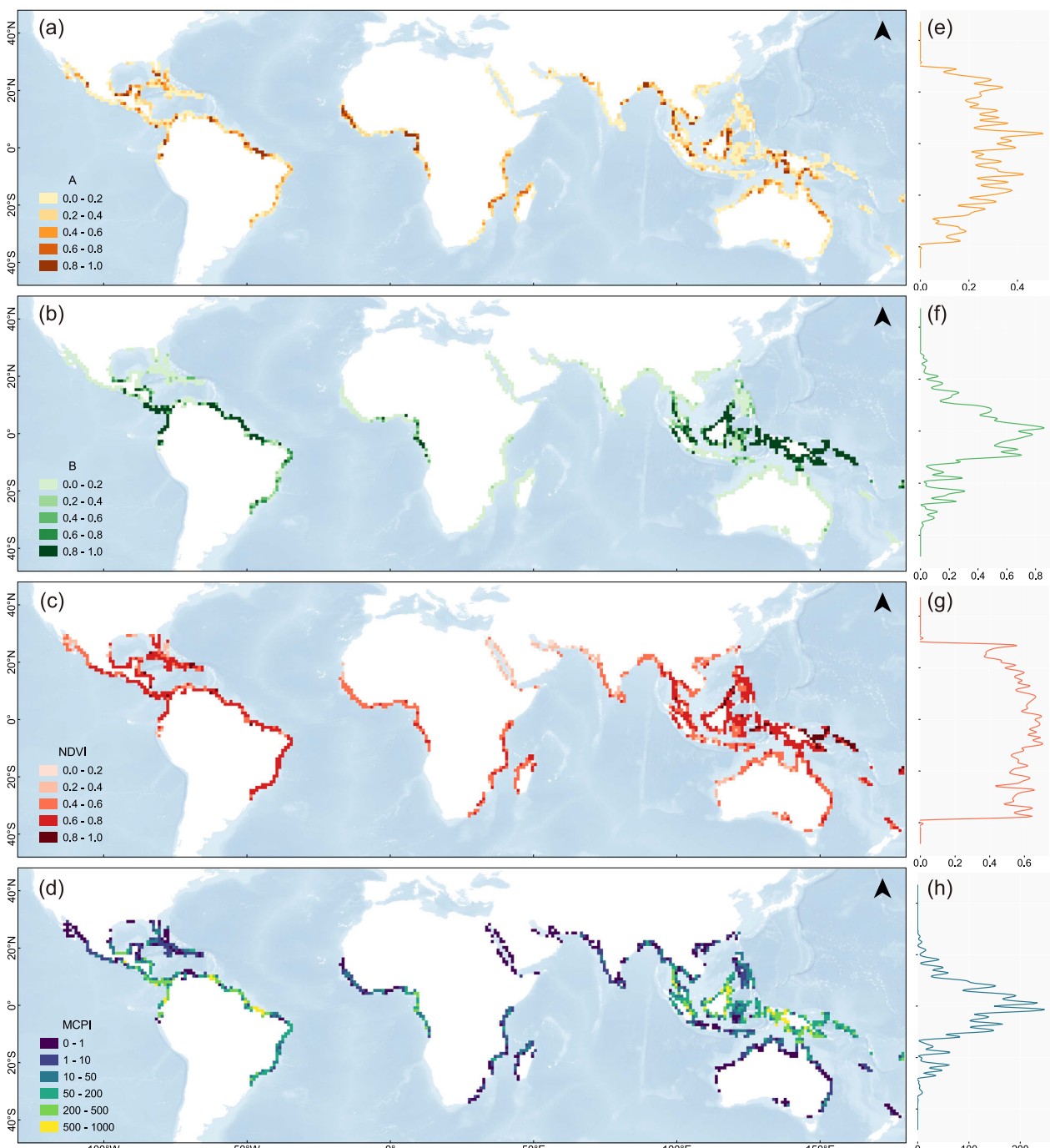

**Fig. 2 | Spatial distribution of global mangrove coastal protection capacity for 2019. a** width-based wave Attenuation (A), (**b**) biomass-based frontal Barrier (B), (**c**) average Normalized Difference Vegetation Index (NDVI), (**d**) Mangrove Coastal Protection Index (MCPI), and (**e–h**) latitude statistics of three indices and MCPI. The background bathymetry data for all spatial maps were derived from the General Bathymetric Chart of the Oceans (GEBCO) dataset, with blue shading indicating increasing water depth from light to dark.

The calculated ACH changes can be attributed to actual height reduction and possible changes in tree density due to thinning effects (more patches of lower height noted above). It should be noted that these changes could be influenced by possible data-related issues from transect-patchiness to some extent. Whilst resolution of confounding interactions in some areas may await more accurate future data, the 2019 height notes that many deep-water coastal areas are already of low height (<10 m) or biomass (Fig. 2b). We also discuss later the regional spatial height variations using clusters which attempt to unravel regional and local inconsistencies in height changes suggested by differences in the mean and median rates, as detailed in the "Cluster Exchange Results" section.

### NDVI trends
As reported in the Supplementary Discussion, NDVI deceptively increased between 2007 and 2019 by 2.34%, but this increase was composed of a dominant global greening increase and an apparent rapid 19% decrease from loss of biomass[35,36]. Overall, NDVI increased from greening and hence played a minor role at the global scale.

## Temporal changes in global MCPI

Global MCPI (using *A*, *B*, and NDVI; scaled by 1000) and change rates were calculated for 2007 and 2019 (Fig. 4). From 2007 to 2019, global MCPI declined heterogeneously with an average change of −31.07 (−25.01%), which was much more severe compared to the contemporaneous average-area decline (−2.11%). Note also that this is less than the 29.4% height decline due in large part to the 10 m height-protection limit. Significant declines in MCPI were predominantly in Central America, Caribbean, Africa, South Asia, and Australia.

**Table 1 | Key statistics of AMW (Average cross-shore Mangrove Width), A (width-based wave Attenuation), ACH (Average Canopy Height), B (biomass-based frontal Barrier), NDVI, and MCPI (Mangrove Coastal Protection Index)**

| Measure | AMW (km) | A | ACH (m) | B | NDVI | MCPI |
|---|---|---|---|---|---|---|
| Q1 | 0.40 | 0.11 | 1.81 | 0.01 | 0.50 | 0.91 |
| Q2 (Median) | 0.77 | 0.20 | 4.13 | 0.10 | 0.61 | 11.99 |
| Q3 | 1.64 | 0.38 | 8.88 | 0.74 | 0.71 | 95.09 |
| Average | 1.54 | 0.28 | 6.00 | 0.34 | 0.59 | 80.38 |
| Std Dev | 2.34 | 0.23 | 5.41 | 0.40 | 0.16 | 144.46 |
| Maximum | 25.86 | 0.99 | 28.36 | 1.00 | 0.94 | 743.48 |

Q values refer to quartiles (Q1 is 25%, Q2 is 50%, and Q3 is 75%).

Conversely, increases were mainly in South America, Southeast Asia, and Northern islands of Oceania. Using only reliable data, severest decreases were located at the southwest corner of Florida (−534.3) and the Gulf of Guinea; while largest increases were near northeast Brazil (384.4) and the eastern Andaman Sea, as shown in Table 3.

Spatial patterns of annual change rate (Fig. 4b) differed from change values, as some regions such as South Asia exhibited high change rates compared to modest actual changes. Regions with strong mangrove coastal protection capacity had varying temporal variations. For example, eastern Venezuela, Brazil, and western New Guinea Island had progressive MCPI increases, but central Gulf of Guinea consistently declined (Figs. 2d and 4a). Given potential data issues, more credence should be attached to spatial variations than temporal trends.

Factor-wise, MCPI varied mainly from ACH, followed by AMW, and minor NDVI contributions. Comparison of changes in MCPI and mangrove area (Fig. 4, Supplementary Fig. 5 and Table 3) showed global change in MCPI between 2007 and 2019 (77.82% of study area) were more extensive than area (70.58% of study area), and changes were not necessarily positively correlated. For instance, in northeast Brazil, western Myanmar, and southeastern New Guinea, mangrove area decreased while MCPI increased. Conversely, in Gulf of California, western Gulf of Guinea, western India, Bay of Bengal, northern Java Island, and eastern Australia, area increased without MCPI

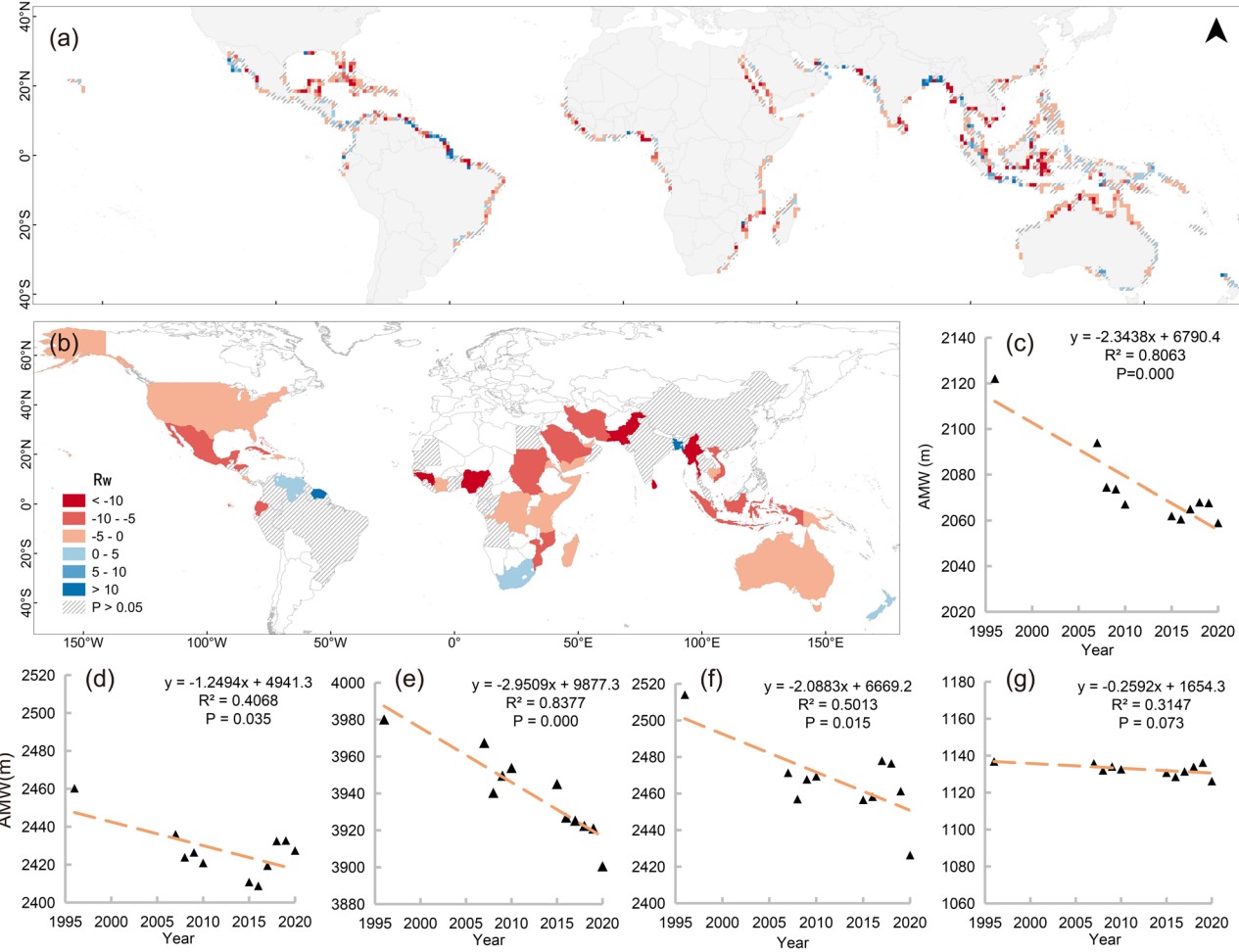

**Fig. 3 | Results of trend regression analysis for Average cross-shore Mangrove Width (AMW) from 1996 to 2020. a, b** represent the spatial patterns of annual AMW change rates (R_w, the rate of width change) extracted using 1° × 1° grids and national boundaries, respectively. The unit is meters per year. **c** depicts global average trend of AMW. **d, e, f, g** represent AMW trends in the Americas, Africa, Asia, and Oceania, respectively.

improvement. This was also the case for the Amazon basin which increased in area but MCPI dropped to almost zero.

## Cluster exchange results

Our intent with the CENA (reported in Methods) was to explore spatial-temporal changes in the factors used to construct MCPI. Note that this analysis does not discriminate between real or data-related changes. Instead, it tells when and where those changes are occurring so that we may more reliably interpret and draw conclusions from the results. The Clusters are described before investigating the changes using the CENA process. The normalized-factor profiles of the 8 clusters are shown in Fig. 5a. Overall, the factor profiles reflect distributions seen in Fig. 1 with low AMW, moderate-high NDVI, and a significant distributional change in ACH. Cluster 3 is least compromised with moderate-high factors, but all others are compromised (red/orange cells in Fig. 5b) either with low ACH and/or AWM, such as Clusters 6 to 8. Cluster 8 is the lowest performing for ACH and AMW (and hence MCPI).

The matrix and network plot in Fig. 6 shows Cluster exchanges taking place across the years. For example, in 2007, Cluster 1 (along the matrix row for Cluster 1) shows 2 cells transferred to Cluster 7—as evident also by the network plot showing a unidirectional arrow from 1 to 7. Looking down the column for Cluster 1, for 2019, shows the 12 common cells, but cells from no other Clusters transferred to Cluster 1. Since Cluster 1 has widest width, this indicates that no other cell grew wider from 2007 to 2019 to join Cluster 1. By contrast, for the highest Cluster 2, 3 cells from Cluster 4 did "increase" to join it in 2019. Note also, the massive new contributions to the lowest-performing Cluster 8 in 2019.

Exchange of cells indicates growth and decline of Clusters with the most prominent being the substantial growth in Cluster 8 of low ACH and AMW (Fig. 6 and Supplementary Fig. 6). Referring back to Fig. 5, this is primarily from Clusters with higher ACH but low AMW, in the order Cluster 6, 5, and 4 by increasing ACH, respectively. This indicates that apparent ACH reduction is primarily responsible for the growth of Cluster 8, with lower ACH Clusters contributing more than higher ones; hence contributions are inversely related to ACH, or closeness (in Height) to Cluster 8. The reductions in ACH can be attributed to two potential mechanisms: a genuine reduction in height, which may result from recovery following mortality events (caused by factors such as human influence, cyclones, drought conditions, and hydrological limitations)[37–40]; and a reduction in tree density measured differently in 2005 and 2019 because ACH is an average measure based on data resolutions and transects perpendicular to the coastline (Supplementary Figs. 16 and 17).

Variations of the MCPI factors and MCPI for Cluster 1 in Supplementary Fig. 7 show that Height reduction caused MCPI to decline. Whereas for Clusters 2, 3, and 4, all factors and MCPI increased.

**Table 2 | Key statistics of regions and selected areas for mangrove width change**

| Regions | Width change (m) | Annual change rate (%) | Annual change speed (m/year) | p value |
|---|---|---|---|---|
| Global | −63.11 | −0.124 | −2.34 | 0.000 |
| Asia | −87.71 | −0.145 | −2.09 | 0.015 |
| Africa | −79.59 | −0.083 | −2.95 | 0.000 |
| Americas | −32.86 | −0.056 | −1.25 | 0.035 |
| Oceania | −10.66 | −0.039 | / | 0.073 |
| Pakistan (country) | −981.44 | −0.797 | −26.91 | 0.002 |
| Bangladesh (country) | 838.96 | 0.319 | 44.10 | 0.002 |
| Borneo (grid) | −3079.99 | −1.125 | −162.92 | 0.000 |
| Amazon basin (grid) | 1058.23 | 20.154 | 66.45 | 0.001 |

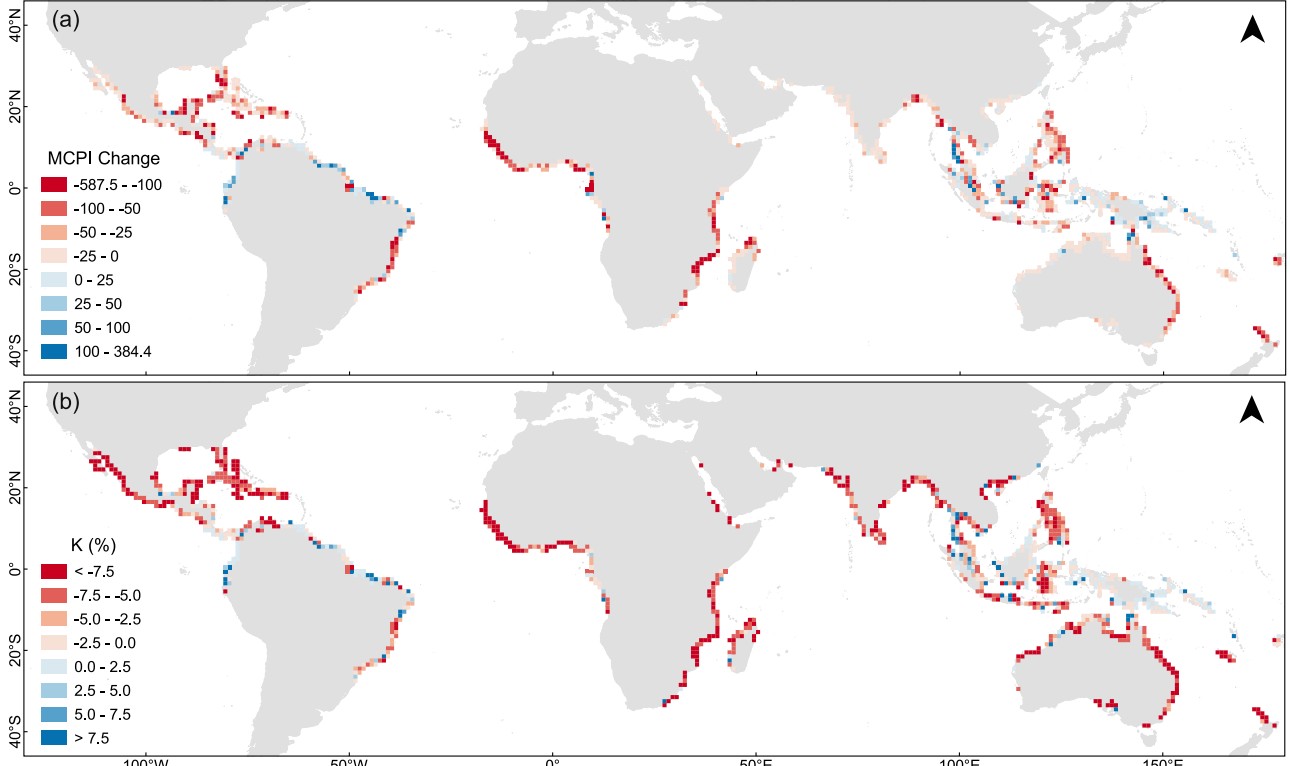

**Fig. 4 | Spatial variations in mangrove coastal protection index (MCPI) changes from 2007 and 2019 (with canopy-height data from 2005). a, b** represent spatial distribution of MCPI change values and annual change rates (K), respectively.

**Table 3 | Key statistics of typical areas in MCPI (Mangrove Coastal Protection Index) and mangrove area change (the typical regions are several specific grids in the area, chosen by the most significant MCPI change and Area change)**

| Regions | MCPI change | MCPI change rate (%) | Area change (ha) | Area change rate (%) | Cluster change |
|---|---|---|---|---|---|
| Global | −31.1 | −25.0 | −217.6 | −2.1 | / |
| MCPI↓: Southwest Florida | −534.3 | −80.2 | −4287.6 | −4.3 | 3→7 |
| MCPI↓: Gulf of Guinea | −457.8 | −83.8 | −1244.1 | −0.5 | 1→1 |
| MCPI↑: Baía de Turiaçu (Northeastern Brazil) | 384.4 | 119.2 | −4261.0 | −2.8 | 3→3 |
| MCPI↑: Eastern Andaman Sea | 334.6 | 226.3 | 101.2 | 0.3 | 6→5 |
| Area↓: Borneo | −122.6 | −23.9 | −35492.6 | −24.4 | 3→3 |
| Area↑: Amazon basin | −133.1 | −94.6 | 11462.2 | 123.9 | 4→8 |

Please refer to the "Cluster Exchange Results" section below for details on "Cluster change".

However, going from Cluster 4 to 8, height reduces hence membership changes down that path to Cluster 8 gradually reduces MCPI.

The contributions to Cluster 8 from other Clusters are depicted in the bootstrap Cluster probability correlation matrix plot in Supplementary Fig. 8. Most Cluster memberships within 2019 were negatively correlated (to those in 2007), particularly for Cluster 8.

This reinforces the suggestion, shown spatially in Supplementary Fig. 9, that greatest ACH changes appear to be associated with all the major tropical/sub-tropical oceanic boundary-current regions. More generally, the example of New Guinea noted earlier suggests that coasts next to deep water may be under assault from ocean storm surges and land-fall cyclones, compared to shallow areas away from the deep sea. Hence both warm and cold current coasts are affected suggesting that if the change is related to ocean-related destruction in these regions, these may arise from a static sea-level rise and a dynamic component due to acceleration (hence kinetic-energy increase) in boundary currents as well as energetic storm events. Boundary-current regions, such as the East Australia Current, are hotspots of global warming, sea-level rise, storms, and marine heatwaves[41]. If mangroves in global hotspot regions are low in protection capacity and also declining, then they would not be able to sustain increasing climate change forces such as more frequent and intense storms, rising sea levels, and widespread sea surface density variations forcing currents[42,43]. Also, it is well-known that any increase in boundary-current velocities is linearly related to dynamic sea-level height increases (above sea-level rise) required to drive these currents. For example, intensification of the East Australian Current (EAC) has been noted by Kelly et al.[41] who describe a rapid extension of the EAC since the 1990s such that seasonal cycles off mid-eastern Tasmania in 2004–2005 were similar to those much further north previously. Dynamically boundary-currents are driven by cross-shore pressure gradients, hence similar mechanisms may operate in other regions; further fueled by high kinetic energies (through swift currents and eddies) associated with these oceanic systems, climate-change intensified storm surges, and storms. As noted previously, wider shelves would be able to attenuate these forces more effectively.

Cyclones may also contribute to the decline of mangroves. The comparison of tropical cyclone distribution (Supplementary Fig. 18) with mangrove height changes (Supplementary Fig. 9) from 2005 to 2019 shows some overlapping hotspots. These areas, such as the Philippines and southern China, frequently experience storm events that coincide with declines in mangrove height. Severe storm events can result in extensive mangrove mortality, with recovery taking several years or more[38,44]. For instance, in southwestern Florida, MCPI decreased significantly from reductions in width, height, and NDVI, likely due to the impact of Hurricane Irma[45] (Supplementary Fig. 17). Low-lying topographic conditions, hydrologic isolation caused by natural or artificial coastal barriers, and storm surges all contributed to poor drainage of mangroves, resulting in extensive diebacks post-storm[46]. These mangroves demonstrated low resilience and had not recovered by 2019, as indicated by the detected decline in ACH (very low or zero).

Climate change-induced sea-level anomalies and extreme high temperatures may also impact the health of mangroves, such as those located in the Gulf of Carpentaria, Australia[47]. Additionally, harvesting of mangroves may reduce tree density and above-ground biomass, thereby accelerating local declines in mangrove protection capacity[48]. Our modeling does not distinguish between these destruction modes and changes may in some cases be confounded by data-related issues. All discussions here are potential possibilities, intended to provide references, modeling, analytical techniques, caveats for mangrove protection policies, and suggestions on further research of data-related issues.

## MCPI implications

Globally, despite some positive MCPI increases in certain regions, AMW, ACH, and NDVI changes, all point to continuing declines in low-existing, global MCPI as of 2019. However, protective effectiveness of mangroves along coastlines involves complicated factors, including mangrove species, local hydrodynamic characteristics, and intensity/direction of storm events[23]. We selectively employed a representative subset to broadly represent functionalities of mangrove coastal protection capacity and to provide valuable insights for coastal mangrove management. While analyzes of trends and change may be confounded by data issues, the finer-resolution 2019 data suggests that very substantive parts of the global coastline next to deep water are poorly protected by mangroves which may continue to decline in protection ability as destructive drivers increase. In future, additional indicators and local adaptations of our MCPI model could be used to enhance local applications. The 10 m limit for height protection also potentially hides destruction of higher mangroves.

Still, different and in many cases dramatic, change patterns between MCPI and traditionally-used mangrove area indicate that MCPI captures expanded insights into mangrove protection attributes[49,50]. First and foremost, we need to understand more deeply the processes at play that are dramatically stressing mangroves along deep-water boundary-current coasts and possibly changing mangrove height distributions (Fig. 1 and Supplementary Fig. 9). Mangroves here need protection, as they cannot even protect themselves. If mangrove thinning is responsible for declines, then investigations are required to understand if area increases may be from migration of young mangroves landward or seaward. Landward migration may be from seawater floods breaking through at the leading-edge, and seaward from sediment buildup offshore of deltas from coastal sand migration, or terrestrial sources.

When formulating policies for mangrove conservation, countries should not solely focus on expanding the geographical distribution of mangroves. Critical growth parameters like tree

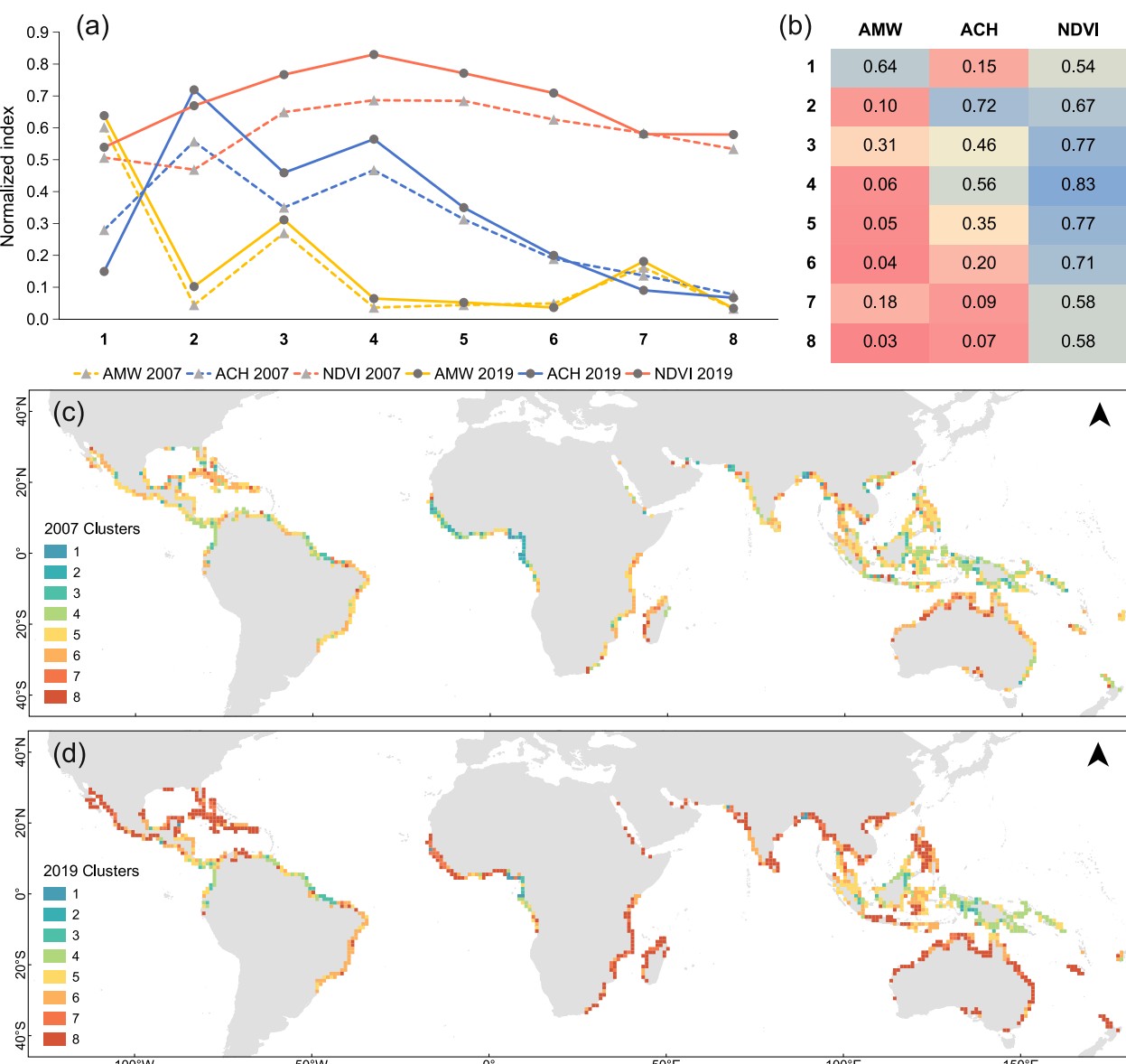

**Fig. 5 | Characteristics and distribution maps of Clusters. a** shows profiles of normalized factors (Average cross-shore Mangrove Width, Average Canopy Height and Normalized Difference Vegetation Index) with respect to the 8 Clusters for the 2 years, 2007 and 2019; (**b**) shows the mean value of the factors by Cluster for 2019, with a red-yellow-blue color scale indicating increasing values; (**c**, **d**) represent spatial distribution of the 8 Clusters for 2007 and 2019, respectively.

density and canopy coverage must be included in monitoring, reafforestation, and conservation plans. For example, if we accept that the 124% area increase in the Amazon Basin, countered by a 95% MCPI decline, is from newly established but shorter mangroves, this may reduce overall ACH (and MCPI). This implies that extensive new mangroves, whether from inland migration or planting, may not have a positive short-term impact on overall coastal protection capacity of mangroves, and would be more vulnerable to destruction by extreme events.

From a long-term perspective, both mangrove width and height show possible declining trends in regions of low MCPI globally. Since climate forces could probably intensify in time, our principal conclusion is that mangroves require significantly enhanced protection to maintain their coastal protection capabilities. Otherwise, it is fair to conclude that low-MCPI mangroves, particularly along deep-ocean boundary-current coasts, are unable to protect themselves, let alone inland areas, even in managed areas like Australia.

## Methods

### Sampling method

To account for directionality of mangrove protection capacity, we used a coastline-based transect sampling method[51]. First, due to the complex topological relationship between mangroves and coastlines[52], we manually adjusted the coastlines to approximate the straight line of outer wrapping of mangroves based on the mangrove distribution extent from GMW. This ensures that our sampling lines cover all mangroves and minimizes under-sampling and over-sampling. Second, equally-spaced transect lines perpendicular to shoreline were generated at 1 km intervals based on the updated coastline to extract mangrove transect statistics (shown in Supplementary Figs. 10 and 11). The sampling line was determined from actual growth of mangroves in each region, using ecoregions from Marine Ecoregions of the World (MEOW)[53] (shown in Supplementary Fig. 13). AMW, ACH, and NDVI, for the respective years were extracted from sampling lines after clipping the mangrove data using refined 1° × 1° grids. All data used are listed in the Data Availability section.

(a)

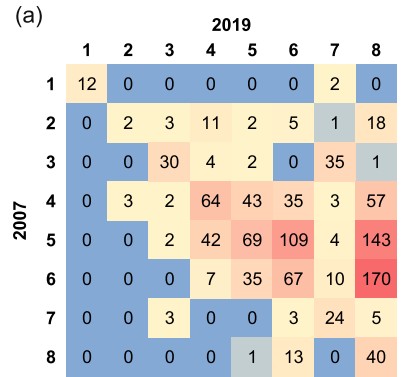

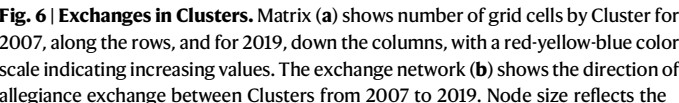

(b)

**Fig. 6 | Exchanges in Clusters.** Matrix (**a**) shows number of grid cells by Cluster for 2007, along the rows, and for 2019, down the columns, with a red-yellow-blue color scale indicating increasing values. The exchange network (**b**) shows the direction of allegiance exchange between Clusters from 2007 to 2019. Node size reflects the number of common cells in the 2 years (matrix diagonal). Node background colors represent different Clusters, consistent with Fig. 5. Arrows reflect exchange directions, and exchange amounts are represented by the matrix numbers.

The specific indicator definitions and calculations are as follows:

(1) *Average mangrove width* (*AMW*): To compute mangrove width, the number of mangrove pixels along the transect was multiplied by the sampling distance. This is represented as the sum of lengths of multiple segmented mangrove lines along the sampling lines (as shown in Supplementary Fig. 10). The average widths of all transects yields the mangrove width for that area:

$$AMW_k = \frac{1}{m}\sum_{i=1}^{m} C \times N_{i,k} \qquad (1)$$

Where, $AMW_k$ represents average mangrove width for *k*th region; $C = 30$ m denotes the pixel size. For *k*th region, $N_{i,k}$ is number of mangrove pixels for *i*th sampling line, and *m* is total number of sampling lines.

For the process-based MCPI, protection from width used the field and modeling results of Zhang et al.[17] as presented in Waves 2016 (Fig. 2.7)[54]. We fitted a wave attenuation factor *A* that varied with width of mangrove *W* (in km):

$$A = 1 - \exp\left(-0.29 \times W\right) \qquad (2)$$

Where *A* varies from 0 (no attenuation at $W = 0$) to 1 ($W \sim \infty$). The half-width of decay is roughly 2.3 km. So about 75% protection is achieved at ~4.6 km. These parameters contrast with idealized laboratory attenuation studies for small waves (0.2–0.9 m) and short periods in shallow water, which suggest that *Rhizophora* mangroves with widths of more than 300 m (several wavelengths) are needed to reduce incoming waves by more than 50%, while widths of more than a kilometer attenuate waves by more than 80%[55]. Turbulent processes are well-known to be highly nonlinear, hence our formulation of *A* based on Zhang et al.[17] data is, therefore, more likely to be representative of actual conditions requiring much larger attenuation widths. But the use of width within a wave attenuation factor is well supported by both field and laboratory studies. *A* was used in place of AMW in MCPI.

(2) *Average canopy height* (*ACH*): Global canopy height dataset for the years 2005[56] and 2019[57] was masked using mangrove distribution range data. We approximated the height for 2005 to represent the height in 2007 because the closest mangrove distribution data available from GMW was for 2007. Canopy height was extracted using sampling lines, and arithmetic height mean from all lines in the region as follows:

$$ACH_k = \frac{1}{n}\sum_{j=1}^{n}\frac{1}{m}\sum_{i=1}^{m} CH_{i,j,k} \qquad (3)$$

Where, $ACH_k$ represents ACH in the *k*th region; $CH_{i,j,k}$ denotes canopy height of *i*th pixel crossed by *j*th sampling line in *k*th region; *m* signifies the total number of pixels containing mangroves along the *j*th sampling line, while *n* represents the total number of sampling lines containing mangroves in the *k*th region.

For the process-based MCPI, we reasoned that total aboveground biomass (leaves, branches, root, and trunk) were all critical in moderating the effects of storms, rather than just height alone, as biomass is more representative of the total 3D frontal drag area that waves and wind have to work against. This assumes very crudely that densities of the different components of the tree are similar or can be parametrized as such within a modeling constant. We relied on the classic formulations of Cintrón and Schaeffer-Novelli[29] for relationships between biomass, breast-height diameter, and height, with an implicit assumption that canopy leaves/branches/twigs are included in biomass:

$$B \sim \left[d^2 h\right]^m \qquad (4)$$

Where *d* is the diameter at breast height (about 1.3 m above ground), *h* is the mangrove ACH, and the value of *m* from the ref. 29 is 0.85. Hence:

$$B \sim d^{1.7} h^{0.85} \qquad (5)$$

In turn, diameter *d* (cm) is approximately related to height *h* (m) up to about 10 m (see Table 6.7, page 111, in ref. 29):

$$d \approx 0.75 * h; \text{hence } d \sim h \qquad (6)$$

These equations can then be used to derive the Biomass (*B*) relation:

$$B \sim h^{2.55} \qquad (7)$$

Noting that Eq. (6) is only valid up to about 10 m, after which *d* may grow larger, we converted the biomass relation to an index, which in effect underestimates growth of *d* for high

strands (hence is conservative):

$$B = \left(\frac{h}{10}\right)^{2.55} \text{ for } h <= 10\,\text{m}; \text{ and } B = 1 \text{ for } h > 10\,\text{m} \qquad (8)$$

$B$ was used in place of ACH in MCPI.

(3) *Average normalized difference vegetation index (NDVI)*: Landsat Collection 2 surface reflectance was processed to remove cloud using the Quality Assurance (QA) band. Average value was computed for all qualifying images within the study period. Images were masked with mangrove distribution and NDVI was extracted using sampling lines, and arithmetic means from all sampling lines for the region. Higher NDVI value signify denser vegetation and stronger coastal protection capacity according to the formulas:

$$NDVI = \frac{\rho_{nir} - \rho_r}{\rho_{nir} + \rho_r} \qquad (9)$$

$$NDVI_k = \frac{1}{n}\sum_{j=1}^{n}\frac{1}{m}\sum_{i=1}^{m}NDVI_{i,j,k} \qquad (10)$$

Where $\rho_{nir}$ represents near-infrared reflectance; $\rho_r$ represents red-band reflectance; $NDVI_k$ is average NDVI in $k$th region; $NDVI_{i,j,k}$ is NDVI for $i$th pixel along $j$th sampling line in $k$th region; $m$ is total count of mangrove pixels for $j$th sampling line, and $n$ is number of sampling lines containing mangroves in $k$th region.

Illustrations of how factors were extraction are presented in Supplementary Figs. 10 and 11.

## Construction of the mangrove coastal protection index

The three indicators comprising horizontal structure (AMW), vertical structure (ACH), and growth condition (NDVI) were extracted at each grid cell and scaled to the range of 0–1 using the equations above, and then scaled by 1000 to calculate the mangrove coastal protection index (MCPI):

$$MCPI_k = A_k \times B_k \times NDVI_k \times S \qquad (11)$$

Where, $MCPI_k$ represents the mangrove coastal protection capacity in the $k$th region; $A_k$, $B_k$, and $NDVI_k$ represent the width-based wave attenuation, the biomass-based frontal barrier, and NDVI in the $k$th region, respectively; and $S$ is a scaling factor set at 1000.

## Evaluations of MCPI

The validity of global coastal protection capacity was tested by examining changes in NDVI in hinterland areas beyond mangroves before and after tropical storms. Storms causing damage to vegetation in coastal areas may decrease NDVI[58–60], but mangrove protection may mitigate damage[61], thereby reducing the decline in NDVI.

Global tropical storm data from International Best Track Archive for Climate Stewardship (IBTrACS)[62,63] was utilized for this study. Initially, all tropical storm tracks intersecting with coastlines and occurring between 2017 and 2023 (extending several years before and after 2019) were selected, with maximum sustained wind speeds exceeding 64 knots, representing severe tropical storms. A 5-km buffer zone was generated along storm tracks to define their impact areas. Coastal mangrove areas intersected with storm-impact zones were used to identify mangrove areas affected by tropical storms. Average NDVI changes in these areas 1-month before and after each storm event were calculated using Sentinel-2 imagery. Finally, the relationship between NDVI changes caused by each storm event and corresponding MCPI values in those regions was established.

After excluding areas with missing images, a total of 53 mangrove conservation areas that experienced severe tropical storm events were identified, as shown in Supplementary Fig. 12 and Supplementary Table 1. The results indicate a Spearman rank-order correlation coefficient ($\rho$) of 0.37 ($p = 0.0053$) between MCPI and NDVI changes in these areas. This demonstrates a significant monotonic positive relationship that provides support for the MCPI assessment.

## Trend analysis

Mangrove width for 11 individual years between 1996 and 2020 was extracted to analyze the trend of mangrove width using linear regression for annual change-rate of width over the course of 24 years. The specific calculation methodology was:

$$\theta_{\text{slope}} = \frac{n \times \sum_{i=1}^{n} i \times W_i - \sum_{i=1}^{n} i \sum_{i}^{n} W_i}{n \times \sum_{i=1}^{n} i^2 - \left(\sum_{i=1}^{n} i\right)^2} \qquad (12)$$

Where, $\theta_{slope}$ represents the fitted slope of the mangrove width-change curve; $i$ comprises the study years (1996, 2007, 2008, 2009, 2010, 2015, 2016, 2017, 2018, 2019, 2020); total number of study years is $n = 11$; $W_i$ is mangrove width for $i$th year. Positive slope (meters per year) suggests enhanced coastal protection capacity and vice-versa. Significance was tested with a $p$ value of the univariate model set at 0.05. Regions where $p$ value exceeded significance level were deemed not to have a linear trend.

Due to availability of global forest canopy-height data, only 2007 and 2019 were used for analyzing MCPI. A single-factor dynamic-change model was used to measure changes across these years as follows:

$$K = \frac{U_b - U_a}{U_a} \times \frac{1}{T} \times 100\% \qquad (13)$$

Where $K$ represents the dynamic change factor; $U_a$ and $U_b$ denote specific attribute values (such as mangrove width, coastal protection index, etc.) at initial (a) and final (b) periods; $T$ is duration between periods. If MCPI is chosen as the factor and the unit is set to years, then $K$ represents the annual change rate of MCPI.

## Cluster exchange network analysis (CENA)

To efficiently explore changes across factors, regions, and years, we used clustering analysis to characterize the typology of networked-linked exchanges between nodes representing the clusters. The challenge here is that we want to explore changes across two time periods: 2007 and 2019. For this, we used CENA developed by Lyne et al.[30] to study socio-economic network exchanges across Australia's geographic regions. Application of CENA involved clustering of normalized height, width, and NDVI across both 2007 and 2019; in order to capture the complete typology and network linkages across both years, and hence for each year as well. Thus, this clustering does not consider the causes of changes between years and across space within each year.

Hierarchical Ordered Partitioning and Collapsing Hybrid (HOPACH)[64] was used to identify "maximally homogeneous" clusters (based on the "cosangle" distance metric). Probabilities of fuzzy-cluster allegiances were from non-parametric bootstrapping. The exchange of these network matrixed allegiances across years and geographic space identified characteristic exchanges of mangroves factors, and hence implied-MCPI change. We only used one level of 8 clusters as it was sufficient to identify the major interchange. The R package "Network" was used for network plotting.

## Data availability

The mangrove extent data was provided by The Global Mangrove Watch Version 3.0 (GMW)[65], downloaded from https://zenodo.org/record/6894273#.YyMn4tXMKdw. The coastline data was sourced from Natural Earth 1:110 million vector boundaries of coastline, downloaded from https://www.naturalearthdata.com/. Ecoregion data was from Marine Ecoregions of the World (MEOW) provided by World Wildlife Fund (WWF), downloaded from https://www.worldwildlife.org/publications/marine-ecoregions-of-the-world-a-bioregionalization-of-coastal-and-shelf-areas. NDVI was extracted from Landsat Collection 2 Tier-1 Surface Reflectance provided by the United States Geological Survey (USGS), including Landsat 5, 7, and 8 images; and Sentinel-2 Multi-Spectral Instrument Level 2A dataset provided by European Space Agency (ESA). Canopy height data was from two sources: Global Forest Canopy Height dataset (2019)[57] provided by University of Maryland Global Land Analysis Discovery team; downloaded from https://glad.umd.edu/dataset/gedi. Global Forest Canopy Height dataset (2005)[56] was provided by NASA's Earth Observatory. Storm data was from IBTrACS[62,63], downloaded from https://www.ncei.noaa.gov/products/international-best-track-archive. Source data are provided with this paper.

## Code availability

All analyzes in this manuscript are reproducible, and main scripts could be found in the repository "Assessment of mangrove coastal protection capacity"[66] (https://doi.org/10.5281/zenodo.13866971).

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

## Acknowledgements

This work is supported by the Strategic Priority Research Program of Chinese Academy of Sciences (Grant No. XDB0740300).

## Author contributions

Conceptualization: D.F. and X.X. Data curation: X.X. and V.L. Funding acquisition: D.F. Investigation: X.X. and V.L. Methodology: D.F. X.X., and V.L. Visualization: X.X. and H.Y. Project administration: D.F. and F.S. Software: X.X. D.F., and J.T. Writing, original draft: X.X. and V.L. Review, and editing: X.X., D.F., F.S., V.L., H.Y., J.T., X.H., and J.W. All authors subsequently reviewed the manuscript and read and approved the final version.

## Competing interests

The authors declare no competing interests.
