## [Transparent Peer Review file · Nature Communications]

Global distribution and decline of mangrove coastal protection extends far beyond area loss

Corresponding Author: Dr Dongjie Fu

Version 0:

Reviewer comments:

Reviewer #1

(Remarks to the Author)

This is an interesting study and I think the results would certainly be of interest to the wider mangrove community. The article is generally well written, but more detail is needed in places (see comments below). However, I do have some queries with the methodology, particularly the formulation of the MCPI index (see comment below) and the choice of height surface which seems to differ quite a bit from the Simard et al (2019) which focused on mangrove height. Given the importance of height for this study, I wonder whether the results might change quite a bit using the Simard et al., layer rather than the ones used in this study.

I also wonder whether biomass might be a useful metric to include? The Simard et al (2019) study shows there are significant regional differences in mangrove biomass which might also impact the protection value of those mangrove stands?

Table 2: The maximum height and other height statistics are quite different from those quoted by Simard et al (2019). Why are these so different to those from Simard et al., and how accurate are the heights used for this study? Have the sources of mangrove height been validated for mangroves? How consistent are the two height datasets in areas where change wouldn't be expected?

Simard, M., Fatoyinbo, L., Smetanka, C., Rivera-Monroy, V.H., Castañeda-Moya, E., Thomas, N., Stocken, T.V. der, 2019. Mangrove canopy height globally related to precipitation, temperature and cyclone frequency. *Nat Geosci* 12, 40-45. <https://doi.org/10.1038/s41561-018-0279-1>

Line 133. Where does a mangrove areas loss of 2.11% come from? Bunting et al., (2022) states a net loss of 3.4% (1996-2020).

Fig 5. I do not understand what this is showing in relation to the study as the mangrove have been masked out.

Line 222. How were the coastline vectors adjusted? Methods should be reproducible and this is not.

Line 228. How was the mangrove width defined with respect to gaps in the mangrove canopy. For example, it would be common for the line to have multiple mangrove boundary intersections, due to holes and/or gaps in the canopy - such as river channels. Did the gap have to be above a particular threshold for the end of the canopy to have been identified or was it just the first or last intersection? If it was the last intersection how long was the line?

Line 232. The GMW products are on a 25 m grid? How were they resampled on to a 30 m grid? If so, why were they resampled?

line 234. There is no reference for the canopy height product used.

line 241. Why NDVI? The thresholds used in Table 3 seems quite arbitrary. I would have thought that using a metric such as canopy cover would have been more appropriate as it is a physical measurement.

line 260. The formulation of the MCPI seems a little surprising to me. Given the units and range of values are quite different just multiplying them together means that they are effectively weighted. For example, NDVI values are going to be between 0-1 while heights 0-60 and widths even larger. Should the different metrics not be normalised to something before combining? Or something more physically based used?

Reviewer #2

(Remarks to the Author)

Review of: Well Beyond Area Loss: Global Declines in Intrinsic Mangrove Coastal Protection Capacity

This study created a standardized index (MCPI) to quantify protective capacity of mangrove forests based on mangrove forest width, average canopy height, and NDVI. The MCPI and its three predictors were then applied globally. The figures are very easy to understand and the methods are clearly explained. The methods and the global trends are interesting, but further depth on interpretation would improve the applicability, insight, and conclusions of the paper.

Major comments:

Why was NDVI selected as a predictor of coastal protection? It is well documented that wide forests and large trees offer better coastal protection. However, what documentation can you provide that NDVI is an important predictor of coastal protection? No references as to the importance of NDVI for quantifying protective capability are offered in the manuscript. Granted, NDVI has the benefit of being a globally available variable, and it can be an indicator of mangrove vitality. But beyond it being a useful variable for global studies, what merit does NDVI have to hold equal weight with canopy height and mangrove width for an index of protective capacity?

The authors provide an analysis on temporal changes in mangrove width, but a similar stand-alone analysis of changes in canopy height and NDVI is not provided. Thus, while there is discussion on how changes in MCPI vary around the world and in comparison with AMW, there is little information about what exactly is driving this trend, which is actually the most interesting potential point in this paper. Is it merely changes in NDVI? And is there any proof that this change in NDVI is actually impacting protective capacity of the mangroves? What is actually causing the changes in the particular areas of the world with a big discrepancy between MDPI and AMW? This question piked my curiosity in the abstract of the paper, but the real reasons for the discrepancy between MDPI and AMW was never addressed.

Line 183-186. This study on NDVI decline in the hinterlands beyond mangroves was conducted as an additional part of this study, correct? If so, this should be part of the methods and results in this paper, not first introduced in the middle of the discussion. It is not clear what area of land is being included in the NDVI comparison. This area of land needs to be standardized for distance from shore and the mangroves. This is difficult given the shape of the two regions and the inclusion of other coastal land that most likely afforded protection but does not count as mangroves in the Kendrapara province. This seems like a poor geographic choice to illustrate the utility of MCPI. It would be extremely beneficial to the manuscript to show an example of how this MCPI is useful for modeling protective capacity, but select comparable shorelines that received similar impacts from the same wind and water direction and standardize and clearly explain what hinterlands are being quantified. And of course, provide more than one example. An n=1 is not sufficiently convincing. A more robust analysis proving the utility of the MCPI would lend a lot to this manuscript (and an independent analysis of NDVI may show the importance (or not) of the metric in coastal protection).

Minor comments:

Line 22: Delete "In response to these issues," Mangroves are not present in response to the issues of coastal erosion, and the sentence structure works fine without this phrase.

Lines 29-32 "Second, tropical cyclones, storm surges and winds are hindered as mangroves attenuate surge height and water/wind flow velocity from aerial roots, trunks, and leaves, which hinders free water/wind flow through the forest^{16,17,18}. They also provide protection against wind damage during storms^{19,20}." Have the first sentence focus solely on water flow and the second sentence focus on wind flow to improve flow of the sentences.

Line 34 delete comma after "This study synthesizes aspects of mangroves intrinsic coastal protection capacity"

Line 61 Define MEOW acronym at first use here

Much of the contents of Tables 1, 2, and 3, and the accompanying text are redundant.

Line 66 "On average, AMW was relatively low." This sentence does not provide any concrete information.

Line 73: "ACH was therefore in the mid-range region." This sentence does not make sense. Are you trying to say that average ACH was within the mid-range measure of 1-10 m in height?

Line 81: "NDVI was therefore reasonably high." This sentence does not provide any concrete information or justification for its statement.

General comment: there should be a space between a number and its unit (e.g., “9 m”). This is not consistently applied throughout the manuscript and its tables. Also, there should be a space between numerical operators and numbers (e.g., “ $p = 0.002$ ”). This is also not consistently applied.

Lines 115-119: These sentences belong in the discussion, not the results.

Table 5: Suggest changing caption to “Key statistics of regions and selected areas...”

Lines 200-202: Refer to this as “tree density” rather than “planting density”, as planting density implies artificially planted forests rather than natural forests.

Line 208: Are there any references that would back up the conjecture that tree age might be significant? I would not think that age would be important compared to size and density of the trees.

Line 222-223 A citation is needed in this sentence: “First, coastline vector provided by Natural Earth was adjusted to approximate the straight line of outer wrapping of mangroves to extract target indicators.”

Line 224: Reference Fig. 6 in this sentence (it would be useful to have the reference appear earlier in the methods section)

Line 236-237: This sentence needs a citation: “Taller mangroves have more developed vertical structures with higher pneumatophore heights as better barriers to wind damage.”

Line 239 and 247: rephrase these sentences so they do not start with a lower case “m”

Fig. 7. This is a very helpful figure!

For all figures and Tables, I suggest explaining what acronyms means in the caption of the figures (or avoid the use of acronyms)

Line 262: explain and cite what regions you used. Reference Fig. 8 here.

Line 281. I understand that this methods and data availability format is specific to the journal, but I suggest mentioning at the beginning of the methods that all data origins are listed at the end of the paper so the reader is not wondering about all the unreferenced mentions of data in the methods.

Reviewer #3

(Remarks to the Author)

This paper aims to develop a mangrove coastal protection index to provide an indication of the protection capacity of mangroves to the hinterland area. I can see the value of Indexes and Indices, but they need to be constructed for a purpose, at a relevant scale and using appropriate input data. In addition, an index is a simplified modelling form and still requires testing and validation. My commentary here largely pertains to these aspects. I have also provided detailed comments below.

The purpose of this paper is not immediately apparent, with a lot of definitive text pertaining to this being the first assessment. I would urge the authors to think more deeply about the purpose of the index and how it can be applied. To this end, the authors need to justify the relevance of an index at a global scale when events (e.g. cyclones, storms surge) occur at a localized scale, as does the management that would influence the coastal protection capacity of mangroves. To this end it may be informative for the authors to develop the same index using high resolution datasets (Lidar, aerial or UAV multispectral imagery) to assess the variation in application of the index when it is applied at a scale relevant for management and decision making. An extension of this would be for the authors to consider validating the index against a series of storm events in a more quantitative manner than Fig 5 currently does. This is important for verifying the validity of the technique.

Regarding the appropriateness of input data, the index relies on 3 primary inputs: NDVI, height and forest width; however, I did not feel the authors justified well enough why these inputs relate to coastal protection. There is a lot of evidence within the literature about forest widths and their capacity to attenuate waves, but the relationship with height is less apparent. Biomass above the wave zone does not serve to offer coastal protection and may actually do the opposite with very tall trees being more prone to windthrow and instability. I suspect there is an ideal height range that maximises coastal protection capacity and the authors could consider whether the height aspects should be classified to be optimized for this height range. My rough guess is that trees taller than 10-15 m do not add much more protection capacity than trees of 10-15 m. Indeed broader/fatter, sprawling tree structures with complex biomass structure within the wave zone would maximize protection (here I am thinking large old *Avicennia* of 6-12 m). There is also some literature regarding the influence of root structure and wood density on capacity to withstand windthrow stress.

Following from this, the authors should also provide more rationalization for NDVI being included in the index. NDVI indicates the degree of ‘greenness’ and this is a factor that is hard to relate to protection capacity. I could conceive that it could be rationalized on the basis of being related to variation between species and could connect to the different capacity of species to modify the effects of storms, but this rationalization is not provided. In addition, as NDVI varies between species, it may be better to think about incremental change in NDVI, rather than the raw NDVI value, as this would provide an

indication of within species improvement or decline in condition.

Finally, I was really frustrated by the reporting of 'average' values and percentages. These tell us little about the nature of the outputs, and it is not difficult to provide additional information to the readers. To this end, the authors should look at the distribution of their results and determine whether an average or a median should be reported because the data is skewed (I suspect height, width and NDVI are all skewed). The final values for the mangrove coastal protection index could also be normalized as it is difficult to work out whether a value of 50 is good, if you do not know what the upper limit is. I also encourage the authors to think about better data visualization approaches that allow for the distribution of results to be displayed (binned distribution plots, violin or box plots).

Detailed comments:

Ln 5: What are historical forces?

Ln 8-10: Does not indicate the purpose of the MCPI, just its construction. When should this be used? What does it indicate? With this in mind, is there a rationale for these parameters?

Ln 20: rising sea levels and frequent extreme events, amongst other aspects

Ln 23: environmentally friendly environmentally sustainable. Could you also consider social aspects here too?

Ln 24: health is not what you are using. Height, extent and greenness are presumed to be indicators of health, but really it should be the temporal change in these variables that indicate health (i.e. change from a baseline)

Ln 28: mangroves occupy a range of settings with estuarine environment being a large component, but not the only. There are many open coast mangroves in this analysis, and presumably some carbonate settings too

Ln 35: 'Numerous evaluations of mangrove coastal protection' is a little contrary to statements in the abstract about being the first – there is clearly work going on

Ln 36: 'planting' density' only relevant where afforestation and reforestation is occurring

Ln 37: generally I do not think there is enough rationalization of the input component in the MCPI. Here is a missed opportunity to provide the reason for including mangrove width in the index. This needs expansion

Ln 40-41: I cannot follow this sentence. Perhaps needs rephrasing

Ln 43: I do not think scarce research is the reason for no studies of coastal protection on a global scale. I suspect that there is issues with data quality, and issues with the relevance of an analysis at this scale for decisions at the local scale. This is a location where the statement is too definitive and does not recognize the many reasons for a study of this type not occurring previously. I am not sure why the authors are pressing to make the point that this is the first.

Ln 59: AMW did not exhibit a clear pattern – but what was expected? Should we have expected a clear pattern when the development of intertidal surfaces suitable for mangrove habitations is dependent upon many coastal processes, and is modified by human activities

Ln 59-60: AMW was influenced by factors including natural conditions, national policies and land use management – could these all be reasons for why it should be excluded from the index? There is no strong rationalization provided for width, height and greenness indicating coastal protection

Ln 61: MEOW – too many acronyms – I was getting lost

Ln 61: AMW at the global scale is not a helpful measure, and I would say this for every average measurement that is relevant over large spatial scales. It would be more helpful to provide an indication of the distribution of AMW. In addition the skewed nature of AMW (more smaller values, rather than larger values) would mean that the median is a better value to report (and perhaps to use in the index).

Ln 61: HUGE mangrove widths – perhaps you could say something about why there are large mangrove widths in this region

Ln 70-71 – first instance of reporting averages in cells and ecoregions – I am not sure this distinction is clear enough Is the average in an ecoregion the average of the average of grid cells? Is this appropriate?

Ln 72: exceeding an ACH of 20 m

Ln 77: need to explain average in grid cells vs ecoregions

Ln 85-86 – repetitive. Methods detailed in methods

Ln 86-87: These values are difficult to interpret because a) they are not normalized, and b) we do not know the distributions. Reporting of an average vs a median should also be justified

Ln 97: I think the authors should provide more confidence about the independence of height and NDVI – I suspect there is some covariance that should be explored and reported

Ln 119: eastern coastline is not pronounced – this is presumably because of averaging which masks the highly variable nature along this coastline which has a very crenulated shoreline. I think more detail is needed here rather than reporting an average

Ln 129: I know the methods is where the detail is, but a bit of information about how the MCPI was compiled would be helpful here. E.g. the NDVI, ACH and AMW were multiplied with a scaling factor to indicate MCPI

Ln 130: This sentence is confusing was MDPU calculated for each year, and the change only for 2007 – 2019?

Ln 134: This sentence demonstrates the problem with indexes such as the MCPI. They compile all the input factors into a single variable which means that you cannot pull apart the primary component influencing the Index at a given location. This is the first mention of Australia – but it was not highlight in the context of NDVI, ACH and AMW in previous sections – what was the primary cause here?

Ln 143: What do numbers in brackets mean?

Ln 165-166: YES – this is because it is what the index is based on. I would prefer greater connection to processes that cause these changes being described here and linking this to the relevant literature. e.g., lower rates of clearance, high rainfall and warmer temperatures. Interestingly, these factors also correlate with the occurrence of severe cyclones, which I suspect could be discussed further

Ln 172-174 – YES this is problematic as storms operate at a more local scale, meaning this approach should be applied at the local scale using finer resolution. I think the authors should spend more time justifying doing this type of analysis at a global scale. How can it be used to be informative

Ln 175-176: The statement about the mangrove canopy is not correct. Rather, it is only the component of the biomass that is submerged that contributes to wave attenuation. For this reason, tall trees may not offer any more coastal protection services than 10 m tall trees if the wave heights from storms are < 10 m, and one could argue that they may offer more protection if the bulk of the diffusing biomass (i.e. the canopy) is within the wave zone, rather than above it.

Ln 176: Windbreak capabilities – what does this mean. This is not included in the methods. In addition, onshore winds would affect the biomass above the waves and not below the waves; and onshore winds will not modify waves that have been propagated from offshore.

Ln 209: What is co-contra-variations???

Ln 211-212: This sentence is written as if a manager may be able to influence these variables and increase coastal protection. The reality is that height is largely climatically defined and with a temporal aspect (i.e. leave trees in place until they reach maturity), and width is influenced by human factors that can be managed, but also be geomorphological factors that are difficult to modify

Ln 212: What is the difference between forest width (AMW?) and growth (is this NDVI and ACH?)

Ln 214-215: could the authors speculate as to how much change is because of species increasing in dominance that have lower NDVI and height compared to other species. E.g. Avicennia generally is lower in stature and NDVI than Rhizophora within a system.

Ln 222: Natural Earth needs a reference

Ln 222-223: I do not understand the straight line outer warping sentence – I think more detail is needed here

Ln 225-226: Was this extraction pertaining to certain years?

Ln 257-258: Reference to AMW, ACH and NDVI should come earlier and they need more rationalization of why they indicate coastal protection capacity. For example, how does greenness (NDVCI) modify coastal protection capacity (I do not think it does – rather it is correlated with species that modify capacity). How do trees of > 10 m improve coastal protection capacity compared to a tree of 10-15 m (I do not think it does, particularly as taller trees can become more unstable, and there may be less wave attenuating biomass within the wave zone).

Ln 262: S: I see scaling factors often and the only purpose they serve is to make the number easier to report (i.e. changes a small number to a large number). I think it is more critical to normalize the values (i.e. a score between 0-100) to guide comparison, and to shore the distributions of the normalized scores.

Ln 265: First time a year is mentioned in the methods; I think this is needed sooner

Ln 267: Slope of the linear regression could be confused with intertidal slope

Ln 273: Global mangrove height data from Simard from SRTM survey undertaken in 2000. Potapov et al. was with GEDI survey in 2019. May need to reflect upon whether it is appropriate to use 2007 timestep when the data is pertaining to 2000.

General comments:

- The language is more definitive at times than it should be. For example, are the authors really that confident that this is the first time a global assessment of protection capacity has occurred, of that there is currently no research on how to measure coastal protection at a global scale. Even the first sentence “mangrove have protected coastal area” is more definitive than observations would indicate.

- The grouping amongst ecoregions is very broad and allows for differences amongst species and some latitudinal trends to be differentiated. However, I would have liked to see more consideration of latitudinal patterns as others have done previously. This could be achieved by binning values across latitudes and displaying in a graph. I think all of the global mangrove height and biomass papers do this approach and I find it useful.

Figures and Tables

- There are a lot of tables reporting percentages. I suspect it would be better to convert these to graphs that provided more detail about the statistical distribution of values, and the spatial distribution of values

- Fig 2 – remove the smoothing line between points in c-g. These are discrete values that should not be connected, and not with a smoothed line.

- Fig 5: I can appreciate the point the authors are trying to make, but I do not think this figure is showing it well enough. The third arrow from the top indicates a lot of damage despite a remarkable width of mangroves

- Fig 7 bottom graph needs a y axis label (height (m)). This figure is not reference in the document anywhere. It may be the case for other figures

Version 1:

Reviewer comments:

Reviewer #1

(Remarks to the Author)

The authors have been thorough in their responses to the reviewers' comments. I believe they have adequately responded to the comments, and therefore, the manuscript is much improved.

However, I am not entirely convinced by the extended height change analysis. While the results and interpretation are plausible, I am not convinced that the two canopy height products are directly comparable and that such a trend can be reliably measured from these datasets. The two datasets were created from two different methodologies using entirely different data sources. Using just two data points, which probably have quite a considerable uncertainty about them for the analysis presented within the 'Supplementary Analyses' document, feels like a step too far. For example, Fig 4. uses these two points to back-project when this trend started, and there are a lot of assumptions being made, which I do not feel have been sufficiently backed up. I suggest that this additional height change analysis be removed and suggested as further work. It is not the focus of this article, and I feel the authors are overreaching with that further analysis, which has been carried out

as part of this revision.

Additionally, I have one minor suggestion for the main text:

Line 399: "Validation of MCPI" > "Evaluation of MCPI". I propose using 'evaluation' instead of 'validation' as it better reflects the nature of your work. 'Validation' implies the use of suitable reference dataset(s), while in this case, you are comparing trends in related datasets and then inferring whether your results align with those trends.

(Remarks on code availability)

I could not see the code, zendo said "The record is publicly accessible, but files are restricted to users with access."

Reviewer #2

(Remarks to the Author)

The authors have completed extensive revision on the analyses and its conclusions, and this revision is much changed from the first draft. The authors have meticulously responded to comments from reviewers.

Major comments on the revision:

Do you have evidence to support the theory that thinning and declining density is driving the decline in average height? As stated in the manuscript, this seems to be a theory rather than something that was quantified in any way. You suggest that it could be recovery following deforestation (line 258), but couldn't decreased height also be due to recovery following hurricane damage, mortality events due to altered hydrology or burial events, etc.? Declining density is stated as a fact in this paper, including in the abstract, but I don't see any data of density measurements. If there are data supporting thinning density (or other supporting documents in the literature), please state this more clearly. If not, be more clear that this is a theory in the paper and the abstract.

Similarly, did you complete any quantitative analyses about the theory that current strengthening is driving the decline in mangroves? If so, please highlight that data and the quantitative analyses more clearly (and add more literature sources that back up the theory). If no quantitative analyses were performed, please make it more clear that this is a theory (particularly as it seems to be stated as fact, including in the abstract). There is a pretty clear latitudinal trend in the height data in Supp Fig. 9. A lot of time is spent discussing how the EAC strengthening may be driving this trend, but little time is spent on cyclone activity and its possible influence, and there are many papers published on latitudinal trends in mangrove height and biomass in relation to cyclone activity. The influence of cyclones deserves more attention here.

Minor comments:

Line 11: "rapid global 800% growth in Cluster 8 of low height, width, and MCPI" The reader does not know about clusters yet, so avoid cluster language in favor of "mangrove forests with characteristically low height, width, and MCPI increased by 800%"

Fig. 1. Nice addition of the violin plots! This helps to show how the variables are changing over time.

Fig. 3 I suggest making the y axis on all graphs the same to enable comparison of trends across areas. Currently it is hard to make any comparisons among graphs

Line 204-205: "severest decreases were located at the southwest corner of Florida (-534.3)" You end year for comparison is 2019. Hurricane Irma hit the southwest coast of FL in 2017 and caused extensive mangrove damage. While a lot of the mangroves have recovered at this point, this is probably driving the decline at the 2019 timepoint (only 2 years post-hurricane).

Table 3 caption "Exchange of clusters is reported later." Rather than saying "later," refer the reader to the specific section of the text that explains clusters

Line 302: Perhaps recovery from mortality events (due to cyclones, drought conditions, restricted hydrology, etc.) is a reason for thinning

(Remarks on code availability)

Reviewer #3

(Remarks to the Author)

This is the second time I have looked at this manuscript and I can see that the authors have addressed the concerns I raised in my first review. The revised manuscript, with improved clarity and incorporation of biomass has improved the predictive capacity of MCP1. I particularly appreciated the additional analyses incorporated into the manuscript. I do not have further comments on the manuscript and commend the authors for the effort to address reviewer comments.

(Remarks on code availability)

I have tried to access the code below, but access is restricted.

Version 2:

Reviewer comments:

Reviewer #1

(Remarks to the Author)

The authors have provided an extensive explanation behind the points raised and appropriately revised the manuscript. I am, therefore, content that the article goes forward for publication.

(Remarks on code availability)

The code is available, and it is relatively straightforward to follow, but there is no README explaining how to use the scripts provided and the code could have more comments explaining what the blocks are doing so the user could more easily follow and use the code.

Reviewer #2

(Remarks to the Author)

The authors have once again provided meticulous responses to the reviewer requests. I appreciate the addition of the cyclone impacts on mangroves and additional analyses and feel the theories are more accurately portrayed in this revision. My requested edits have been sufficiently addressed.

(Remarks on code availability)

Responses to Reviewers

The review comments are highlighted in blue, while our responses are in **black**. Track-changes completed in the revised manuscript are shown in **purple**.

We made major modifications to the manuscript as requested to address similar critical issues raised by reviewers. These changes also led to revised insights into factors causing mangrove destruction, process-based understanding of protection factors, and drivers of destruction. We thank the reviewers for raising these issues which has led to major new and novel findings from our work. We expect these new findings will spur spirited debate and discussion amongst mangrove researchers and oceanographers.

Major changes are listed here and referenced (**R1**; **R2**; etc) in the response to reviewers:

R1: New MCPI:

The MCPI formulation was revised to a process-based approach as suggested by reviewers. We re-modelled width-protection using a wave-attenuation formulation derived from field and modelling studies. Height-protection was re-modelled to reflect biomass using classic relationships between breast-height diameter, height, and biomass. NDVI was kept the same and it was the only factor to show a consistent minor increase. This increase was reinterpreted as a dominant global greening (possibly from climate warming and greenhouse gas increase) that hid a substantial decrease related to mangrove destruction. Hence NDVI's contribution to mangrove protection was overall deceptively minimal amongst the factors, but we now know that it hides *surrogate* indications of significant destruction.

R2: New Width Formulation (lines 344-355):

For the process-based MCPI, protection from width used the field and modelling results of Zhang et al.¹⁷ as presented in Waves 2016 (Figure 2.7)⁴⁵. We fitted a wave attenuation factor A that varied with width of mangrove W (in km):

$$A = 1 - (-0.29 * W) \quad (2)$$

Where, A varies from 0 (no attenuation at $W = 0$) to 1 ($W \sim \infty$). The half-width of decay is roughly 2.3 km. So about 75% protection is achieved at ~4.6 km. These parameters contrast

with idealized laboratory attenuation studies for small waves (0.2 to 0.9 m) and short periods in shallow water, which suggest that *Rhizophora* mangroves with widths of more than 300 m (several wavelengths) are needed to reduce incoming waves by more than 50%, while widths of more than a kilometer attenuate waves by more than 80%⁴⁶. Turbulent processes are well-known to be highly nonlinear, hence our formulation of A based on Zhang et al.¹⁷ data is therefore more likely to be representative of actual conditions requiring much larger attenuation widths. But the use of width within a wave attenuation factor is well supported by both field and laboratory studies. A was used in place of AMW in MCPI.

Reference:

- 17. Zhang, K. et al. The role of mangroves in attenuating storm surges. *Estuarine, Coastal and Shelf Science* 102–103, 11–23 (2012).
- 45. Beck, M. W. & Lange, G.-M. *Managing coasts with natural solutions: Guidelines for measuring and valuing the coastal protection services of mangroves and coral reefs.* (2016).
- 46. Maza, M., Lara, J. L. & Losada, I. J. Predicting the evolution of coastal protection service with mangrove forest age. *Coastal Engineering* 168, 103922 (2021).

R3: New Height Formulation (lines 365-379):

For the process-based MCPI, we reasoned that total above ground biomass (leaves, branches, root and trunk) were all critical to moderating the effects of storms, rather than just height alone, as biomass is more representative of the total 3D frontal drag area that waves and wind have to work against. This assumes very crudely that densities of the different components of the tree are similar or can be parametrized as such within a modelling constant. We relied on the classic formulations of Cintrón and Schaeffer-Novell²⁹ for relationships between biomass, breast-height diameter, and height, with an implicit assumption that canopy leaves/branches/twigs are included in biomass:

$$B = [d^2 h]^m \quad (4)$$

Where, d is the diameter at breast-height (about 1.3 m above ground), h is the mangrove average canopy height (ACH), and the value of m from the reference²⁹ is 0.85. Hence:

$$B = d^{1.7} h^{0.85} \quad (5)$$

In turn, diameter d (cm) is approximately related to height h (m) up to about 10 m (see Table 6.7, page 111, in reference²⁹):

$$\frac{d}{h} = 0.75 * h; \quad \text{hence } d = 0.75 * h^2 \quad (6)$$

These equations can then be used to derive the Biomass (B) relation:

$$B = h^{2.55} \quad (7)$$

Noting that equation (6) is only valid up to about 10 m, after which d may grow larger, we converted the biomass relation to an index, which in effect underestimates growth of d for high

strands (hence is conservative):

$$= (h/10)^{2.55} \text{ for } h \leq 10 \text{ m; and } B = 1 \text{ for } h > 10 \text{ m} \quad (8)$$

B was used in place of ACH in MCPI.

Reference:

- 29. Snedaker, S. C. & Snedaker, J. G. *The Mangrove Ecosystem: Research Methods*. (1984).

R4: Potential issues with 2019 height data:

We very carefully investigated why the height distribution of 2019 was so radically different to 2005. This included careful spot-checks of mangrove “heights” at 3 locations around Australia (checks were visual and qualitative) using Google Earth to check distributional maps (back to 1990) and then going forward in time. These checks revealed that “average-height” as determined from our transect lines was in fact averaging across thinning strands of mangrove. In other words, the major destruction mode was one of mangroves thinning out, rather than puzzlingly reducing in height. This implies that “average-height” is indeed a surrogate of “tree density” which is the main factor subject to destruction in mangroves.

This verifies our suspicions of why mangroves were “growing” shorter as shown in the scatterplot below. If there were just linear biases between datasets (intercept-offset or a scaling factor), we would expect the scatterplot to be more linear than observed. Indeed, width and NDVI are far more linearly related between the years: width: ($adj-R = 0.997$, $slope = 0.984$); and NDVI: ($adj-R = 0.988$, $slope = 1.023$). What we observe with height is a radical *deformation* of the *statistical distribution* which suggests complex dynamic destructive processes at play (with some exceptions) that are highly dependent on height itself.

Supplementary Fig. 14 | Scatterplot of 2019 height against 2005 height. Dashed-line is 1:1 line, hence most 2019 heights are shorter.

Even more remarkable, we found a global decline in mangrove “height” especially along coasts swept by oceanic boundary currents. In fact, **all** boundary-current coasts, including the well-managed East Australian coast, experienced continuous thinning strands of mangrove since an estimated baseline-start in 1989. The destructive process was so consistent and severe that we were able to model the global destruction rate as a non-linear autoregressive process that allowed us to determine a mangrove height for full protection (13.45 m), the height where destruction was maximum (8.9 m), and height where destruction relative-rate was maximum (6.7 m). These observations and models are a significant new advance in our understanding of mangrove status and its bleak future along boundary-current coasts. In future, our global modelling methodology can be adapted to local conditions to estimate when destruction began, what local mangrove height offers full protection, and how destruction rate varies with mangrove “height”. These findings would not have been possible if not for the critical comments from reviewers and we thank the reviewers for spurring us on to dig much deeper into the data and models for answers.

R5: Factor Combination Patterns:

As an alternative to using MEOW for reporting MCPI changes, we considered analysing and summarizing changes against more-meaningful differentiated clusters of the three MCPI factors. The reasoning is that as MCPI is based on the three factors, clusters would statistically represent defined objective statistical separations of factor-combinations (as opposed to the expert-based MEOW) such that each cluster would represent a specific MCPI range related to the ranges of factors in each cluster. Changes between clusters would then provide an indicator of a *significant change* in cluster allegiance, represented by one or more factors changing outside the nominal ranges of the parent cluster.

We used a novel Cluster Exchange Network Analysis (CENA) technique with proven applications in human socio-economic geography. This approach allowed us to dissect how highly-differentiated components (network nodes) and exchanges (network links) influenced MCPI at different locations. Detailed analyses can be found in the ‘**Cluster Exchange Results**’ section of the Results and Discussion (Lines 226-289) and the ‘**Cluster Exchange Network Analysis (CENA)**’ section of the Methods (Lines 434-447).

The methodology description in the main text has been revised accordingly as follows (lines 67-72):

“To summarize and comprehend changes from 2007 to 2019, we used a novel Cluster Exchange Network Analysis (CENA) technique adapted from Lyne et al.³⁰ for studying changes in socio-economic geography. Application of CENA involved clustering normalized ACH, AMW, and NDVI for both years and then analysing changes in cluster profiles and allegiances between the two years. Cluster changes, and corresponding global geo-temporal changes of MCPI, were mapped to analyse distributional changes in factors and mangrove coastal protection capacity.”

The main results are shown in Fig.5:

Fig. 1 Characteristics and distribution maps of Clusters. (a) shows profiles of normalized factors (AMW, NDVI, ACH) with respect to the 8 Clusters for the two years, 2007 and 2019; (b) shows the mean value of the factors by Cluster for 2019; (c) and (d) represent spatial distribution of the 8 Clusters for 2007 and 2019, respectively.

Reference:

- 30. Lyne, V. et al. A typological study on stability of structures in systems: case studies from socio-economics and ecology. in 5th International Congress on Environmental Modelling and Software 2010: Modelling for Environment's Sake (eds. Swayne, D. A., Yang, W., Rizzolli, A., Voinov, A. & Filatova, T.) pp 2503:2512 (International Environmental Modelling and Software Society (iEMSs), 2010).

R6: Validation of MCPI (lines 399-416 and Supplementary Fig. 12):

The validity of global coastal protection capacity was tested by examining changes in NDVI in hinterland areas beyond mangroves before and after tropical storms. Storms causing damage to vegetation in coastal areas will decrease NDVI^{48,49}, but mangrove protection may mitigate damage, thereby reducing the decline in NDVI⁵⁰.

Global tropical storm data from International Best Track Archive for Climate Stewardship (IBTrACS)^{51,52} was utilized for this study. Initially, all tropical storm tracks intersecting with coastlines and occurring between 2017 and 2021 (extending several years before and after 2019) were selected, with maximum-sustained wind speeds (MSWS) exceeding 64 knots, representing severe tropical storms. A 5-kilometer buffer zone was generated along storm tracks to define their impact areas. Coastal mangrove areas intersected with storm-impact zones were used to identify mangrove areas affected by tropical storms. Average NDVI changes in these areas one-month before and after each storm event were calculated using Sentinel-2 imagery. Finally, the relationship between the NDVI changes caused by each storm event and the corresponding MCPI values in those regions was established.

After excluding areas with missing images, a total of 53 mangrove conservation areas that experienced severe tropical storm events were identified, as shown in Supplementary Fig. 12 and Supplementary Table 1. The results indicate a Spearman rank-order correlation coefficient (ρ) of 0.37 ($p = 0.0053$) between MCPI and NDVI changes in these areas. This demonstrates a significant monotonic positive relationship that provides support for the MCPI assessment.

Supplementary Fig. 12| Validation results for MCPI (Mangrove Coastal Protection Index). (a) depicts the relationship between the NDVI changes caused by storm events and mangrove MCPI, with a Spearman's rank-order correlation of 0.377 ($p=0.0053$), (b) demonstrates a case of poor protection effectiveness in Great Abaco, with a significant decrease in NDVI (-0.366) under the influence of the storm event 'DORIAN', and (c) illustrates the Sundarbans mangrove forest in Bangladesh, where the NDVI did not significantly decrease after the storm 'BULBUL - MATMO' and even showed overall improvement (0.071), serving as an example of effective protection.

References:

- 51. Knapp, K. R., M. C. Kruk, D. H. Levinson, H. J. Diamond, and C. J. Neumann, 2010: The International Best Track Archive for Climate Stewardship (IBTrACS): Unifying tropical cyclone best track data. *Bulletin of the American Meteorological Society*, 91, 363-376. doi:10.1175/2009BAMS2755.1
- 52. Knapp, K. R., H. J. Diamond, J. P. Kossin, M. C. Kruk, C. J. Schreck, 2018: International Best Track Archive for Climate Stewardship (IBTrACS) Project, Version 4. [since1980]. NOAA National Centers for Environmental Information. doi:10.2592/1/82ty-9e16.

We trust these over-arching answers meet expectations of the major revisions of the manuscript required by all reviewers. Responses to other concerns are addressed below, and with reference to the above revisions as **R1** to **R6**.

Responses to Reviewer #1:

*This is an interesting study and I think the results would certainly be of interest to the wider mangrove community. The article is generally well written, but more detail is needed in places (see comments below). However, I do have some queries with the methodology, particularly **the formulation of the MCPI index** (see comment below) and **the choice of height surface** which seems to differ quite a bit from the Simard et al (2019) which focused on mangrove height. Given the importance of height for this study, I wonder whether the results might change quite a bit using the Simard et al., layer rather than the ones used in this study.*

Response [1]:

Thank you so much for your recognition of our manuscript and your constructive comments. In this revision, we improved our methods and made a series of revisions to explain our methodology more explicitly. We also addressed each question individually. Please refer to the point-by-point responses above and below for the specific revisions we made to the original manuscript.

I also wonder whether biomass might be a useful metric to include? The Simard et al (2019) study shows there are significant regional differences in mangrove biomass which might also impact the protection value of those mangrove stands?

Response [2]:

Thank you so much for your constructive comments. As discussed above, we redefined the Mangrove Coastal Protection Index (MCPI) to make it more process-based (**R1; R2; R3**). We also followed your excellent suggestion and used biomass as a metric in place of height. Our new results indicate that indeed, there are significant regional variations in biomass distribution,

leading to notable differences in mangrove coastal protection capacity across different regions globally (see Fig. 2 below).

The method mentioned in **R6** was used to validate the new MCPI results. Spearman's rank-order correlation (ρ) increased from 0.365 ($p = 0.007$) to 0.377 ($p = 0.0053$), indicating a slight improvement in the new MCPI compared to the original results. Overall, the updated MCPI does not differ significantly from the original version, and it still demonstrates widespread and severe declines. This suggests that one or more of the factors are responsible for the declines.

Fig. 2 Spatial distribution of global mangrove coastal protection capacity for 2019. (a) width-based wave attenuation (A), (b) biomass-based frontal barrier (B), (c) average Normalized Difference Vegetation Index (NDVI), (d) Mangrove Coastal Protection Index (M CPI) and (e)-(h) latitude statistics of three indices and M CPI.

Table 2: The maximum height and other height statistics are quite different from those quoted by Simard et al (2019). Why are these so different to those from Simard et al., and how accurate are the heights used for this study? Have the sources of mangrove height been validated for mangroves? How consistent are the two height datasets in areas where change wouldn't be expected?

Response [3]:

Thank you so much for pinpointing the confusion about the height layer, and for your valuable suggestions. Notwithstanding our general response in **R4**, we further investigated the height data layers to answer your questions and suggestions.

Our research aims to investigate the temporal changes in global mangrove coastal protection capacity, so reliable data on temporal change of height is of great importance to us. Canopy height, serving as one of the parameters in calculating the Mangrove Canopy Protection Index (MCPI), requires data from at least two time points to assess different conditions between earlier and current periods. Thus, we selected two global forest canopy height datasets: (i) Global Forest Canopy Height (2019) from GLAD and (ii) Global Forest Canopy Height (2005) from NASA, which is sourced from Simard et al. (2011). Validation was conducted using field measurement data and GEDI validation data, and the RMSE was 6.6 m and 6.1 m respectively.

As for the difference between the height data of Simard et al. (2019) and our study, that is because Simard et al. (2019) calculated the “**maximum**” canopy height of the world, which reached 62 m, while our manuscript extracted the “**average**” canopy height within the 1° grid, so it is reasonable to be smaller. **The statistics of mean height in Simard et al. (2019) are similar to our results in 2007** (For example, Simard et al. (2019) reported an average mangrove height of 23.5 m in Gabon, whereas the average height of mangrove grids in Gabon from our results is 19.26 m; They reported 21.6 m for Equatorial Guinea, whereas we observed 18.21 m); corresponding to about ~20% difference. Besides, Simard's data is from 2000 to 2009, while the data shown in the Table 1 of our manuscript are for 2019. Temporal *changes* in canopy height over the past ten years may also lead the data to be “inconsistent” (in fact this *is* the explanation we propose in our response **R4**).

We did not directly select the data from Simard et al. (2019) because the average canopy height in the area is more meaningful for mangrove coastal protection capacity. Unfortunately, the dataset itself has not been validated specifically for mangrove areas. We know that there may be **consistency issues between the two datasets** because their algorithms and image sources are inconsistent. However, this is the only option available that covers the time span we need. **At the same time, the two height datasets are from different time periods, hence differences could legitimately arise from actual changes in tree height rather than model errors; as we suggest in response R4.**

Considering that both datasets have undergone accuracy validation, with RMSE within acceptable ranges, and they also have high citation counts (Simard et al. (2011) with 1200 citations, Potapov et al. (2019) with 623 citations), we believe their errors are acceptable.

Meanwhile, we processed the data to minimize any possible **linear systematic bias in the data**,

by normalizing the data to the range of 0-1 and then calculated the MCPI based on it.

Reference:

- Potapov, P., Li, X., Hernandez-Serna, A., Tyukavina, A., Hansen, M. C., Kommareddy, A., ... & Hofton, M. (2021). Mapping global forest canopy height through integration of GEDI and Landsat data. *Remote Sensing of Environment*, 253, 112165.
- Simard, M., Pinto, N., Fisher, J. B., & Baccini, A. (2011). Mapping forest canopy height globally with spaceborne lidar. *Journal of Geophysical Research: Biogeosciences*, 116(G4).

Line 133. Where does a mangrove areas loss of 2.11% come from? Bunting et al., (2022) states a net loss of 3.4% (1996-2020).

Response [4]:

Thank you very much for raising the question. We apologize for the lack of clarity in our explanation, which led to your confusion. As MCPI is an index, its sum value does not have physical significance, so we reported the average change across all grid cells globally. For comparison, the area change of 2.11% is also calculated as the **average** change per grid cell, representing the percentage of mangrove area loss within each grid cell. In contrast, the 3.4% reported by Bunting et al. (2022) represents the **total** global loss.

We revised this statement accordingly as follows (lines 200-201):

“From 2007 to 2019, global MCPI declined heterogeneously with an average change of -31.07 (-25.01%), which was much more severe compared to the contemporaneous average-area decline (-2.11%).”

References:

- Bunting, P., Rosenqvist, A., Hilarides, L., Lucas, R. M., Thomas, N., Tadono, T., ... & Rebelo, L. M. (2022). Global mangrove extent change 1996 -2020: Global mangrove watch version 3.0. *Remote Sensing*, 14(15), 3657.

Fig 5. I do not understand what this is showing in relation to the study as the mangrove have been masked out.

Response [5]:

Thank you for pointing out the issues with this figure. The original Fig. 5 was intended to illustrate NDVI changes in the **hinterland areas behind mangroves**, where the mangroves themselves were not masked out but color-modified to blue to distinguish them from ‘NDVI change’ values presented in the red-green color scheme. We apologize for the inappropriate handling of this color modification, which resulted in your confusion.

In the revised manuscript, we employed a more quantitative validation method for MCPI, as detailed in the Method section (please refer to revision-response **R6**). Hence, the original Fig. 5 has been removed, and Supplementary Fig. 12 has been substituted in its place.

Supplementary Fig. 12| Validation results for MCPI (Mangrove Coastal Protection Index). (a) depicts the relationship between the NDVI changes caused by storm events and mangrove MCPI, with a Spearman's rank-order correlation of 0.377 ($p=0.0053$), (b) demonstrates a case of poor protection effectiveness in Great Abaco, with a significant decrease in NDVI (-0.366) under the influence of the storm event 'DORIAN', and (c) illustrates the Sundarbans mangrove forest in Bangladesh, where the NDVI did not significantly decrease after the storm 'BULBUL - MATMO' and even showed overall improvement (0.071), serving as an example of effective protection.

Line 222. How were the coastline vectors adjusted? Methods should be reproducible and this is not.

Response [6]:

Thank you very much for your suggestion. Indeed, we acknowledge the importance of method repeatability. However, the irregular shapes of both mangroves and coastlines introduce complexity into their topological relationship, posing challenges to our task of extracting mangroves using transects. Therefore, to ensure that our transects cover all mangroves while minimizing instances of both under-sampling and over-sampling, we manually adjusted the coastlines based on the mangrove distribution extent from GMW. We will share the edited coastlines to facilitate reproducibility of our results.

The manuscript is revised accordingly as follows (lines 327-330):

“First, due to the complex topological relationship between mangroves and coastlines⁴³, we manually adjusted the coastlines to approximate the straight line of outer wrapping of mangroves based on the mangrove distribution extent from GMW. This ensures that our

sampling lines cover all mangroves and minimizes under-sampling and over-sampling.”

References:

- 43. Spencer, T., Möller, I. & Reef, R. Mangrove Systems and Environments. in *Reference Module in Earth Systems and Environmental Sciences* (Elsevier, 2016). doi:10.1016/B978-0-12-409548-9.10262-3.

Line 228. How was the mangrove width defined with respect to gaps in the mangrove canopy. For example, it would be common for the line to have multiple mangrove boundary intersections, due to holes and/or gaps in the canopy - such as river channels. Did the gap have to be above a particular threshold for the end of the canopy to have been identified or was it just the first or last intersection? If it was the last intersection how long was the line?

Response [7]:

Thank you very much for pointing out our vagueness of expression. Our method for extracting mangrove width is based on the GEE reducer: `ee.reduce.sum()`, which directly obtains the number of mangrove pixels intersecting sampling lines. The intersecting distance is calculated by multiplying the number of mangrove pixels by the sampling scale. This represents the cumulative distance for each segmented line, as illustrated in Supplementary Fig. 10 (the original Fig.6). Therefore, mangrove width is not determined solely by the distance between the first and last intersecting points, and our method does not rely on a threshold for gap intervals.

Supplementary Fig. 10 | Illustration of the method for extracting indicators using sampling lines.

We revised the description for clarity, as shown below (lines 338-340):

“Average Mangrove Width (AMW): To compute mangrove width, the number of mangrove pixels along the transect was multiplied by the sampling distance. This is represented as the

sum of lengths of multiple segmented mangrove lines along the sampling lines (as shown in Supplementary Fig. 10).”

Line 232. The GMW products are on a 25 m grid? How were they resampled on to a 30 m grid? If so, why were they resampled?

Response [8]:

Thank you very much for your comments. Indeed, we know that to be precise, the GMW products have a resolution of 25 m, but the difference between 30 m and 25 m resolutions is not significant.

In GEE, Image assets exist in the form of image pyramids at multiple scales, whereby each pixel at a given level of the pyramid is computed from the aggregation of a 2 x 2 block of pixels at the next lower level (as the average or as samples of the lower-level pixels). When using an image, GEE chooses a level of the pyramid with the closest scale less than or equal to the scale specified by users’ analysis and resamples (using nearest neighbor by default) as necessary. Therefore, when we sampled the 25 m dataset using a 30 m stride, nearest-neighbor sampling was employed.

Considering that both the height data in 2019 and Landsat images utilized in this study possess a resolution of 30 m, we opted to handle them uniformly at this resolution for ease of processing.

To further validate the rationale behind our approach, we conducted width extraction at 25 m globally for 2019 and compared the results with those extracted at 30 m. The findings indicate that the discrepancy in search resolution is negligibly small at our study scale (average error of -0.21 m and standard deviation of 4.53 m, Supplementary Fig. 15).

Supplementary Fig. 15 | Differences in Mangrove Width Extraction at 25m and 30m Resolutions. (a) depicts the spatial distribution of absolute differences in mangrove width extraction between the two resolutions, while (b) presents the histogram of value differences.

References:

- Google Earth Engine User Guide, URL: <https://developers.google.com/earth-engine/guides/scale>

line 234. There is no reference for the canopy height product used.

Response [9]:

Thank you very much for your kind suggestion. We apologize for our oversight. The reference for canopy height has been added to the revised manuscript, and specific datasets can be accessed under the Data accessibility part.

We made the revision as follows (lines 357-358):

“Average Canopy Height (ACH): Global canopy height dataset for the years 2005⁴⁷ and 2019⁴⁸ was masked using mangrove distribution range data.”

References:

- 47. Simard, M., Pinto, N., Fisher, J. B. & Baccini, A. Mapping Forest canopy height globally with spaceborne lidar. *J. Geophys. Res.* 116, G04021 (2011).
- 48. Potapov, P. et al. Mapping global forest canopy height through integration of GEDI and Landsat data. *Remote Sensing of Environment* 253, 112165 (2021).

line 241. Why NDVI? The thresholds used in Table 3 seems quite arbitrary. I would have thought that using a metric such as canopy cover would have been more appropriate as it is a physical measurement.

Response [10]:

Thank you so much for pinpointing the question of the choice of layer to represent mangrove growth condition. We did have considered the question carefully before data-processing. We finally chose the index NDVI instead of canopy cover for the following reason.

(i) NDVI, also known as the Normalized Difference Vegetation Index, is a satellite-based metric strongly associated with factors like vegetation coverage, aboveground biomass, and net primary productivity. It's widely used to evaluate ecosystem health and track vegetation growth (Chu et al., 2019; Pettorelli et al., 2005).

(ii) Indeed, canopy cover, as you mentioned, is a crucial indicator of ecosystem health. However, NDVI is closely associated with it and provides similar information. In practice, NDVI can be used to estimate canopy coverage with convincing accuracy (Carlson and Rizley, 1997; Perry et al., 2012; Tenreiro et al., 2021), and biomass as well (Meng et al., 2013; Zhu and Liu, 2015).

Therefore, we opted for NDVI, which is more readily accessible through remote sensing techniques. Higher NDVI values are typically associated with denser and healthier vegetation, while lower values may suggest vegetation loss or poor growth. Numerous studies have utilized temporal and spatial variations in NDVI to understand vegetation growth status, long-term dynamics, and their responses to environmental changes (Eisfelder et al., 2023; Gao et al., 2022; Piedallu et al., 2019).

As discussed in R3, we associated height with biomass, NDVI is still considered a semi-independent index reflecting mangrove growth/health (rather than cover which is assumed to

be related to biomass) and thus has been retained.

As for the thresholds in original Table 3, we simply intended to use them to demonstrate the distribution of the data. Following your suggestion, we made modifications and reported quartile statistics such as Q1, median, Q2 and standard deviation alternatively, see in new Table 1:

Table 1 Key statistics of AMW, A, ACH, B and MCPI

Measure	AMW (km)	A	ACH (m)	B	NDVI	MCPI
Q1	0.40	0.11	1.81	0.01	0.50	0.91
Q2 (Median)	0.77	0.20	4.13	0.10	0.61	11.99
Q3	1.64	0.38	8.88	0.74	0.71	95.09
Average	1.54	0.28	6.00	0.34	0.59	80.38
Std Dev	2.34	0.23	5.41	0.40	0.16	144.46
Maximum	25.86	0.99	28.36	1.00	0.94	743.48

References:

- Carlson, T. N., & Ripley, D. A. (1997). On the relation between NDVI, fractional vegetation cover, and leaf area index. *Remote sensing of Environment*, 62(3), 241-252.
- Chu, H., Venevsky, S., Wu, C., & Wang, M. (2019). NDVI-based vegetation dynamics and its response to climate changes at Amur-Heilongjiang River Basin from 1982 to 2015. *Science of the Total Environment*, 650, 2051-2062.
- Eisfelder, C., Asam, S., Hirner, A., Reiners, P., Holzwarth, S., Bachmann, M., ... & Kuenzer, C. (2023). Seasonal Vegetation Trends for Europe over 30 Years from a Novel Normalised Difference Vegetation Index (NDVI) Time-Series—The TIMELINE NDVI Product. *Remote Sensing*, 15(14), 3616.
- Gao, W., Zheng, C., Liu, X., Lu, Y., Chen, Y., Wei, Y., & Ma, Y. (2022). NDVI-based vegetation dynamics and their responses to climate change and human activities from 1982 to 2020: A case study in the Mu Us Sandy Land, China. *Ecological Indicators*, 137, 108745.
- Meng, J., Du, X., & Wu, B. (2013). Generation of high spatial and temporal resolution NDVI and its application in crop biomass estimation. *International Journal of Digital Earth*, 6(3), 203-218.
- Perry, E. M., Fitzgerald, G. J., Poole, N., Craig, S., & Whitlock, A. (2012). NDVI from active optical sensors as a measure of canopy cover and biomass. *The International Archives of the Photogrammetry, Remote Sensing and Spatial Information Sciences*, 39, 317-319.
- Pettorelli, N., Vik, J. O., Mysterud, A., Gaillard, J. M., Tucker, C. J., & Stenseth, N. C. (2005). Using the satellite-derived NDVI to assess ecological responses to environmental change. *Trends in ecology & evolution*, 20(9), 503-510.
- Piedallu, C., Cheret, V., Denux, J. P., Perez, V., Azcona, J. S., Seynave, I., & Gégout, J. C. (2019). Soil and climate differently impact NDVI patterns according to the season and the stand type. *Science of the Total Environment*, 651, 2874-2885.
- Tenreiro, T. R., García-Vila, M., Gómez, J. A., Jiménez-Berni, J. A., & Fereres, E. (2021). Using

NDVI for the assessment of canopy cover in agricultural crops within modelling research. Computers and Electronics in Agriculture, 182, 106038.

- Zhu, X., & Liu, D. (2015). Improving forest aboveground biomass estimation using seasonal Landsat NDVI time-series. ISPRS Journal of Photogrammetry and Remote Sensing, 102, 222-231.

line 260. The formulation of the MCPI seems a little surprising to me. Given the units and range of values are quite different just multiplying them together means that they are effectively weighted. For example, NDVI values are going to be between 0-1 while heights 0-60 and widths even larger. Should the different metrics not be normalised to something before combining? Or something more physically based used?

Response [11]:

Thank you very much for pinpointing the confusion. We apologize for the lack of clarity in our expression. In fact, we considered the issue you raised. To ensure comparability, the normalization have been done before the calculation. All three indices are scaled to 0-1 based on the maximum and minimum values, and then multiplied.

For the revised process-based version of MCPI (more physically based), the three indices A, B, and NDVI have also been scaled to the range of 0-1. Please refer to the major revision-responses **R1** to **R4** above.

We revised the description for clarity, as shown below (lines 392-394):

“The three indicators comprising horizontal structure (AMW), vertical structure (ACH), and growth condition (NDVI) were extracted at each grid cell and scaled to the range of 0-1 using the equations above, and then scaled by 1000 to calculate the mangrove coastal protection index (MCPI).”

Responses to Reviewer #2

This study created a standardized index (MCPI) to quantify protective capacity of mangrove forests based on mangrove forest width, average canopy height, and NDVI. The MCPI and its three predictors were then applied globally. The figures are very easy to understand and the methods are clearly explained. The methods and the global trends are interesting, but further depth on interpretation would improve the applicability, insight, and conclusions of the paper.

Response [1]:

Thank you so much for your careful reading and your constructive comments. In this revision, we substantially improved our methods and provided in-depth analysis to make our conclusions more meaningful as explained in general in revision-responses **R1** to **R4**. We will also address each of your questions in the point-by-point responses below.

Major comments:

Why was NDVI selected as a predictor of coastal protection? It is well documented that wide forests and large trees offer better coastal protection. However, what documentation can you provide that NDVI is an important predictor of coastal protection? No references as to the importance of NDVI for quantifying protective capability are offered in the manuscript. Granted, NDVI has the benefit of being a globally available variable, and it can be an indicator of mangrove vitality. But beyond it being a useful variable for global studies, what merit does NDVI have to hold equal weight with canopy height and mangrove width for an index of protective capacity?

Response [2]:

Thank you so much for your query on the merit of NDVI. As you suggest, generally, NDVI played only a minor role in the change between the time-period examined here. Hence its overall influence was relatively minor as changes are minimal, so it effectively only plays the role of a fixed scale factor in MCPI, rather than a changing factor. However, the situation at the local level and short timescales may not be so clear as leaves which contribute to NDVI are responding faster than height and width.

For background, NDVI, known as the Normalized Difference Vegetation Index, is a **satellite-based metric** strongly associated with factors like vegetation coverage, above-ground biomass, and net primary productivity. Currently, assessments of mangrove coastal protection capacity primarily rely on **local experimental simulations**. Therefore, there are few studies directly demonstrating the role of NDVI in protection capacity. However, NDVI can be logically associated with protective capacity based on the following reasoning.

It's widely used to evaluate ecosystem health and track vegetation growth (Chu et al., 2019; Pettorelli et al., 2005). By quantifying the difference between the near-infrared-band and the red-band, it is widely used as an indicator of vegetation health and degradation to assess environmental change (Ruan et al., 2022). Healthy vegetation absorbs red light and reflects

near-infrared light due to chlorophyll and cell structure, resulting in lower red-band reflection values and higher near-infrared-band reflection values. Unhealthy or sparse vegetation exhibits opposite properties. So, with the range of 0-1, NDVI is able to reflect vegetation conditions. Higher NDVI values are typically associated with denser and healthier vegetation, while lower values may suggest vegetation loss or poor growth (Yengoh et al., 2015).

High mangrove density enhances wave dissipation by minimizing gaps between roots and trunks, increasing interaction with waves and reducing wave height. Dense mangroves with interlocking aerial roots obstruct incoming waves and dissipate wave energy through drag force. Studies show a positive correlation between mangrove density and wave attenuation across different wave conditions, emphasizing the role of dense mangrove forests in coastal protection (Hashim et al., 2013; Iimura et al., 2012; Kamil et al., 2021; Mazda et al., 2006).

Therefore, we believe that NDVI can reflect vegetation health and density—which as noted in our revision-response **R3** is also reflected by our new biomass formulation. Hence, local NDVI does provide complementary **and** additional information on **health** to complement width and height.

References:

- Chu, H., Venevsky, S., Wu, C., & Wang, M. (2019). NDVI-based vegetation dynamics and its response to climate changes at Amur-Heilongjiang River Basin from 1982 to 2015. *Science of the Total Environment*, 650, 2051-2062.
- Hashim, A. M., & Catherine, S. M. P. (2013). A laboratory study on wave reduction by mangrove forests. *APCBEE procedia*, 5, 27-32.
- Iimura, K., & Tanaka, N. (2012). Numerical simulation estimating effects of tree density distribution in coastal forest on tsunami mitigation. *Ocean Engineering*, 54, 223-232.
- Kamil, E. A., Takaijudin, H. & Hashim, A. M. Mangroves As Coastal Bio-Shield: A Review of Mangroves Performance in Wave Attenuation. *Civil Engineering Journal* 7, 1964-1981 (2021).
- Mazda, Y., M. Magi, Y. Ikeda, T. Kurokawa and T. Asano. Wave Reduction in a Mangrove Forest Dominated by *Sonneratia* sp. *Wetlands Ecology and Management* 14, 365-378(2006). doi:10.1007/s11273-005-5388-0.
- Pettorelli, N., Vik, J. O., Mysterud, A., Gaillard, J. M., Tucker, C. J., & Stenseth, N. C. (2005). Using the satellite-derived NDVI to assess ecological responses to environmental change. *Trends in ecology & evolution*, 20(9), 503-510.
- Ruan, L., Yan, M., Zhang, L., Fan, X., & Yang, H. (2022). Spatial-temporal NDVI pattern of global mangroves: A growing trend during 2000–2018. *Science of The Total Environment*, 844, 157075.
- Yengoh, G. T., Dent, D., Ols son, L., Tengberg, A. E., & Tucker III, C. J. (2015). Use of the Normalized Difference Vegetation Index (NDVI) to assess Land degradation at multiple scales: current status, future trends, and practical considerations. Springer.

The authors provide an analysis on temporal changes in mangrove width, but a similar standalone analysis of changes in canopy height and NDVI is not provided. Thus, while there is discussion on how changes in MCPI vary around the world and in comparison with AMW, there

is little information about what exactly is driving this trend, which is actually the most interesting potential point in this paper. Is it merely changes in NDVI? And is there any proof that this change in NDVI is actually impacting protective capacity of the mangroves? What is actually causing the changes in the particular areas of the world with a big discrepancy between MDPI and AMW? This question piked my curiosity in the abstract of the paper, but the real reasons for the discrepancy between MDPI and AMW was never addressed.

Response [3]:

Thank you so much for your question on areas with MCPI and AMW discrepancies. We only conducted trend analysis on width because it has a time-series dataset covering 24 years (1996-2020). Height, on the other hand, has data for only two time periods, and as discussed NDVI plays a minor role. The discrepancy between MCPI and AMW relates to the role of “height” as explained below.

In the revised manuscript, we conducted non-linear auto-regressive modelling of height data to simulate its temporal changes where height-decay was dependent on height itself. Due to text limitations, this new modeling is briefly described in the "Height Trends" section, with detailed content available in the Supplementary Materials (Supplementary Analyses: Mangrove Destruction Trend). The model results suggest that “height” as derived from our transect and grid averaging is in fact representing “tree density” which is the main factor being “destroyed” in mangrove strands. Hence, while area may not *appear* to be changing, substantial *changes* are going on **within** the “**area**” of mangrove. Thus, area and tree-density are independent variables in differing change-dynamic dimensions.

The "Height Trends" section is as follows (lines 175-193):

“Height Trends

For height analyses, 2007 forest canopy height data were missing, hence we used available lower-resolution data for 2005. Globally, mean (median) ACH declined from 8.8 m (8.18 m) in 2005 to 6.2 m (4.48 m) in 2019, representing a (mean) decline of 2.1%/year or 29.4% (~41% biomass decline) across the 14 years. The median decline rate was higher at 3.2%/year, suggesting that distributional properties were also changing inconsistently. This represents almost a 20-fold (28-fold biomass) greater decline compared to the global width change noted in Table 2. As mangroves do not simply grow shorter, this “apparent” decline reflects a reduction in “average height” through thinning of tree density. Critically, the global nature of this decline led us to explore non-linear modelling of systematic global height destruction, which is reported in the Supplementary Materials.

Our global-scale novel quadratic nonlinear (median) height destruction model (using height data from 2005 and 2019) suggests global destruction began near 1989 when mangroves of about 13.4 m height were able to provide near-full protection. Near 8.9 m (~2003), height destruction occurred at the maximum rate of 0.41 m/year, and at 6.7 m (after 2008) the relative-rate of destruction (1/year) was at a maximum of 0.052/year, representing destruction at that height of 0.35 m/year. For lower heights (<6.7 m), relative rates of destruction declined quadratically, and asymptotically, to zero at zero height (never reached mathematically due to exponential decay). We discuss later, the regional spatial height variations using clusters which attempt to unravel regional and local inconsistencies in height destruction rates suggested by changing differences in the mean and median rates.”

Moreover, for a more in-depth analysis of the primary drivers of MCPI changes, we incorporated a novel Cluster Exchange Network Analysis (CENA) technique. The description of the cluster exchange analyses is provided in the revision-response **R5**.

Line 183-186. This study on NDVI decline in the hinterlands beyond mangroves was conducted as an additional part of this study, correct? If so, this should be part of the methods and results in this paper, not first introduced in the middle of the discussion. It is not clear what area of land is being included in the NDVI comparison. This area of land needs to be standardized for distance from shore and the mangroves. This is difficult given the shape of the two regions and the inclusion of other coastal land that most likely afforded protection but does not count as mangroves in the Kendrapara province. This seems like a poor geographic choice to illustrate the utility of MCPI. It would be extremely beneficial to the manuscript to show an example of how this MCPI is useful for modeling protective capacity, but select comparable shorelines that received similar impacts from the same wind and water direction and standardize and clearly explain what hinterlands are being quantified. And of course, provide more than one example. An n=1 is not sufficiently convincing. A more robust analysis proving the utility of the MCPI would lend a lot to this manuscript (and an independent analysis of NDVI may show the importance (or not) of the metric in coastal protection).

Response [4]:

Thank you so much for your careful reading and your constructive comments. We redesigned the method for validating MCPI by using the change in NDVI before and after the storm. We selected several severe tropical storms and standardized the storm impact areas and mangrove conservation areas. Furthermore, we moved this part to Methods and Supplementary Materials sections.

The method was updated accordingly in the manuscript (please refer to revision-response **R6**).

Minor comments:

Line 22: Delete “In response to these issues,” Mangroves are not present in response to the issues of coastal erosion, and the sentence structure works fine without this phrase.

Response [5]:

We appreciate your detailed feedback and suggestions. We made the revision as follows (lines 26-28):

“Considering these challenges, mangroves may offer economic, socially beneficial, environmentally friendly, and sustainable coastal protection, compared to engineered coastal defense structures^{8,9}.”

Lines 29-32 “Second, tropical cyclones, storm surges and winds are hindered as mangroves attenuate surge height and water/wind flow velocity from aerial roots, trunks, and leaves, which hinders free water/wind flow through the forest^{16,17,18}. They also provide protection against wind damage during storms^{19,20}. ” Have the first sentence focus solely on water flow and the second sentence focus on wind flow to improve flow of the sentences.

Response [6]:

Thank you for your thoughtful comments and recommendations. We made the revision as follows (lines 34-38):

“Second, mangroves are crucial in mitigating the impact of tropical cyclones and storm surges¹⁶. Their complex array of aerial roots, trunks, and leaves can attenuate surge heights and slow the velocity of water flow by impeding the free movement of water through the forest¹⁷⁻¹⁹. Furthermore, these resilient trees offer substantial protection against wind-damage during storms, as their foliage and branches buffer wind-flow velocity^{20,21}.”

Line 34 delete comma after “This study synthesizes aspects of mangroves intrinsic coastal protection capacity”

Response [7]:

We really appreciate your detailed feedback and suggestions. We made the corresponding revisions according to your suggestions (lines 39-40).

“This study synthesizes aspects of mangroves intrinsic coastal protection processes in order to broadly parametrize the protection ability of coastal mangroves.”

Line 61 Define MEOW acronym at first use here

Response [8]:

Thank you for your meticulous review. We apologize for our oversight. In the updated manuscript, we removed the sentence regarding MEOW. An explanation and citation are provided later at its first occurrence (see in lines 332-333):

“The sampling line was determined from actual growth of mangroves in each region, using ecoregions from Marine Ecoregions of the World (MEOW)⁴⁴ (shown in Supplementary Fig. 13)”

Supplementary Fig. 13 Ecoregions within different realms relative to the study from Marine Ecoregions of the World (MEOW).

Much of the contents of Tables 1, 2, and 3, and the accompanying text are redundant.

Response [9]:

Thank you for your valuable comments. These tables are intended to simplify the text and make it easier for readers to access data information. In the updated manuscript, we integrated these tables into one by removing redundant information and adding statistics like median and standard deviation to illustrate data distribution (Table 1):

Table 2 Key statistics of AMW, A, ACH, B and MCPI

Measure	AMW (km)	A	ACH (m)	B	NDVI	MCPI
Q1	0.40	0.11	1.81	0.01	0.50	0.91
Q2 (Median)	0.77	0.20	4.13	0.10	0.61	11.99
Q3	1.64	0.38	8.88	0.74	0.71	95.09
Average	1.54	0.28	6.00	0.34	0.59	80.38
Std Dev	2.34	0.23	5.41	0.40	0.16	144.46
Maximum	25.86	0.99	28.36	1.00	0.94	743.48

Line 66 “On average, AMW was relatively low. ” This sentence does not provide any concrete information.

Line 73: “ACH was therefore in the mid-range region.” This sentence does not make sense. Are you trying to say that average ACH was within the mid-range measure of 1-10 m in height?

Line 81: “NDVI was therefore reasonably high.” This sentence does not provide any concrete

information or justification for its statement.

Response [10]:

Thank you for your detailed feedback and constructive criticism. These expressions are indeed not specific enough. In the revised manuscript, we removed these vague statements and replaced them with more-informative violin graphs to visually display the data distribution (see Fig.1).

Fig. 1 Statistical “violin” with included Box-whiskers plot. The violin-width indicates the frequency distribution of data points at that variable-value—indicated by the y-axis. Data points are plotted with horizontal “jitter” to prevent overlap. The box represents the median (Q2) and interquartile range (IQR) between the 25th (Q1) and 75th (Q3) percentiles. Variables are: (a) Average cross-shore Mangrove Width (AMW), (b) NDVI, and (c) Average Canopy Height (ACH), plotted for the two years.

General comment: there should be a space between a number and its unit (e.g., “9 m”). This is not consistently applied throughout the manuscript and its tables. Also, there should be a space between numerical operators and numbers (e.g., “ $p = 0.002$ ”). This is also not consistently applied.

Response [11]:

Thank you for your meticulous review. We apologize for our oversight and have made the corresponding revisions according to your suggestions. This issue is also rectified throughout our updated manuscript.

Lines 115-119: These sentences belong in the discussion, not the results.

Response [12]:

Thank you for your constructive suggestions. We made the corresponding revisions according to your suggestions. In order to present our results and discussion more effectively, we merged the Results and Discussion sections into one cohesive section titled "**Results and Discussion**" in the revised manuscript.

Table 5: Suggest changing caption to “Key statistics of regions and selected areas...”

Response [13]:

Thank you for your thoughtful comments and recommendations. We made the corresponding revisions according to your suggestions (see in line 167):

“Table 2 Key statistics of regions and selected areas for mangrove width change.”

Lines 200-202: Refer to this as “tree density” rather than “planting density”, as planting density implies artificially planted forests rather than natural forests.

Response [14]:

Thank you for your thoughtful comments and recommendations. We made the corresponding revisions according to your suggestions. In other parts of the manuscript, we used the term "tree density" instead of "planting density".

As for the sentence, we deleted it in the revised version.

Line 208: Are there any references that would back up the conjecture that tree age might be significant? I would not think that age would be important compared to size and density of the trees.

Response [15]:

Thank you so much for your careful reading and your constructive comments. The age of trees does influence their ability to protect coastlines. As mature trees have larger trunk diameters and more firmly rooted bases, they become increasingly resistant to wave damage with age (Alongi et al., 2008; Danielsen et al., 2005; Kamil et al., 2021). However, we lack globally suitable data on tree age. Therefore, we did not include tree age in our study; it is only mentioned in the discussion section.

Nonetheless, as you rightly point out, the information it provides overlaps with tree height and growth conditions. In the revised manuscript, we modified the relevant sentence as follows (lines 297-298):

“In future, additional indicators and local-adaptations of our MCPI model could be used to enhance local applications.”

References:

- Alongi, D. M. “Mangrove Forests: Resilience, Protection from Tsunamis, and Responses to Global Climate Change.” *Estuarine, Coastal and Shelf Science* 76 (2008): 1-13. doi:10.1016/j.ecs.2007.08.024.
- Danielsen, F., M. K. Sørensen, M. F. Olwig, V. Selvam, F. Parish, N. D. Burgess, T. Hiraishi, V. M. Karunakaran, M. S. Rasmussen, L. B. Hansen, A. Quarto and N. Suryadiputra. “The Asian Tsunami: A Protective Role for Coastal Vegetation.” *Science* 310 (2005): 643. doi:10.1126/science.1118387.
- Kamil, E. A., Takaijudin, H. & Hashim, A. M. Mangroves As Coastal Bio-Shield: A Review of Mangroves Performance in Wave Attenuation. *Civil Engineering Journal* 7, 1964–1981 (2021).

Line 222-223 A citation is needed in this sentence: “First, coastline vector provided by Natural Earth was adjusted to approximate the straight line of outer wrapping of mangroves to extract target indicators.”

Response [16]:

Thank you very much for your comment. We revised the manuscript accordingly. A citation was added and more detailed explanations of our approach to coastline processing are provided.

The manuscript was revised accordingly as follows (lines 327-330):

“First, due to the complex topological relationship between mangroves and coastlines⁴³, we manually adjusted the coastlines to approximate the straight line of outer wrapping of mangroves based on the mangrove distribution extent from GMW. This ensures that our sampling lines cover all mangroves and minimizes under-sampling and over-sampling.”

References:

- 43. Spencer, T., Möller, I., & Reef, R. (2016). Mangrove systems and environments. In Reference Module in Earth Systems and Environmental Sciences. Elsevier.

Line 224: Reference Fig. 6 in this sentence (it would be useful to have the reference appear earlier in the methods section)

Response [17]:

Thank you for your thoughtful comments and recommendations. Based on your suggestion, we have moved the reference to the original Fig. 6 (now Supplementary Fig. 10) forward. The manuscript was revised as follows (lines 330-332):

“Second, equally-spaced transect lines perpendicular to shoreline were generated at 1 km intervals based on the updated coastline to extract mangrove transect-statistics (shown in Supplementary Figs. 10 and 11).”

Line 236-237: This sentence needs a citation: “Taller mangroves have more developed vertical structures with higher pneumatophore heights as better barriers to wind damage.”

Response [18]:

Thanks for your thoughtful comments and recommendations. Considering your feedback as well as that of other reviewers, we recognized that our previous treatment of height was indeed inadequate. Therefore, we adjusted the treatment of height, using it as an approximation to model biomass to evaluate protection capacity. Using the method mentioned in **Response [4]** and revision-response **R6** to validate the new MCPI results, Spearman's ρ increased from 0.365 ($p = 0.007$) to 0.377 ($p = 0.0053$), indicating a slight improvement in the new MCPI compared to the original results. Overall, the updated MCPI does not differ significantly from the original version, and stills demonstrates widespread and severe declines.

As for the sentence, we deleted it from the revised manuscript, and instead provided detailed explanations of the new treatment method for the height index in its place, along with added references. Please refer to the revision-response **R3**.

Line 239 and 247: rephrase these sentences so they do not start with a lower case “m”

Response [19]:

Thanks for your thoughtful comments and recommendations.

The sentences were rephrased (lines 361-364 and 386-388):

“Where, ACH_k represents average canopy height in k^{th} region; $CH_{i,j,k}$ denotes canopy height of i^{th} pixel crossed by j^{th} sampling line in k^{th} region; m signifies the total number of pixels containing mangroves along the j^{th} sampling line, while n represents the total number of sampling lines containing mangroves in the k^{th} region.

Where, ρ_{nir} represents near-infrared reflectance; ρ_r represents red-band reflectance; $NDVI_k$ is average NDVI in k^{th} region; $NDVI_{i,j,k}$ is NDVI for i^{th} pixel along j^{th} sampling line in k^{th} region; m is total count of mangrove pixels for j^{th} sampling line, and n is number of sampling lines containing mangroves in k^{th} region.”

Fig. 7. This is a very helpful figure!

Response [20]:

Thanks for your recognition. We are glad the figure is helpful.

For all figures and Tables, I suggest explaining what acronyms means in the caption of the figures (or avoid the use of acronyms)

Response [21]:

Thanks for your thoughtful comments and recommendations. We made the corresponding revisions according to your suggestions (see Fig.1-5 and Table 1, 3):

Fig. 1 Statistical “violin” with included Box-whiskers plot. The violin-width indicates the frequency distribution of data points at that variable-value—indicated by the y-axis. Data points are plotted with horizontal “jitter” to prevent overlap. The box represents the median (Q2) and interquartile range (IQR) between the 25th (Q1) and 75th (Q3) percentiles. Variables are: (a) Average cross-shore Mangrove Width (AMW), (b) NDVI, and (c) Average Canopy Height (ACH), plotted for the two years.

Fig. 2 Spatial distribution of global mangrove coastal protection capacity for 2019. (a) width-based wave Attenuation (A), (b) biomass-based frontal Barrier (B), (c) average Normalized Difference Vegetation Index (NDVI), (d) Mangrove Coastal Protection Index (MCPI) and (e)-(h) latitude statistics of three indices and MCPI.

Fig. 3 Results of trend regression analysis for Average cross-shore Mangrove Width (AMW) based on data from 1996 to 2020. (a) and (b) represent the spatial patterns of annual mangrove width change rates extracted using $1^\circ \times 1^\circ$ grids and national boundaries, respectively. The unit is meters per year. (c) depicts global average trend of mangrove width. (d), (e), (f), and (g) represent width change trends in the Americas, Africa, Asia, and Oceania, respectively.

Fig. 4 Spatial variations in MCPI (Mangrove Coastal Protection Index) changes from 2007 and 2019

(with canopy-height data from 2005). (a), (b) represent spatial distribution of change values and annual change rates respectively.

Fig. 5 Characteristics and distribution maps of Clusters. (a) shows profiles of normalized factors (Average cross-shore Mangrove Width, Average Canopy Height and Normalized Difference Vegetation Index) with respect to the 8 Clusters for the two years, 2007 and 2019; (b) shows the mean value of the factors by Cluster for 2019; (c) and (d) represent spatial distribution of the 8 Clusters for 2007 and 2019, respectively.

Table 1 Key statistics of AMW (Average cross-shore Mangrove Width), A (width-based wave Attenuation), ACH (Average Canopy Height), B (Biomass-based frontal barrier), NDVI, and MCPI (Mangrove Coastal Protection Index). Q values refer to quartiles (Q1 is 25%, Q2 is 50%, and Q3 is 75%).

Table 3 Key statistics of typical areas in MCPI (Mangrove Coastal Protection Index) and mangrove area change (the typical regions are several specific grids in the area, chosen by the extremum of MCPI change and Area change). Exchange of clusters is reported later.

Line 262: explain and cite what regions you used. Reference Fig. 8 here.

Response [22]:

Thanks for your thoughtful suggestions. Please allow us to explain. Our data processing involves two levels of partitioning:

- (i) The first level consists of the ecoregions depicted in the original Figure 8. There are a total of 90 ecoregions, distributed across 10 different realms, as represented by various colors in the original Figure 8. We utilize the distinct growth conditions of mangroves in these ecoregions to determine the length of the sampling lines. The ecoregions and realms are derived from the Marine Ecoregions of the World (MEOW) dataset (Spalding et al., 2007).
- () The second level consists of tiles, which serve as the actual units for our data processing and are the units in which the final results are presented. The term "region" mentioned in original Line 262 refers to the mangrove tiles, sourced from the Global Mangrove Watch (GMW) dataset (Bunting et al., 2022). The calculation of MCPI for 2019 involves a total of 1,215 mangrove tiles.

According to your suggestion, we added a reference to the original Figure 8 at the beginning of the Method section to enhance the clarity of our presentation (lines 332-333):

“The sampling line was determined from actual growth of mangroves in each region, using ecoregions from Marine Ecoregions of the World (MEOW) ⁴⁴ (shown in Supplementary Fig. 13)”

References:

- Bunting, Pete et al. Global Mangrove Watch (1996 - 2020) Version 3.0 Dataset. Zenodo <https://doi.org/10.5281/ZENODO.6894273> (2022).
- Spalding, Mark D., et al. Marine ecoregions of the world: a bioregionalization of coastal and shelf areas. *BioScience* 57.7, 573-583(2007).

Line 281. I understand that this methods and data availability format is specific to the journal, but I suggest mentioning at the beginning of the methods that all data origins are listed at the end of the paper so the reader is not wondering about all the unreferenced mentions of data in the methods.

Response [23]:

Thank you for your thoughtful comments and recommendations. Revisions were made according to your suggestions. We added the statement (line 335):

“All data used are listed in the Data Availability section.”

Responses to Reviewer #3:

This paper aims to develop a mangrove coastal protection index to provide an indication of the protection capacity of mangroves to the hinterland area. I can see the value of Indexes and Indices, but they need to be constructed for a purpose, at a relevant scale and using appropriate input data. In addition, an index is a simplified modelling form and still requires testing and validation. My commentary here largely pertains to these aspects. I have also provided detailed comments below.

Response [1]:

Thank you so much for your careful reading and your constructive comments. In this revision, we improved our methods and revised our methodology more explicitly. We also provided in-depth analyses, including at the local level, to make our conclusions more meaningful. We addressed each of your questions individually. Please refer to the point-by-point responses below for the specific revisions we made to the original manuscript.

*The **purpose** of this paper is not immediately apparent, with a lot of definitive text pertaining to this being the first assessment. I would urge the authors to think more deeply about the purpose of the index and how it can be applied. To this end, the authors need to justify the relevance of an index at a global scale when events (e.g. cyclones, storms surge) occur at a localized scale, as does the management that would influence the coastal protection capacity of mangroves. To this end it may be informative for the authors to develop the same index using high resolution datasets (Lidar, aerial or UAV multispectral imagery) to assess the variation in application of the index when it is applied at a scale relevant for management and decision making. An extension of this would be for the authors to consider validating the index against a series of storm events in a more quantitative manner than Fig 5 currently does. This is important for verifying the validity of the technique.*

Ln 172-174 – YES this is problematic as storms operate at a more local scale, meaning this approach should be applied at the local scale using finer resolution. I think the authors should spend more time justifying doing this type of analysis at a global scale. How can it be used to be informative

Response [2]:

Thank you for your thoughtful comments and recommendations. To avoid making the response too lengthy, we have consolidated the similar questions you raised above and provided a unified response. You raise conceptual, modelling, scale, and application-relevance issues in the MCPI formulation which we address broadly below and then with reference to specific issues.

Conceptual Issues

Process-based MCPI: Our revised MCPI formulation is process-based as described in revision-responses R1 to R5. In addition, our analyses indicate somewhat independent changes in height, width, and NDVI (which displays both a global greening not evident in other factors

and a hidden potential destruction component related to height). We trust that these revisions address your concerns regarding a more purposeful index of process-based factors.

Factor Combinations: Complexities by their very nature have similar and/or differing relationships and dynamics depending on the scale of the problem, ranging all the way from global to “local”. The finest level is at the scale of the data which retains all the complexities but it needs to be organized into meaningful aggregated trends and patterns that can be used for planning and action. We chose to focus on global and regional scales to compare our index and contrast it against the singular index/factor based on area.

However, we go well beyond this initial objective as our revised analyses clearly show that it is the makeup of changes within areas that are far more dramatic than actual area change itself. This implies that **regardless of scale**, be it global or local, area changes by themselves are a deceptive index of significant changes in mangrove structure and composition. In our revision, we attempt to unpack the cause of those changes across the three factors and show how local variations aggregate and manifest as changes across regions and the globe. This in effect is what hierarchical tree-based statistics attempts in organizing complex associations and differences amongst data at the bottom of the tree. We therefore implemented the cluster analyses described in revision-response **R5** (but ended using only one level of clustering as this was sufficient for our purposes). This provided us with the global-to-cluster connection that we sought, which helped to explain the complex changes more reasonably across factors, across time, and across spatial scales. For example, the role of ocean boundary currents became apparent once we saw the mapped changes in the poor-performing Cluster 8 (see map in R5, Fig. 5).

We agree that storm events occur at a local scale and that mangrove coastal protection capacity may vary for different events, considering factors such as storm intensity, wave height, and local environment. So, scaling these local issues up to the regional/global-scale is equivalent to asking if the frequency and intensity of events and process factors have imprints across regions and the globe. In other words, our analyses show how we can scale up local observations to larger scales.

Our use of "first" refers to the **first globally consistent evaluation** of mangrove coastal protection capacity. And with our revisions, we are justified in claiming that we understand more clearly why protection capacity has declined much more sharply than area declines. Hence, despite the regional/local variations mentioned above, we believe that our process-based standardized assessment remains meaningful globally, regionally, and in relation to climate change.

According to your suggestions, we employed a more quantitative verification of MCPI by conducting a correlation analysis between the impacts of 53 global storm events and mangrove MCPI. Please refer to the Methods section for detailed information (revision-response **R6**):

(Lines 399-416 and Supplementary Fig. 12)

“The validity of global coastal protection capacity was tested by examining changes in NDVI in hinterland areas beyond mangroves before and after tropical storms. Storms causing damage to vegetation in coastal areas will decrease NDVI^{49,50}, but mangrove protection may mitigate damage⁵¹, thereby reducing the decline in NDVI.

Global tropical storm data from International Best Track Archive for Climate Stewardship (IBTrACS)^{52,53} was utilized for this study. Initially, all tropical storm tracks intersecting with coastlines and occurring between 2017 and 2021 (extending several years before and after 2019) were selected, with maximum-sustained wind speeds (MSWS) exceeding 64 knots, representing severe tropical storms. A 5-kilometer buffer zone was generated along storm tracks to define their impact areas. Coastal mangrove areas intersected with storm-impact zones were used to identify mangrove areas affected by tropical storms. Average NDVI changes in these areas one-month before and after each storm event were calculated using Sentinel-2 imagery. Finally, the relationship between the NDVI changes caused by each storm event and the corresponding MCPI values in those regions was established.

After excluding areas with missing images, a total of 53 mangrove conservation areas that experienced severe tropical storm events were identified, as shown in Supplementary Fig. 12 and Supplementary Table 1. The results indicate a Spearman rank-order correlation coefficient (ρ) of 0.37 ($p = 0.0053$) between MCPI and NDVI changes in these areas. This demonstrates a significant monotonic positive relationship that provides support for the MCPI assessment. ”

Supplementary Fig. 12| Validation results for MCPI (Mangrove Coastal Protection Index). (a) depicts the relationship between the NDVI changes caused by storm events and mangrove MCPI, with a Spearman's rank-order correlation of 0.377 ($p=0.0053$), (b) demonstrates a case of poor protection effectiveness in Great Abaco, with a significant decrease in NDVI (-0.366) under the influence of the storm event 'DORIAN', and (c) illustrates the Sundarbans mangrove forest in Bangladesh, where the NDVI did not significantly decrease after the storm 'BULBUL - MATMO' and even showed overall improvement (0.071), serving as an example of effective protection.

References:

- Knapp, K. R., M. C. Kruk, D. H. Levinson, H. J. Diamond, and C. J. Neumann, 2010: The International Best Track Archive for Climate Stewardship (IBTrACS): Unifying tropical cyclone best track data. *Bulletin of the American Meteorological Society*, 91, 363-376. doi:10.1175/2009BAMS2755.1
- Knapp, K. R., H. J. Diamond, J. P. Kossin, M. C. Kruk, C. J. Schreck, 2018: International Best Track Archive for Climate Stewardship (IBTrACS) Project, Version 4. [since1980]. NOAA National Centers for Environmental Information. doi:10.2592/1/82ty-9e16.

Regarding the appropriateness of input data, the index relies on 3 primary inputs: NDVI, height and forest width; however, I did not feel the authors justified well enough why these inputs relate to coastal protection. There is a lot of evidence within the literature about forest widths and their capacity to attenuate waves, but the relationship with height is less apparent. Biomass above the wave zone does not serve to offer coastal protection and may actually do the opposite with very tall trees being more prone to windthrow and instability. I suspect there is an ideal height range that maximises coastal protection capacity and the authors could consider whether the height aspects should be classified to be optimized for this height range. My rough guess is that trees taller than 10-15 m do not add much more protection capacity than trees of 10-15 m. Indeed broader/fatter, sprawling tree structures with complex biomass structure within the wave zone would maximize protection (here I am thinking large old Avicennia of 6-12 m). There is also some literature regarding the influence of root structure and wood density on capacity to withstand windthrow stress.

Ln 175-176: The statement about the mangrove canopy is not correct. Rather, it is only the component of the biomass that is submerged that contributes to wave attenuation. For this reason, tall trees may not offer any more coastal protection services than 10 m tall trees if the wave heights from storms are < 10 m, and one could argue that they may offer more protection if the bulk of the diffusing biomass (i.e. the canopy) is within the wave zone, rather than above it.

Ln 176: Windbreak capabilities – what does this mean. This is not included in the methods. In addition, onshore winds would affect the biomass above the waves and not below the waves; and onshore winds will not modify waves that have been propagated from offshore.

Response [3]:

Thank you very much for your constructive comments. Based on your suggestion and similar concerns from the other reviewers, we refined the construction of the Mangrove Coastal Protection Index (MCPI) to make it more physically meaningful as explained in revision-responses **R1** to **R5**. To avoid making the response too lengthy, we have consolidated the similar questions you raised above and provided a unified response.

For the width aspect, we fitted a width-based wave attenuation factor A using the field and modelling results of Zhang et al. (2012). Regarding the height indicator, it's worth noting that the coastal protection considered in this study includes, but is not limited to, wave attenuation. The wind protection effect provided by dense foliage is also encompassed within our concept of coastal protection from biomass.

However, considering the “ideal height range” you suggested, we recognized that our previous treatment of height was indeed inadequate. We redefined the treatment of height, using it as an approximation to model biomass to evaluate protection capacity—see revision-response R3. Following your suggestion we also set a height threshold of 10 meters, assuming that trees taller than 10 meters do not provide greater protection than those at 10 meters. Notwithstanding your comments about heights higher than wave height being less relevant, our detailed non-linear modelling of height reported in the Supplementary Materials (see **Mangrove Destruction Trend**) identified the full-protection height as 13.45 m and that **maximum height destruction** occurred at 8.9 m, hence 10 m is also within the height band of **maximum destruction** rather than **maximum protection**. Thus, contrary to notions about wave versus mangrove height related to protection, it is destruction of **tree density** as waves and currents work their way **between** trees that is the main causal agent of mangrove destruction.

Clearly, more work is required to understand the hydrodynamic forces at play but one suggestion is that, like a garden hose, clamping down on the spray nozzle opening (as an analogy of the opening between trees) alters the jet pressure/force. Newton’s first law also tells us that high attenuation comes at the expense of high forces being exerted upon the frontal elements causing attenuation. Putting these two thoughts together suggests these high forces appear to be **hydraulically eroding** the surrounding soil/root-holds around trees, rather than bundling trees over from brute force alone. Our visual checks appear to qualitatively verify this as the main destructive mechanism, as does the quantitative “height” changes.

Our model suggests that it is only below 6.7 m (or more accurately, height averaged out to 6.7 m) when the relative rate of destruction starts to decline. The model also establishes a baseline time of around 1989 when destruction commenced globally. We debated amongst ourselves about whether we should report the “alarming” MCPI changes from the 1989 baseline or not. In the end we decided to just report the 2007 to 2019 changes, as we were not brave enough to provoke the elephant sitting uncomfortably in the room.

For specific details, please refer to the revision-responses **R1** to **R5**.

References:

- Lee, W. K., Tay, S. H. X., Ooi, S. K. & Friess, D. A. Potential short wave attenuation function of disturbed mangroves. *Estuarine, Coastal and Shelf Science* 248, 106747 (2021).
- Snedaker, S. C. & Snedaker, J. G. *The Mangrove Ecosystem: Research Methods*. (1984).
- Zhang, K. et al. The role of mangroves in attenuating storm surges. *Estuarine, Coastal and Shelf Science* 102–103, 11–23 (2012).

Following from this, the authors should also provide more rationalization for NDVI being

*included in the index. NDVI indicates the degree of 'greenness' and this is a factor that is hard to relate to protection capacity. I could conceive that it could be rationalized on the basis of being related to variation between species and could connect to the different capacity of species to modify the effects of storms, but this rationalization is not provided. In addition, as NDVI varies between species, it may be better to think about **incremental change in NDVI**, rather than the raw NDVI value, as this would provide an indication of within species improvement or decline in condition.*

Ln 24: health is not what you are using. Height, extent and greenness are presumed to be indicators of health, but really it should be the temporal change in these variables that indicate health (i.e. change from a baseline)

Response [4]:

Thank you so much for your careful reading and your constructive comments. To avoid making the response too lengthy, we have consolidated the similar questions you raised above and provided a unified response. Our explanations are as follows.

NDVI, known as the Normalized Difference Vegetation Index, is a satellite-based metric strongly associated with factors like vegetation coverage, aboveground biomass, and net primary productivity. It's widely used to evaluate ecosystem health and track vegetation growth (Chu et al., 2019; Pettorelli et al., 2005). By quantifying the difference between the near-infrared-band and the red-band, it is widely used as an indicator of **vegetation health and degradation** to assess environmental change (Ruan et al., 2022). Healthy vegetation absorbs red light and reflects near-infrared light due to chlorophyll and cell structure in the leaves, resulting in lower red-band reflection values and higher near-infrared-band reflection values. Unhealthy or sparse vegetation exhibits opposite properties. So, with the range of 0-1, NDVI is able to reflect vegetation conditions in pixels; Higher NDVI values are typically associated with **denser and healthier vegetation**, while lower values may suggest vegetation loss or poor growth (Yengoh et al., 2015).

High mangrove **density** enhances wave dissipation as narrower gaps between roots and trunks, increasing waves, thereby reducing wave-height by dissipating wave energy through drag force. Studies show a positive correlation between mangrove density and wave attenuation across different wave conditions, emphasizing the role of dense mangrove forests in coastal protection (Kamil et al., 2021; Mazda et al., 2006; Hashim et al., 2013; Imura et al., 2012). Therefore, NDVI can reflect vegetation health and density, hence provides complementary and supplementary information for coastal protection capability beyond width and height.

Considering the diversity of mangrove species, the change in NDVI is indeed more meaningful than NDVI itself. However, NDVI itself can still reflect vegetation health and density based on the above reasoning. Plus, according to our results, NDVI played only a minor role in the change between the time-period examined here. Hence its overall influence was relatively minor as changes are minimal, so it effectively only plays the role of a fixed scale factor in MCPI, rather than a changing factor. Taking into account existing studies (Ruan et al., 2022) that have used NDVI as a measure of mangrove health for global-scale research, we opted to

include NDVI as an indicator to obtain additional information beyond width and height.

Additionally, we computed MCPI for two time periods, and the changes in NDVI that you are interested in will be reflected in minor changes in MCPI as you suspected. But it could be of relevance where local mangrove health is severely compromised from local loss of leaves and leaf-greenness.

References:

- Chu, H., Venevsky, S., Wu, C., & Wang, M. (2019). NDVI-based vegetation dynamics and its response to climate changes at Amur-Heilongjiang River Basin from 1982 to 2015. *Science of the Total Environment*, 650, 2051-2062.
- Hashim, A. M., & Catherine, S. M. P. (2013). A laboratory study on wave reduction by mangrove forests. *APCBEE procedia*, 5, 27-32.
- Imura, K., & Tanaka, N. (2012). Numerical simulation estimating effects of tree density distribution in coastal forest on tsunami mitigation. *Ocean Engineering*, 54, 223-232.
- Kamil, E. A., Takaijudin, H. & Hashim, A. M. Mangroves As Coastal Bio-Shield: A Review of Mangroves Performance in Wave Attenuation. *Civil Engineering Journal* 7, 1964-1981 (2021).
- Mazda, Y., M. Magi, Y. Ikeda, T. Kurokawa and T. Asano. Wave Reduction in a Mangrove Forest Dominated by *Sonneratia* sp. *Wetlands Ecology and Management* 14, 365-378(2006). doi:10.1007/s 11273-005-5388-0.
- Pettorelli, N., Vik, J. O., Mysterud, A., Gaillard, J. M., Tucker, C. J., & Stenseth, N. C. (2005). Using the satellite-derived NDVI to assess ecological responses to environmental change. *Trends in ecology & evolution*, 20(9), 503-510.
- Ruan, L., Yan, M., Zhang, L., Fan, X., & Yang, H. (2022). Spatial-temporal NDVI pattern of global mangroves: A growing trend during 2000-2018. *Science of The Total Environment*, 844, 157075.
- Yengoh, G. T., Dent, D., Ols son, L., Tengberg, A. E., & Tucker III, C. J. (2015). Use of the Normalized Difference Vegetation Index (NDVI) to assess Land degradation at multiple scales: current status, future trends, and practical considerations. Springer.

Finally, I was really frustrated by the reporting of 'average' values and percentages. These tell us little about the nature of the outputs, and it is not difficult to provide additional information to the readers. To this end, the authors should look at the distribution of their results and determine whether an average or a median should be reported because the data is skewed (I suspect height, width and NDVI are all skewed). The final values for the mangrove coastal protection index could also be normalized as it is difficult to work out whether a value of 50 is good, if you do not know what the upper limit is. I also encourage the authors to think about better data visualization approaches that allow for the distribution of results to be displayed (binned distribution plots, violin or box plots).

Ln 61: AMW at the global scale is not a helpful measure, and I would say this for every average measurement that is relevant over large spatial scales. It would be more helpful to provide an

indication of the distribution of AMW. In addition the skewed nature of AMW (more smaller values, rather than larger values) would mean that the median is a better value to report (and perhaps to use in the index).

Ln 86-87: These values are difficult to interpret because a) they are not normalized, and b) we do not know the distributions. Reporting of an average vs a median should also be justified

Figures and Tables

- There are a lot of tables reporting percentages. I suspect it would be better to convert these to graphs that provided more detail about the statistical distribution of values, and the spatial distribution of values

Response [5]:

Thank you very much for your constructive and encouraging comments. To avoid making the response too lengthy, we have consolidated the similar questions you raised above and provided a unified response.

We apologize for not providing satisfactory data presentation in the original version. We made improvements in the revised version. In the updated manuscript, we integrated the original Tables 1- 4 into one and reported quartile statistics Q1, median, Q3, and standard deviation. We provide violin graphs to visually display the data distribution (new Table 1 and Fig.1):

Table 1 Key statistics of AMW (Average cross-shore Mangrove Width), A (width-based wave Attenuation), ACH (Average Canopy Height), B (Biomass-based frontal barrier), NDVI, and MCPI (Mangrove Coastal Protection Index). Q values refer to quartiles (Q1 is 25%, Q2 is 50%, and Q3 is 75%).

Measure	AMW (km)	A	ACH (m)	B	NDVI	MCPI
Q1	0.40	0.11	1.81	0.01	0.50	0.91
Q2 (Median)	0.77	0.20	4.13	0.10	0.61	11.99
Q3	1.64	0.38	8.88	0.74	0.71	95.09
Average	1.54	0.28	6.00	0.34	0.59	80.38
Std Dev	2.34	0.23	5.41	0.40	0.16	144.46
Maximum	25.86	0.99	28.36	1.00	0.94	743.48

Fig. 1 Statistical “violin” with included Box-whiskers plot. The violin-width indicates the frequency distribution of data points at that variable-value—indicated by the y-axis. Data points are plotted with horizontal “jitter” to prevent overlap. The box represents the median (Q2) and interquartile range (IQR) between the 25th (Q1) and 75th (Q3) percentiles. Variables are: (a) Average cross-shore Mangrove Width (AMW), (b) NDVI, and (c) Average Canopy Height (ACH), plotted for the two years.

Regarding the issue with MCPI values, in the original version, normalization was performed before the calculation. All three indices were scaled to 0-1 based on the maximum and minimum values and then multiplied together. However, the results obtained by multiplying the normalized three indicators together sometimes yielded values that were too close to zero. For display convenience, we multiplied the results by a scale factor of 1000. Therefore, the actual range of MCPI was (0, 1000).

For the revised version of MCPI, the three indices A, B, and NDVI have also been scaled to the range of 0-1 and then multiplied by the same scale factor of 1000, resulting in a final range of (0, 1000).

Detailed comments:

Ln 5: What are historical forces?

Response [6]:

Thank you for raising the question. We apologize for the lack of precision in our statement. Mangroves are affected by extreme weather events, offshore hurricanes, storm surges, and other disasters, which are exacerbated in the background of climate change. Additionally, they are also influenced by human activities. Historical forces originally refer to these influencing

factors.

We now have revised our statement to (lines 5-6):

“Mangroves protect coasts from extreme weather and erosion but may be experiencing destruction from climate-change and harvesting.”

Ln 8-10: Does not indicate the purpose of the MCPI, just its construction. When should this be used? What does it indicate? With this in mind, is there a rationale for these parameters?

Response [7]:

Thank you so much for your careful reading and your constructive comments.

Purpose and application: Our aim is to provide a unified evaluation standard and enhance the public's understanding of mangrove coastal protection capacity on a global scale. The objective is to offer a globally comprehensive and comparable assessment of mangrove coastal protection capacity, that can be used to compare protection capabilities between different regions against various coastal threats.

Indication: The Mangrove Coastal Protection Index (MCPI) signifies the **inherent** coastal protection capability of mangroves, rather than the specific protective effects from singular factors and for a particular storm event.

Rationale: In the revised version, we adopted a more theoretically grounded approach to constructing MCPI, as detailed in **Response [3]-[4]**, and revision-responses **R1** to **R4**, which we do not elaborate upon here.

The manuscript is revised accordingly as follows (lines 7-9):

“Here, we quantified and analysed a process-based measure of Mangrove Coastal Protection Index (MCPI) incorporating cross-shore width, canopy-height, and normalized-difference-vegetation-index (health-index)”

Ln 20: rising sea levels and frequent extreme events, amongst other aspects

Response [8]:

Thank you so much for your careful reading and your constructive comments.

The manuscript is revised accordingly as follows (lines 22-23):

“Global climate change has intensified coastal erosion, flooding, and storm surges, driven by rising sea levels, intensifying boundary currents, and more frequent extreme weather events, amongst other aspects^{1,2,3}.”

Ln 23: environmentally friendly environmentally sustainable. Could you also consider social aspects here too?

Response [9]:

Thank you so much for your careful reading and your constructive comments. According to suggestion, we have also taken social benefits into consideration. Incorporating mangrove in

hybrid coastal protection can greatly reduce global coastal protection costs (Temmerman et al., 2013; van Zelst et al., 2021). According to Menéndez et al. (2020), flood protection benefits provided by mangrove can exceed \$US 65 billion per year.

Therefore, social effects can be variously measured as economic benefits, corresponding to what we labelled "economic." Meanwhile, mangroves also provide social benefits by protecting coastal residents and property from storm events.

These, together with the natural environmental aspects of being "environmentally friendly and sustainable," underscore the significant role of mangroves in coastal protection.

The manuscript is revised accordingly as follows (lines 26-28):

“Considering these challenges, mangroves may offer economic, socially beneficial, environmentally friendly, and sustainable coastal protection, compared to engineered coastal defense structures^{8,9}.”

References:

- Menéndez, P., Losada, I. J., Torres-Ortega, S., Narayan, S., & Beck, M. W. (2020). The global flood protection benefits of mangroves. *Scientific reports*, 10(1), 1-11.
- Temmerman, S., Meire, P., Bouma, T. J., Herman, P. M., Ysebaert, T., & De Vriend, H. J. (2013). Ecosystem-based coastal defence in the face of global change. *Nature*, 504(7478), 79-83.
- van Zelst, V. T., Dijkstra, J. T., van Wesenbeeck, B. K., Eilander, D., Morris, E. P., Winsemius, H. C., ... & de Vries, M. B. (2021). Cutting the costs of coastal protection by integrating vegetation in flood defences. *Nature communications*, 12(1), 6533.

Ln 28: mangroves occupy a range of settings with estuarine environment being a large component, but not the only. There are many open coast mangroves in this analysis, and presumably some carbonate settings too

Response [10]:

Thank you for your careful reading, and your considerations are very meticulous.

We deleted “Thriving in muddy estuarine coastal areas”, and the manuscript is revised accordingly as follows (lines 32-34):

“Their intricate and expansive crown-root systems entrap sediments and dissipate turbulent kinetic-energy which also enhances sediment deposition^{14,15}”

Ln 35: ‘Numerous evaluations of mangrove coastal protection ’is a little contrary to statements in the abstract about being the first – there is clearly work going on

Response [11]:

Thank you so much for your careful reading and your constructive comments. We believe this is not contradictory. When we refer to the "first time", we are referring to the global-scale assessment of mangrove **inherent** coastal protection capacity. Some previous studies quantified the coastal protection role of mangroves on a global scale, such as Jia et al. (2023), who used data on population and area near mangroves to assess mangrove services. However, to our

knowledge, there has not yet been a comprehensive assessment of mangroves themselves on a global scale due to various challenges in quantitative evaluation.

But this does not mean that there are no local-scale studies. Without local assessments as a foundation, our global-scale assessment would not be feasible. We integrated the conclusions of existing numerical simulations and laboratory experiments to conduct this global assessment.

We correspondingly modified this expression: (lines 42-44):

“Multiple evaluations of mangrove coastal protection capacity, mostly through laboratory experiments and numerical simulations, indicate that key factors include: forest width, crown diameter, root density, tree density, forest vertical structure, species and ecological condition^{13,16,22,23}.”

Reference:

- Jia, M., Wang, Z., Luo, L., Zhang, R., & Zhang, H. (2023). Global status of mangrove forests in resisting cyclone and tsunami. *The Innovation Geoscience*, 1(2), 100024-1.

Ln 36: 'planting' density 'only relevant where afforestation and reforestation is occurring

Response [12]:

Thank you for your thoughtful comments and recommendations. We changed the 'planting density' to 'tree density' according to your suggestions.

The manuscript is revised accordingly as follows (lines 42-44):

“Multiple evaluations of mangrove coastal protection capacity, mostly through laboratory experiments and numerical simulations, indicate that key factors include: forest width, crown diameter, root density, tree density, forest vertical structure, species and ecological conditions^{13,17,22,23}.”

Ln 37: generally I do not think there is enough rationalization of the input component in the MCPI. Here is a missed opportunity to provide the reason for including mangrove width in the index. This needs expansion

Response [13]:

Thank you for your careful reading and your constructive comments.

The manuscript is revised accordingly as follows (lines 44-47):

“One critical factor is distance extending inland from the coastline, which provides inland protection against tsunamis and hurricanes^{14,24}. A wider mangrove belt acts as a buffer zone, absorbing and dissipating the energy of incoming waves and storm surges before they reach populated coastal areas¹⁷.”

Please also see revision-response **R2** where we describe a more process-based formulation for width that parametrizes wave attenuation.

Ln 40-41: I cannot follow this sentence. Perhaps needs rephrasing

Response [14]:

Thanks so much for your careful reading and your constructive comments.

The manuscript is revised accordingly as follows (lines 49-51):

“However, the variability and complexity of mangrove ecosystems has led to limited research on protection-capacity being scattered across regions of significant disparity. Consequently, existing interpretations of protection capacity are inconsistent or missing.”

Ln 43: I do not think scarce research is the reason for no studies of coastal protection on a global scale. I suspect that there is issues with data quality, and issues with the relevance of an analysis at this scale for decisions at the local scale. This is a location where the statement is too definitive and does not recognize the many reasons for a study of this type not occurring previously. I am not sure why the authors are pressing to make the point that this is the first.

General comments:

- The language is more definitive at times than it should be. For example, are the authors really that confident that this is the first time a global assessment of protection capacity has occurred, of that there is currently no research on how to measure coastal protection at a global scale. Even the first sentence “mangrove have protected coastal area’ is more definitive than observations would indicate.

Response [15]:

Thank you very much for your critique and guidance. We indeed made an overly absolute assertion regarding the "first-time" claim. When we refer to the "first time", we are referring to the global-scale assessment of mangrove **inherent** coastal protection capacity. Some literature has quantified the coastal protection role of mangroves on a global scale, such as Jia et al. (2023), who used data on population and area near mangroves to assess mangrove services. However, to our knowledge, there has not yet been a comprehensive assessment of mangroves themselves on a global scale due to various challenges in quantitative evaluation.

As you pointed out, the lack of global-scale studies on mangrove coastal protection capacity is attributable to several reasons, including the complexity of mechanisms, poor comparability of data, and substantial regional variations. Another critical issue is the challenge of estimating mangrove offshore width due to the complexity of mangrove patches and coastline morphology. This is why studies like that of Jia et al. (2023) only consider protection provided by mangroves with widths exceeding 100 or 1500 meters (determined by creating the maximum circle within mangrove patches and filtering based on circle diameter).

The basis for integrating multiple factors into a global quantitative assessment lies in designing **sampling lines perpendicular to the coastline** using the Google Earth Engine (GEE) platform. Through this new method, we are able to obtain specific mangrove widths globally and extract relevant indicators.

As long as there is a need for research, we cannot afford to remain stagnant simply due to

uncertainty about the data. In the global-scale research, it is difficult to guarantee that data is completely accurate. For remote sensing datasets, we prioritize accuracy verification, considering errors within a certain range as acceptable. In order to ensure data accuracy to the greatest extent, we selected reputable data sources, conducted a series of data preprocessing steps, and verified key areas using high-resolution imagery and on-site inspections before the study.

The three main datasets used were:

1. Width:

The Global Mangrove Watch Version 3.0 (GMW): Published in 2022, with 113 citations, it represents the most comprehensive record of global mangrove change achieved to date.

2. Height:

(i) Global Forest Canopy Height (2019) from Potapov et al. (2019)

(ii) Global Forest Canopy Height (2005) from Simard et al. (2011).

Validation was conducted using field measurement data and GEDI validation data, and the RMSE was 6.6m and 6.1m respectively. Considering that both datasets have undergone accuracy validation, with RMSE within acceptable ranges, and they also have high citation counts ((i) with 1200 citations; (ii) with 623 citations), we believe their errors are acceptable.

3. NDVI

NDVI was extracted from Landsat Collection 2 Tier 1 Surface Reflectance data provided by the United States Geological Survey (USGS), with systematic atmospheric correction and cloud masking processing. This ensured the accuracy and comparability of the time-series data.

The manuscript is revised accordingly as follows (lines 52-54):

“Therefore, there is a lack of consistent and comprehensive studies of changes in coastal protection ability of mangrove forests applicable at global scales”

Reference:

- Bunting, P., Rosenqvist, A., Hilarides, L., Lucas, R. M., Thomas, N., Tadono, T., ... & Rebelo, L. M. (2022). Global mangrove extent change 1996-2020: Global mangrove watch version 3.0. *Remote Sensing*, 14(15), 3657.
- Jia, M., Wang, Z., Luo, L., Zhang, R., & Zhang, H. (2023). Global status of mangrove forests in resisting cyclone and tsunamis. *The Innovation Geoscience*, 1(2), 100024-1.
- Potapov, P., Li, X., Hernandez-Serna, A., Tyukavina, A., Hansen, M. C., Kommareddy, A., ... & Hofton, M. (2021). Mapping global forest canopy height through integration of GEDI and Landsat data. *Remote Sensing of Environment*, 253, 112165.
- Simard, M., Pinto, N., Fisher, J. B., & Baccini, A. (2011). Mapping forest canopy height globally with spaceborne lidar. *Journal of Geophysical Research: Biogeosciences*, 116(G4).

Ln 59: AMW did not exhibit a clear pattern – but what was expected? Should we expected a clear pattern when the development of intertidal surfaces suitable for mangrove habitations is dependent upon many coastal processes, and is modified by human activities

Response [16]:

Thank you so much for your careful reading and your constructive comments. We apologize for our inadequate expression. We did not have an anticipated pattern; rather, this description merely reflects the results of width distribution. In contrast to the pronounced latitudinal distribution pattern of height, the distribution of width appears more irregular. As you mentioned, this irregularity is influenced by numerous factors driven by human activities.

The manuscript is revised accordingly as follows (lines 96-97):

“Globally, the width-based wave Attenuation factor (A) did not exhibit widespread spatial aggregation (Fig. 2a).”

Ln 59-60: AMW was influenced by factors including natural conditions, national policies and land use management – could these all be reasons for why it should be excluded from the index? There is no strong rationalization provided for width, height and greenness indicating coastal protection

Response [17]:

Thank you so much for your careful reading and your constructive comments. We apologize for our unclear statement. AMW is not excluded from the index and is an important indicator. It is “factors including natural conditions, national policies, and land use management” that are the reason why AMW did not exhibit widespread spatial aggregation. In the revised version, we used the wave attenuation factor (A) to replace AMW and a biomass-based frontal barrier factor (B) to replace ACH, as detailed in **R2-R3**. The reasons for incorporating NDVI as an indicator are detailed in **Response [4]**.

The manuscript is revised accordingly as follows (lines 96-97):

“Globally, the width-based wave Attenuation factor (A) did not exhibit widespread spatial aggregation (Fig. 2a).”

Ln 61: MEOW – too many acronyms – I was getting lost

Response [18]:

Thank you so much for your careful reading and your constructive comments. We apologize for our oversight. The MEOW means Marine Ecoregions of the World, provided by World Wildlife Fund (WWF) (Spalding et al., 2007).

We removed the sentence regarding MEOW. An explanation and citation are provided later at its first occurrence (see in lines 332-333).

“The sampling line was determined from actual growth of mangroves in each region, using ecoregions from Marine Ecoregions of the World (MEOW)⁴⁴ (shown in Supplementary Fig. 13).”

Supplementary Fig. 13 Ecoregions within different realms relative to the study from Marine Ecoregions of the World (MEOW).

Reference:

- 44. Spalding, Mark D., et al. Marine ecoregions of the world: a bioregionalization of coastal and shelf areas. *BioScience* 57.7, 573-583(2007).

Ln 61: HUGE mangrove widths – perhaps you could say something about why there are large mangrove widths in this region

Response [19]:

Thank you for your suggestion.

The manuscript is revised accordingly as follows (see in 97 -99):

“Unscaled-AMW however showed an extensive range, with Bangladesh recording the highest value of 25.86 km due to its unique geographical and environmental conditions, while the global average was only 1.54 km.”

Ln 70-71 – first instance of reporting averages in cells and ecoregions – I am not sure this distinction is clear enough Is the average in an ecoregion the average of the average of grid cells? Is this appropriate?

Ln 77: need to explain average in grid cells vs ecoregions

Response [20]:

Thanks so much for your careful reading and your constructive comments. Indeed, our previous description was inadequate. In the updated version, we removed the analysis based on **ecoregions** and only used **grids** and **Clusters** as units for data aggregation. This description should be clearer, more meaningful, and more understandable for readers.

The manuscript is revised accordingly as follows (lines 106-107):

“The average global ACH was 6 m, with New Guinea recording the highest value of 28.36 m.”

Ln 72: exceeding an ACH of 20 m

Response [21]:

Thank you so much for your careful reading and your constructive comments.

The manuscript is revised accordingly as follows (lines 107-108):

“About one-fifth of mangroves exceeded an ACH of 10 meters and had B values recorded as 1.0”

Ln 85-86 – repetitive. Methods detailed in methods

Response [22]:

Thank you so much for your careful reading.

The manuscript is revised accordingly as follows (lines 114-115):

“The spatial distribution of the MCPI (scaled by 1000) was computed using formulas detailed in Methods (Fig. 2d).”

Ln 97: I think the authors should provide more confidence about the independence of height and NDVI – I suspect there is some covariance that should be explored and reported

Response [23]:

Thank you so much for your careful reading and your constructive comments. We calculated correlation matrices for three indices across two periods. The results showed that correlation coefficients between height and NDVI were 0.56 in 2007 and 0.71 in 2019, indicating a moderate correlation. These two variables are not completely independent but still provide different aspects of information, hence we retained them. For example, as shown below in the scatterplot of NDVI and height, the relationship between the variables **changes** in time as one varies **differently** to the other. Hence, we do report the different variations causing these changed correlations.

Ln 119: eastern coastline is not pronounced – this is presumably because of averaging which masks the highly variable nature along this coastline which has a very crenulated shoreline. I think more detail is needed here rather than reporting an average

Response [24]:

Thank you so much for your careful reading and your constructive comments. We apologize for the inadequate expression.

Indeed, averaging may mask the highly variable nature along this coastline. However, the smallest processing unit for our data is limited to $1^{\circ} \times 1^{\circ}$ grid cells. Using smaller scale would make it challenging to extract mangrove comprehensively from sampling lines perpendicular to the coastline. Moreover, from the new Fig. 3a (original Fig. 2a), it is evident that using grids as the unit already reveals regional differences in width changes along the eastern coast of Brazil, with many areas showing no sustained changes (denoted by grey diagonal lines), while

some areas exhibit sustained growth (green) or sustained decline (red).

Here, by 'not pronounced,' we mean that there is a lack of **ongoing** trend of mangrove width expansion over the 24 years (grey diagonal lines).

The manuscript is revised accordingly as follows (lines 161-163):

“For Brazil, despite some regions showing extremely high annual mangrove width change rates (such as the Amazon Basin), many mangroves along its eastern coastline did not exhibit a sustained trend³⁴.”

Ln 129: I know the methods is where the detail is, but a bit of information about how the MCPI was compiled would be helpful here. E.g. the NDVI, ACH and AMW were multiplied with a scaling factor to indicate MCPI

Response [25]:

Thank you so much for your careful reading and your comments.

The manuscript is revised accordingly as follows (lines 199-200):

“Global MCPI (using A, B, and NDVI; scaled by 1000) and change rates were calculated for 2007 and 2019 (Fig. 4).”

Ln 130: This sentence is confusing was MDPU calculated fore each year, and the change only for 2007 – 2019?

Response [26]:

Thank you so much for your careful reading and your constructive comments. We apologize for the confusion caused by our unclear statement. We only calculated the MCPI for the years 2007 and 2019 because height data were available only for the years 2005 and 2019. We approximated the height for 2005 to represent the height in 2007 because the closest mangrove distribution data was for 2007 (Global Mangrove Watch 3.0 (GMW) dataset only spans the years 1996, 2007-2010, and 2015-2020).

The manuscript is revised accordingly as follows (lines 199-200):

“Global MCPI (using A, B, and NDVI; scaled by 1000) and change rates were calculated for 2007 and 2019 (Fig. 4).”

Ln 134: This sentence demonstrates the problem with indexes such as the MCPI. They compile all the input factors into a single variable which means that you cannot pull apart the primary component influencing the Index at a given location. This is the first mention of Australia – but it was not highlight in the context of NDVI, ACH and AMW in previous sections – what was the primary cause here?

Response [27]:

Thank you so much for your careful reading and your constructive comments. For a more in-

depth analysis of the primary drivers of MCPI changes, we incorporated a novel Cluster Exchange Network Analysis (CENA) technique described in the revision-response **R5**. This approach allowed us to conduct a typological analysis of associated-factors influencing the Index at a given location. Detailed analyses can be found in the ‘**Cluster Exchange Results**’ section of the Results and Discussion (lines 226-289) and the ‘**Cluster Exchange Network Analysis (CENA)**’ section of the Methods (lines 434-447).

Indeed, Australia was a huge surprise but was clearly identified as changing significantly into the low performing Cluster 8 in 2019. The resolution of the surprise was the commonality across most large-scale changes to Cluster 8 in 2019, namely the presence of boundary current regions—which may also correlate to cyclones in warm currents that transport substantial volumes of warm water. We undertook spot checks of 3 locations to confirm that the height changes were due to tree thinning in areas transitioning into Cluster 8.

Ln 143: What do numbers in brackets mean?

Response [28]:

Thank you so much for your careful reading and your constructive comments. We apologize for the oversight; these were results from some analyses that were not thoroughly explained. In the revised version, we removed this data and incorporated cluster analysis to attribute the changes in MCPI.

The manuscript is revised accordingly as follows (line 211):

“Factor wise, MCPI varied mainly from ACH, followed by AMW, and minor NDVI contributions.”

Ln 165-166: YES – this is because it is what the index is based on. I would prefer greater connection to processes that cause these changes being described here and linking this to the relevant literature. e.g., lower rates of clearance, high rainfall and warmer temperatures. Interestingly, these factors also correlate with the occurrence of severe cyclones, which I suspect could be discussed further

Response [29]:

Thank you so much for your careful reading and your constructive comments. In the revised manuscript, we further discuss the reasons behind the high values of MCPI.

The manuscript is revised accordingly as follows (lines 121-125):

“Thus, high MCPI mangroves are primarily distributed within latitudes of 5°N and 5°S, resulting from the combined influence of natural conditions and human factors. Natural factors such as higher rainfall, warmer temperatures, and lower tropical-cyclone frequency provide better ecological foundations for mangroves^{2,31}. Additionally, the relatively minor impact of human activities in these regions and favorable tidal hydrodynamic conditions help maintain their wider coastal width₁₂.”

Reference:

- 2. Osland, M. J. et al. Mangrove forests in a rapidly changing world: Global change impacts and conservation opportunities along the Gulf of Mexico coast. *Estuarine, Coastal and Shelf Science* 214, 120–140 (2018).
- 12. Alongi, D. M. Present state and future of the world's mangrove forests. *Envir. Conserv.* 29, 331–349 (2002).
- 31. Simard, M. et al. Mangrove canopy height globally related to precipitation, temperature and cyclone frequency. *Nature Geosci* 12, 40–45 (2019).

Furthermore, we conducted a more detailed discussion on the reasons for the MCPI changes, particularly focusing on the influence of oceanic boundary currents. Please refer to the Cluster Exchange Results section (lines 265-289) for details:

“The contributions to Cluster 8 from other Clusters is depicted in the bootstrap Cluster probability correlation matrix plot in Supplementary Fig. 8. Most Cluster memberships within 2019 were negatively correlated (to those in 2007), particularly for Cluster 8. This reinforces the suggestion of destruction as the cause, as shown spatially in Supplementary Fig. 9, which also suggests very remarkably that greatest ACH changes are associated with all the major tropical/sub-tropical oceanic boundary current regions including: (1) East Australia Current; (2) Agulhas Current; (3) Equatorial/Benguela Current; (4) Start of Gulf Stream Current; (5) California Current; (6) Kuroshio Current; and (7) Brazil Current. Note both warm and cold currents appear to be affected suggesting that destruction in these regions may arise from both a static sea-level rise and a dynamic component due to acceleration (hence kinetic-energy increase) in the boundary currents.

Boundary current regions, such as the East Australia Current, are hotspots of global warming, sea-level rise, storms, and marine heatwaves³⁷. It is reasonable to infer that mangroves in global hotspot regions are declining in protection capacity as they are unable to sustain increasing climate change forces such as more frequent and intense storms, and rising sea-levels. Geostrophically, it is well-known that any increase in boundary current velocities is linearly related to dynamic sea-level height increases (above sea-level rise) required to drive these currents. For example, intensification of the East Australian Current (EAC) has been noted by Kelly et al.³⁷ who describe a rapid extension of the EAC since the 1990s such that seasonal cycles off mid-eastern Tasmania in 2004-2005 were similar to those much further north previously. Dynamically boundary currents are driven by cross-shore pressure gradients, hence similar mechanisms will operate in other regions; further fueled by high kinetic energies (through swift currents and eddies) associated with these oceanic systems, and intensified storm surges. Our estimated start of mangrove destruction in 1990 also aligns with the estimate start of the EAC extension. Climate change-induced sea level anomalies and extreme high temperatures may also impact the health of mangroves, such as those located in the Gulf of Carpentaria, Australia³⁸. Additionally, harvesting of mangroves may reduce tree density and aboveground biomass, thereby accelerating local declines in mangrove protection capacity³⁹. Our modelling does not distinguish between these destruction modes, but diagnostically suggests that it is focused along boundary-current regions.”

Supplementary Fig. 9 | Difference of Height factor between 2007 and 2019. Note large Height reductions in numbered boundary current regions of: (1) East Australia Current; (2) Agulhas Current; (3) Equatorial/Benguela Current; (4) Start of Gulf Stream Current; (5) California Current; (6) Kuroshio Current; and (7) Brazil Current. “Warm” currents (low to high latitude) are in **dark-red**, and “cold” ones (high to low latitude) in **dark-blue**.

Ln 209: What is co-contravariations???

Response [30]:

Thank you so much for your careful reading and your constructive comments. We apologize for the confusion caused by our unclear statement.

The manuscript is revised accordingly as follows (lines 300-301):

“Still, different and in many cases dramatic, change patterns between MCPI and traditionally-used mangrove area indicate that MCPI captures new insights into mangrove protection attributes^{40,41}.”

Ln 211-212: This sentence is written as if a manager may be able to influence these variables and increase coastal protection. The reality is that height is largely climatically defined and with a temporal aspect (i.e. leave trees in place until they reach maturity), and width is influenced by human factors that can be managed, but also be geomorphological factors that are difficult to modify

Response [31]:

Thank you so much for your careful reading and your constructive comments. We are sorry for our oversight in the original statement. Indeed, it's challenging for people to directly intervene in the actual height of mangroves. Managers can only protect mangroves by implementing better conservation measures, such as establishing protected areas, limiting harvesting, to maintain tree density and preserve ecosystem health.

We revised our statement according to your suggestions as follows (lines 311-312):

“Critical growth parameters like tree density and canopy coverage must be included in monitoring, reforestation, and conservation plans.”

Ln 212: What is the difference between forest width (AMW?) and growth (is this NDVI and ACH?)

Response [32]:

Thank you so much for your careful reading and your constructive comments. Please allow us to provide some explanation.

The width represents the distribution range of mangroves, indicating areas where mangroves are present. Considering that the resolution of mangrove remote sensing dataset (GMW 3.0) is approximately 30 meters, we can only determine the presence of mangroves within a given area, but we cannot ascertain the growth density of mangroves within each pixel. Therefore, we use other parameters to reflect the growth condition of mangroves, approximating factors like tree density, biomass and canopy condition through height and NDVI.

The manuscript is revised accordingly as follows (lines 311-312):

“Critical growth parameters like tree density and canopy coverage must be included in monitoring, reforestation, and conservation plans.”

Ln 214-215: could the authors speculate as to how much change is because of species increasing in dominance that have lower NDVI and height compared to other species. E.g. Avicennia generally is lower in stature and NDVI than Rhizophora within a system.

Response [33]:

Thank you so much for your careful reading and your constructive comments.

For the case of the Amazon Basin, we found that the decrease in average height (hence MCPI) was due to the planting/establishment of new mangroves. The extensive new mangrove areas are shorter, thus lowering the overall canopy height of the region.

The manuscript is revised accordingly as follows (lines 312-316):

“For example, if we accept that the 124% area increase in the Amazon Basin, countered by a 95% MCPI decline, is from newly established but shorter mangroves, this will reduce overall ACH (and MCPI). This implies that extensive new mangroves, whether from inland migration or planting, will not have a positive short-term impact on overall coastal protection capacity of mangroves.”

Ln 222: Natural Earth needs a reference

Response [34]:

Thank you so much for your careful reading and your comments. We removed the mention of 'Natural Earth' and explained it in the data availability section.

The manuscript is revised accordingly as follows (lines 327-330):

“First, due to the complex topological relationship between mangroves and coastlines⁴³, we manually adjusted the coastlines to approximate the straight line of outer wrapping of mangroves based on the mangrove distribution extent from GMW. This ensures that our

sampling lines cover all mangroves and minimizes under-sampling and over-sampling.”

Ln 222-223: I do not understand the straight line outer wrapping sentence – I think more detail is needed here

Response [35]:

Thank you so much for your careful reading and your constructive comments. We apologize for the confusion caused by our unclear statement. More details have been added.

The manuscript is revised accordingly as follows (lines 327-330):

“First, due to the complex topological relationship between mangroves and coastlines⁴³, we manually adjusted the coastlines to approximate the straight line of outer wrapping of mangroves based on the mangrove distribution extent from GMW. This ensures that our sampling lines cover all mangroves and minimizes under-sampling and over-sampling.”

Ln 225-226: Was this extraction pertaining to certain years?

Response [36]:

Thank you so much for your careful reading and your constructive comments. We apologize for the confusion caused by our unclear statement. All extractions were based on the corresponding year's mangrove extent and dataset.

The manuscript is revised accordingly as follows (lines 333-335):

“AMW, ACH, and NDVI, for the respective years were extracted from sampling lines after clipping the mangrove data using refined $1^{\circ} \times 1^{\circ}$ grids.”

Ln 257-258: Reference to AMW, ACH and NDVI should come earlier and they need more rationalization of why they indicate coastal protection capacity. For example, how does greenness (NDVCI) modify coastal protection capacity (I do not think it does – rather it is correlated with species that modify capacity). How do trees of > 10 m improve coastal protection capacity compared to a tree of 10-15 m (I do not think it does, particularly as taller trees can become more unstable, and there may be less wave attenuating biomass within the wave zone).

Response [37]:

Thank you so much for your careful reading and your constructive comments. In the revised version, we adopted a more theoretically grounded approach to constructing MCPI, as detailed in revision-responses **R1** to **R4**. The rationalization for NDVI is explained in **Response [4]** and height in **Response [3]**.

Ln 262: S: I see scaling factors often and the only purpose they serve is to make the number easier to report (i.e. changes a small number to a large number). I think it is more critical to

normalize the values (i.e. a score between 0-100) to guide comparison, and to shore the distributions of the normalized scores.

Response [38]:

Thank you so much for your careful reading and your constructive comments. The results obtained by multiplying the normalized three indicators together sometimes yielded values that were too close to zero (very low protection capacity). So indeed, we used a scale factor of 1000 for display convenience. Therefore, the actual range of MCPI was (0, 1000) so that we could emphasize the very low values. We added explanations of the scaling factor at several points where MCPI is mentioned to minimize confusion for readers.

For example (lines 114-115, 199-200):

“The spatial distribution of the MCPI (scaled by 1000) was computed using formulas detailed in Methods (Fig. 2d).”

“Global MCPI (using A, B, and NDVI; scaled by 1000) and change rates were calculated for 2007 and 2019 (Fig. 4).”

Ln 265: First time a year is mentioned in the methods; I think this is needed sooner

Response [39]:

Thank you so much for your careful reading and your constructive comments. We apologize for our oversight. We added explanations of the data years earlier in our methods section. The manuscript is revised accordingly as follows (lines 333-335):

“AMW, ACH, and NDVI **for the respective years** were extracted from sampling lines after clipping the mangrove data using refined $1^{\circ} \times 1^{\circ}$ grids.”

Ln 267: Slope of the linear regression could be confused with intertidal slope

Response [40]:

Thank you so much for your careful reading and your constructive comments.

We understand your concern regarding the potential confusion between the slope of the linear regression and intertidal slope. In our manuscript, we do not specifically refer to intertidal slope, and the linear regression slope is solely used to indicate the rate of change of mangrove width over time.

Ln 273: Global mangrove height data from Simard from SRTM survey undertaken in 2000. Potapv et al. was with GEDI survey in 2019. May need to reflect upon whether it is appropriate to use 2007 timestep when the data is pertaining to 2000.

Response [41]:

Thank you so much for your careful reading and your constructive comments.

Please let us make some explanation. The global mangrove height data from Simard et al. (2011) was derived from the Geoscience Laser Altimeter System (GLAS) aboard ICESat (Ice, Cloud, and land Elevation Satellite) in **2005**, not from the SRTM survey conducted in 2000. Given the available timeframe of Global Mangrove Watch (GMW) 3.0, we selected the nearest year, which was 2007, as the timestep.

Reference:

- Bunting, P., Rosenqvist, A., Hilarides, L., Lucas, R. M., Thomas, N., Tadono, T., ... & Rebelo, L. M. (2022). Global mangrove extent change 1996-2020: Global mangrove watch version 3.0. *Remote Sensing*, 14(15), 3657.
- Simard, M., Pinto, N., Fisher, J. B., & Baccini, A. (2011). Mapping forest canopy height globally with spaceborne lidar. *Journal of Geophysical Research: Biogeosciences*, 116(G4).

General comments:

- The grouping amongst ecoregions is very broad and allows for differences amongst species and some latitudinal trends to be differentiated. However, I would have liked to see more consideration of latitudinal patterns as others have done previously. This could be achieved by binning values across latitudes and displaying in a graph. I think all of the global mangrove height and biomass papers do this approach and I find it useful.

Response [42]:

Thank you so much for your careful reading and your constructive comments. We apologize for our oversight and have made the corresponding revisions according to your suggestions. In the revised version, we added a latitude statistical graph to the right, as shown in Fig.2. Indeed, this is very useful!

Fig. 3 Spatial distribution of global mangrove coastal protection capacity for 2019. (a) width-based wave attenuation (A), (b) biomass-based frontal barrier (B), (c) average Normalized Difference Vegetation Index (NDVI), (d) Mangrove Coastal Protection Index (MCPI) and (e)-(h) latitude statistics of three indices and MCPI.

- Fig 2 – remove the smoothing line between points in c-g. These are discrete values that should not be connected, and not with a smoothed line.

Response [43]:

Thank you so much for your careful reading and your constructive comments. We apologize for our oversight and have made the corresponding revisions according to your suggestions (see Fig.3):

Fig. 4 Results of trend regression analysis for average mangrove width (AMW) based on data from 1996 to 2020. (a) and (b) represent the spatial patterns of annual mangrove width change rates extracted using $1^{\circ} \times 1^{\circ}$ grids and national boundaries, respectively. The unit is meters per year. (c) depicts global average trend of mangrove width. (d), (e), (f), and (g) represent width change trends in the Americas, Africa, Asia, and Oceania, respectively.

- Fig 5: I can appreciate the point the authors are trying to make, but I do not think this figure is showing it well enough. The third arrow from the top indicates a lot of damage despite a remarkable width of mangroves

Response [44]:

Thank you so much for your careful reading and your constructive comments. The figure has been removed in the updated manuscript and replaced with content from revision-response **R6**.

- Fig 7 bottom graph needs a y axis label (height (m)). This figure is not reference in the document anywhere. It may be the case for other figures

Response [45]:

Thank you so much for your careful reading and your constructive comments. We apologize for our oversight and have made the corresponding revision (see Fig.3), and have added a reference to this figure in the Methods section (line 389):

“Illustrations of the factor extraction are presented in Supplementary Fig. 10 and

Supplementary Fig. 11.”

Supplementary Fig. 11 | Taking mangroves near the Gulf of Guinea as an example, sampling lines were generated perpendicular to the coastline, and average canopy height was extracted for each sampling line.

Minor revisions were made to the manuscript to address all comments provided by the 3 reviewers. We discussed additional possibilities for height reduction, including the influence of cyclones, ocean currents, and sea level rise, and made detailed modifications to some statements according to the reviewers' suggestions. Additionally, we provided spot checks from typical regions to support suggestions of destruction and thinning for reductions of “average height”. Point-by-point responses are provided below.

Responses to Reviewers

We are grateful to all Reviewers for the helpful and constructive feedback. We have addressed every comment carefully and revised the manuscript accordingly. Below please find our point-by-point responses to each review comment. *The review comments are highlighted in blue*, while our responses are in **black**. Track-changes completed in the revised manuscript and Supplementary material are shown in purple.

Responses to Reviewer #1:

The authors have been thorough in their responses to the reviewers' comments. I believe they have adequately responded to the comments, and therefore, the manuscript is much improved.

Response [1]:

Thank you very much for your positive feedback and for recognizing the improvements in our manuscript. We greatly appreciate your thorough and meticulous review, as well as your constructive comments, which have significantly enhanced the quality of our work.

In this revision, we discussed additional possibilities for height reduction, including the influence of cyclones, ocean currents, and sea level rise, and made detailed modifications to some statements according to the reviewers' suggestions. Additionally, we provided spot checks from typical regions to support suggestions of destruction and thinning for reductions of “average height”. Please refer to the point-by-point responses below for the specific revisions we made to the manuscript.

*However, I am not entirely convinced by the extended height change analysis. While the results and interpretation are plausible, I am not convinced that the two canopy height products are **directly comparable** and that such a **trend can be reliably measured** from these datasets. The two datasets were created from two different methodologies using entirely different data sources.*

Response [2]:

Thank you so much for your careful reading and your constructive comments.

We know that these two datasets used different data resolutions but both carried out careful and extensive calibration, modelling, and validations with defined statistical uncertainties. For our

research purposes, **we cannot avoid comparing the heights between the two years**, as we want to inter-compare changes with area and NDVI, all of which are part of MCPI.

We therefore carefully reviewed and conducted our own analyses of potential issues between the datasets. The statistical analysis indicates that the differences between the two datasets are not merely due to algorithm-induced data shifts. If there were just linear biases between datasets (intercept-offset or a scaling factor), we would expect the scatterplot to be more linear than observed. Indeed, width and NDVI are far more linearly related between the years: width: ($adj-R^2 = 0.997$, $slope = 0.985$); and NDVI: ($adj-R^2 = 0.988$, $slope = 1.023$). What we observe with height is a radical *deformation* of the *statistical distribution* (as seen in the violin plots) which suggests complex dynamic processes at play (with some exceptions) that are highly dependent on height itself (Response Fig. 1). Therefore, we cannot apply a traditional linear bias correction between the two datasets, which may arise from differences in methodology.

Response Fig. 1 | Scatterplot of 2019 height against 2005 height. Grey dashed-line is 1:1 line, hence most 2019 heights are shorter; red dashed line is the fitted line for linear regression with the intercept term set to 0

Regarding the possible data-comparability issue, we find it challenging to distinguish between **possible data errors** and **actual changes** in two canopy height datasets over a 14-year time span. Overall, two plausible mechanisms could confound explanations for changes of “average height”:

1. Possible Data Issues

We went back to the two datasets and conducted various checks. Both products underwent rigorous quality control and error analysis, utilizing LiDAR data and a regression-tree approach to estimate height (Potapov et al., 2021; Simard et al., 2011). We conducted spot checks on the products and reviewed their uncertainty analysis reports.

Both studies analyzed model uncertainty for different forest types, and the main results are as follow:

- (i) The error in the Global Forest Canopy Height (2005) is primarily within taller forests,

with the statement: "Error differs across forest types and increases as a function of canopy height; error increased in closed broadleaved forests such as the Amazon, underscoring the challenges in mapping tall (>40 m) canopies. (Simard et al., 2011)."

(ii) For Global Forest Canopy Height (2019), accuracy assessments indicate that the product may underestimate forest height, particularly for short forests (< 3 m) and tall forests (> 30 m) (Potapov et al., 2021).

Based on our results and Simard et al. (2019), most of the world's mangrove forests are above 5 m and below 30 m height (~70%), which are not the forest types with the largest errors (Response Fig.2). This suggests that the height data for most mangroves is acceptable.

Response Fig. 2 | Global distribution of maximum mangrove canopy height extent (quoted from Simard et al., 2019)

The errors that may result from different data sources and methodologies are as follows:

Both the 2005 and 2019 height-data products used sparse Lidar-based maximum height observations (RH100 for 2005 and RH95 for 2019) with respective footprints of 65 m and 25 m together with interpolation models based upon remote sensing data (coarse for 2005 and fine for 2019). These analyses provided a global maximum-height 1-km-pixel 2005 product and a much-finer 30-m-pixel product for 2019. In addition, heights under 3 m were set to zero for the 2019 data, and the 2005 data did not contain maximum heights less than 6 m. Therefore, smearing of maximum heights across much larger grids in 2005 may possibly explain why heights are higher compared to the fine-scale 2019 product. Therefore, for **mangroves with lower maximum heights**, there may be some comparability issues with the data.

Our original method calculated average-height across transects, which could be affected by various factors such as tree density. Therefore, to verify the accuracy of the original data, we compared maximums for only mangrove pixels on each transect for both years using 30 m resolutions (2005 dataset was resampled to 30 m, using Nearest Neighbor Resampling which will not change the pixel value) which were then aggregated for each $1^\circ \times 1^\circ$ grid cell. This was also to investigate which height ranges in the original datasets are less reliable. Ratios of the two maximum heights from grids are comparable as shown in Response Fig. 3 for **maximum-heights in 2019 above about 9 m but less than 36 m**. Below 9 m, divergence increases. At higher heights, the results are more comparable.

This aligns with our conclusions derived from the dataset's methodologies above. Based

on the global distribution of mangrove heights provided by Simard et al. (2019) (Response Fig. 2), most mangroves (~60%) are within the comparable range. Therefore, we believe the two datasets are comparable across most mangrove areas.

Response Fig. 3 | Ratio of 2005 to 2019 maximum heights as a function of the 2019 maximum height.

Using ICESat-2 data to reproduce the 2019 canopy height product, and comparing it with Simard et al. (2011)'s product (using ICESat-1), might be optimal (as done by Sun et al., 2020), but this is beyond the scope of this study. In future, we may consider producing more comparable mangrove height products for other years. For now, we include this as a worthwhile future research recommendation in the main text.

We have included a number of discussions in the manuscript addressing potential issues with the data:

Lines 76-79:

“Dramatic changes in cluster allegiances and spatial patterns are described along with a discussion of height data-related issues, process dynamics, policy implications, unforeseen implications of the current state, and prognosis for global future coastal destruction of, and protection from, mangroves.”

Lines 189-193:

“The calculated ACH changes can be attributed to actual height reduction and possible changes in tree density due to thinning effects (more patches of lower height noted above). It should be noted that these changes could be influenced by possible data-related issues from transect-patchiness to some extent. Whilst resolution of confounding interactions in some areas may await more accurate future data, the 2019 height notes that many deep-water coastal areas are already of low height (< 10 m) or biomass (Fig. 2b).”

Lines 236-238:

“Note that this analysis does not discriminate between real or data related changes. Instead, it tells when and where those changes are occurring so that we may more reliably interpret and draw conclusions from the results.”

Lines 312-315:

“Our modelling does not distinguish between these destruction modes and changes may in some cases be confounded by data-related issues. All discussions here are potential possibilities, intended to provide references, modelling, analytical techniques, caveats for mangrove protection policies, and suggestions on further research of data-related issues.”

Lines 323-325:

“While analyses of trends and change may be confounded by data issues, the finer-resolution 2019 data suggests that very substantive parts of the global coastline next to deep water are poorly protected by mangroves which will continue to decline in protection ability as destructive drivers increase.”

2. Destruction

Based on the conclusions from the first section above, data issues are only for a small portion, so the changes in average height across the large 1° grids are mostly likely due to actual changes. Destruction (by whatever means: human and/or environmental) along deep-ocean ocean boundary currents appear to have dramatically reduced average-height. Of greater worry is that average-height is a surrogate for biomass that varies as $\sim H^{2.55}$; hence biomass will be decaying at incredible rates between squared-to-cubed of height decay. Therefore, we conducted a height trend analysis, as detailed in the “Height Change Trend” section in the Supplementary Material.

However, we understand your concerns. In this revision, we have removed the height modeling from the main text as it is not the focus of this research. In this round of revisions, we have expanded discussions on the height changes and incorporated spot checks for destruction and thinning based on reviewers’ suggestions.

Modification of statements regarding height changes: (lines 267-272)

“The reductions in ACH can be attributed to two potential mechanisms: a genuine reduction in height, which may result from recovery following mortality events (caused by factors such as human influence, cyclones, drought conditions and hydrological limitations)^{37,38,39,40}; and a reduction in tree density measured differently in 2005 and 2019 because ACH is an average measure based on data resolutions and transects perpendicular to the coastline (Supplementary Fig. 16 and 17).”

Spot checks:

Here are two typical regions we examined (amongst a number): the southwest of Myanmar and the southwest corner of Florida, USA. From the true-color satellite images, it is evident that the green mangrove areas have decreased over time, being replaced by bare soil or flooding. Additionally, the mangroves have become more fragmented. This indicates that although the Global Mangrove Watch (GMW) data in Fig (a) suggest these areas still have mangroves, the

ecological condition may be poor or the mangroves have already been damaged due to storm surges, poor drainage and other factors (Win et al., 2020; Aung et al., 2011; Lagomasino et al., 2021; Radabaugh et al., 2020). The damaged mangroves have reduced in height, which lowers the tree density along the sampling lines, leading to a decrease in the extracted average mangrove height along the transects.

Supplementary Fig. 16 | Spot check of mangrove thinning for southwest Myanmar. (a) shows the distribution of mangroves in 2019 from GMW; (b), (c), and (d) are satellite images from December 1984, December 2005, and December 2019, respectively, obtained from *Google Earth Pro*. Over time, the mangrove areas have become increasingly fragmented, possibly due to sea level rise and flooding. Typical areas of change are highlighted by yellow ovals.

Supplementary Fig. 17 | Spot check for mangrove destruction for the southwestern corner of **Florida, USA**. (a) shows the distribution of mangroves in 2019; (b), (c), and (d) are satellite images from December 2004, December 2017, and November 2019, respectively, obtained from *Google Earth Pro*. The white ovals highlight areas where mangrove loss occurred following Hurricane Irma's strike in September 2017 and had not fully recovered by 2019. The yellow oval areas are an enlargement of the rightmost white oval area to facilitate viewing.

Reference:

- Potapov, P., Li, X., Hernandez-Serna, A., Tyukavina, A., Hansen, M. C., Kommareddy, A., ... & Hofton, M. (2021). Mapping global forest canopy height through integration of GEDI and Landsat data. *Remote Sensing of Environment*, 253, 112165.
- Simard, M., Pinto, N., Fisher, J. B., & Baccini, A. (2011). Mapping forest canopy height globally with spaceborne lidar. *Journal of Geophysical Research: Biogeosciences*, 116(G4).
- Simard, M., Fatoyinbo, L., Smetanka, C., Rivera-Monroy, V. H., Castañeda-Moya, E., Thomas, N., & Van der Stocken, T. (2019). Mangrove canopy height globally related to precipitation, temperature and cyclone frequency. *Nature Geoscience*, 12(1), 40-45.
- Sun, T., Qi, J., & Huang, H. (2020). Discovering forest height changes based on spaceborne lidar data of ICESat-1 in 2005 and ICESat-2 in 2019: A case study in the Beijing-Tianjin-Hebei region of China. *Forest Ecosystems*, 7, 1-12.
- 37. Mafi-Gholami, D., Zenner, E. K. & Jaafari, A. Mangrove regional feedback to sea level rise and drought intensity at the end of the 21st century. *Ecological Indicators* **110**, 105972 (2020).
- 38. Krauss, K. W. & Osland, M. J. Tropical cyclones and the organization of mangrove forests: a review. *Annals of Botany* **125**, 213–234 (2020).
- 39. Win, S., Towprayoon, S. & Chidthisong, A. Mangrove status, its ecosystem, and climate change in Myanmar: A study in Ayeyarwaddy Delta Coastal Zone. *IOP Conf. Ser.: Earth Environ. Sci.* **496**, 012007 (2020).
- 40. Aung, T. T., Than, M. M., Katsuhiko, O. & Yukira, M. Assessing the status of three mangrove species restored by the local community in the cyclone-affected area of the Ayeyarwady Delta, Myanmar. *Wetlands Ecol Manage* **19**, 195–208 (2011).
- 45. Lagomasino, D. *et al.* Storm surge and ponding explain mangrove dieback in southwest Florida following Hurricane Irma. *Nat Commun* **12**, 4003 (2021).
- 46. Radabaugh, K. R. *et al.* Mangrove Damage, Delayed Mortality, and Early Recovery Following Hurricane Irma at Two Landfall Sites in Southwest Florida, USA. *Estuaries and Coasts* **43**, 1104–1118 (2020).

Using just two data points, which probably have quite a considerable uncertainty about them for the analysis presented within the 'Supplementary Analyses' document, feels like a step too far. For example, Fig 4. uses these two points to back-project when this trend started, and there are a lot of assumptions being made, which I do not feel have been sufficiently backed up. I suggest that this additional height change analysis be removed and suggested as further work. It is not the focus of this article, and I feel the authors are overreaching with that further

analysis, which has been carried out as part of this revision.

Response [3]:

Thank you for pinpointing the confusion about height trend analysis. Please allow us to provide some explanation.

The remark of “just two data points” needs qualification as the two points are for the two years. The quadratic modeling of α was completed through statistical analysis of numerous data points over two years. After some experimentation, we selected mangroves shorter than 10 meters and divided them into 9 groups (from 1 m to 9 m) for modeling, as shown in Supplementary Fig. 2 below. Therefore, **the height trend analysis is based on global mangrove data of varying heights over two years**, although Supplementary Fig. 4 only displays two data points. In other words, all along the back-projected trajectory, the decay rate α for each height is supported by numerous data from the two years.

Supplementary Fig. 2| Variation of height-decay factor (α) with height using boxplots (in blue) and a non-linear quadratic model (in red). Horizontal dashed-line is at the maximum decay of the quadratic model fit.

The "assumptions" in the back-projected trend is that the decay/destruction rate is dependent upon height, **as statistically derived in the height-decay model**, nothing more or less. The back-projection is driven by destruction rates derived statistically from the data, as shown above. This model provides the entire basis of the back-projection with no further assumptions apart from setting the height in 2019. While there are statistical uncertainties, no further assumptions have been made.

However, we understand your concerns. In this revision, we have removed the height modeling from the main text (including the abstract and the “Height Trend” section) as it is not the focus of this research. We have retained it only in the Supplementary Materials, awaiting further in-depth research in the future as you suggest. The relevant sections have been modified accordingly:

Height Trends (lines 181-196):

For height analyses, 2007 forest canopy height data were missing, hence we used available **coarser-resolution** data for 2005. Globally, mean (median) ACH **across the 1°-grid**

cells declined from 8.8 m (8.18 m) in 2005 to 6.2 m (4.48 m) in 2019, representing a (mean) decline of 2.1%/year or 29.4% (~41% biomass decline) across the 14 years. The median decline rate was higher at 3.2%/year, suggesting that distributional properties were also changing inconsistently *by becoming more positively skewed, with more patches of lower height (see Fig. 1c)*. This represents almost a 20-fold (28-fold biomass) greater decline compared to the global width change noted in **Error! Reference source not found.**

The calculated ACH changes can be attributed to actual height reduction and possible changes in tree density due to thinning effects (more patches of lower height noted above). It should be noted that these changes could be influenced by possible data-related issues from transect-patchiness to some extent. Whilst resolution of confounding interactions in some areas may await more accurate future data, the 2019 height notes that many deep-water coastal areas are already of low height (< 10 m) or biomass (Fig. 2b). We also discuss later the regional spatial height variations using clusters which attempt to unravel regional and local inconsistencies in height changes suggested by differences in the mean and median rates, as detailed in the 'Cluster Exchange Results' section.

Additionally, I have one minor suggestion for the main text:

Line 399: "Validation of MCPI" > "Evaluation of MCPI". I propose using 'evaluation' instead of 'validation' as it better reflects the nature of your work. 'Validation' implies the use of suitable reference dataset(s), while in this case, you are comparing trends in related datasets and then inferring whether your results align with those trends.

Response [4]:

Thank you so much for your constructive suggestion. We made the corresponding revisions according to your suggestion (see in line 425):

“Evaluation of MCPI”

Reviewer #1 (Remarks on code availability):

I could not see the code, zendo said "The record is publicly accessible, but files are restricted to users with access."

Response [5]:

We apologize for our oversight and have checked our code-sharing link. Since the manuscript has not yet been officially published, we set the access mode to "Restricted." However, it can be accessed through a specially generated dedicated link that is visible to reviewers. We have placed the dedicated link in the "Code availability" section of the manuscript, but the NC system's "Code availability" module seems to have a character length limit, which prevented us from providing the full dedicated link.

You can access it via the dedicated link:

[https://zenodo.org/records/11178528?token=eyJhbGciOiJIUzUxMiJ9.eyJpZCI6IjJkOGQ0YmYzLTBhZWYtNDA3Zi04YTNhLTJmZmUyMjI0ODg2NiIsImRhdGEiOnt9LCJyYW5kb20iOiJkNTE1NGZlMDVjNjE4MzFhZmZmJiMGMGMGYwMzk0ZWYwZTM3NSJ9.Oq3EKykUqjBKz_](https://zenodo.org/records/11178528?token=eyJhbGciOiJIUzUxMiJ9.eyJpZCI6IjJkOGQ0YmYzLTBhZWYtNDA3Zi04YTNhLTJmZmUyMjI0ODg2NiIsImRhdGEiOnt9LCJyYW5kb20iOiJkNTE1NGZlMDVjNjE4MzFhZmZmJiMGMGYwMzk0ZWYwZTM3NSJ9.Oq3EKykUqjBKz_)

[3Vf4GTyDFNpDXxFJHc74Cp08ccRfoDRB0OQNVzssTl0sNzqDZmXO9MjUy9hbPEfVZw5KR1qQ](http://t.cn/A68xaz6C)

or via a shortened URL:

<http://t.cn/A68xaz6C>

Responses to Reviewer #2:

The authors have completed extensive revision on the analyses and its conclusions, and this revision is much changed from the first draft. The authors have meticulously responded to comments from reviewers.

Response [1]:

Thank you very much for your positive feedback and for recognizing the improvements in our manuscript. We greatly appreciate your thorough and meticulous review, as well as your constructive comments, which have significantly enhanced the quality of our work.

In this revision, we discussed additional possibilities for height reduction, including the influence of cyclones, ocean currents, and sea level rise, and made detailed modifications to some statements according to the reviewers' suggestions. Additionally, we provided spot checks from typical regions to support suggestions of destruction and thinning for reductions of "average height". Please refer to the point-by-point responses below for the specific revisions

we made to the manuscript.

Major comments on the revision:

Do you have evidence to support the theory that thinning and declining density is driving the decline in average height? As stated in the manuscript, this seems to be a theory rather than something that was quantified in any way.

You suggest that it could be recovery following deforestation (line 258), but couldn't decreased height also be due to recovery following hurricane damage, mortality events due to altered hydrology or burial events, etc.?

Declining density is stated as a fact in this paper, including in the abstract, but I don't see any data of density measurements. If there are data supporting thinning density (or other supporting documents in the literature), please state this more clearly. If not, be more clear that this is a theory in the paper and the abstract.

Response [2]:

Thank you so much for your careful reading and your constructive comments.

Unfortunately, we do not have field validation data on tree density, so thinning is indeed a theory. What we have done is a spot check on typical areas using high-resolution imagery from Google Earth Pro. The theory that thinning and declining density is driving the decline in mangrove average height is a reasonable explanation for our results.

Here are two typical regions we examined (amongst a number): the southwest of Myanmar and the southwest corner of Florida, USA. From the true-color satellite images, it is evident that the green mangrove areas have decreased over time, being replaced by bare soil or flooding. Additionally, the mangroves have become more fragmented. This indicates that although the Global Mangrove Watch (GMW) data in Fig (a) suggest these areas still have mangroves, the ecological condition may be poor or the mangroves have already been damaged due to storm surges, poor drainage and other factors (Win et al., 2020; Aung et al., 2011; Lagomasino et al., 2021; Radabaugh et al., 2020). The damaged mangroves have reduced in height, which lowers the tree density along the sampling lines, leading to a decrease in the extracted mangrove height along the transects.

Supplementary Fig. 16 | Spot check of mangrove thinning for southwest **Myanmar**. (a) shows the distribution of mangroves in 2019 from GMW; (b), (c), and (d) are satellite images from December 1984, December 2005, and December 2019, respectively, obtained from *Google Earth Pro*. Over time, the mangrove areas have become increasingly fragmented, possibly due to sea level rise and flooding. Typical areas of change are highlighted by yellow ovals.

Supplementary Fig. 17 | Spot check for mangrove destruction for the southwestern corner of **Florida, USA**. (a) shows the distribution of mangroves in 2019; (b), (c), and (d) are satellite images from December 2004, December 2017, and November 2019, respectively, obtained from *Google Earth Pro*. The white ovals highlight areas where mangrove loss occurred following Hurricane Irma's strike in September 2017 and had not fully recovered by 2019. The yellow oval areas are an enlargement of the rightmost white oval area to facilitate viewing.

Based on your suggestion, we recognize that our previous statement was lacking in consideration. Thinning, or the reduction in tree density, is not the only reason for the decrease in the average height along the transects; recovery from mortality events is also a significant factor. In the revised version, we have added a discussion of additional potential causes for the decline in ACH and revised the sentences involving thinning to indicate that it is just one of the possible explanations.

The corresponding revisions are as follows:

lines 85-87:

“Height-ACH reflected **“averaged height”** along the transect, hence **both actual height reduction and possible changes in tree density contributed to calculated changes in ACH.**”

lines 267-272:

“The reductions in ACH can be attributed to two potential mechanisms: a genuine reduction in height, which may result from recovery following mortality events (caused by factors such as human influence, cyclones, drought conditions and hydrological limitations)^{37,38,39,40}; and a reduction in tree density measured differently in 2005 and 2019 because ACH is an average measure based on data resolutions and transects perpendicular to the coastline (Supplementary Fig. 16 and 17).”

lines 329-332:

“First and foremost, we need to understand more deeply the processes at play that are dramatically **stressing** mangroves along **deep-water** boundary-current coasts and **possibly** changing mangrove height distributions (Fig. 1, Supplementary Fig. 9).”

Reference:

- 37. Mafi-Gholami, D., Zenner, E. K. & Jaafari, A. Mangrove regional feedback to sea level rise and drought intensity at the end of the 21st century. *Ecological Indicators* **110**, 105972 (2020).
- 38. Krauss, K. W. & Osland, M. J. Tropical cyclones and the organization of mangrove forests: a review. *Annals of Botany* **125**, 213–234 (2020).
- 39. Win, S., Towprayoon, S. & Chidthisong, A. Mangrove status, its ecosystem, and climate change in Myanmar: A study in Ayeyarwaddy Delta Coastal Zone. *IOP Conf. Ser.: Earth Environ. Sci.* **496**, 012007 (2020).
- 40. Aung, T. T., Than, M. M., Katsuhiko, O. & Yukira, M. Assessing the status of three mangrove species restored by the local community in the cyclone-affected area of the Ayeyarwady Delta, Myanmar. *Wetlands Ecol Manage* **19**, 195–208 (2011).
- 45. Lagomasino, D. *et al.* Storm surge and ponding explain mangrove dieback in southwest Florida following Hurricane Irma. *Nat Commun* **12**, 4003 (2021).
- 46. Radabaugh, K. R. *et al.* Mangrove Damage, Delayed Mortality, and Early Recovery Following Hurricane Irma at Two Landfall Sites in Southwest Florida, USA. *Estuaries and Coasts* **43**, 1104–1118 (2020).

Similarly, did you complete any quantitative analyses about the theory that current strengthening is driving the decline in mangroves? If so, please highlight that data and the quantitative analyses more clearly (and add more literature sources that back up the theory). If no quantitative analyses were performed, please make it more clear that this is a theory (particularly as it seems to be stated as fact, including in the abstract).

There is a pretty clear latitudinal trend in the height data in Supp Fig. 9. A lot of time is spent discussing how the EAC strengthening may be driving this trend, but little time is spent on cyclone activity and its possible influence, and there are many papers published on latitudinal trends in mangrove height and biomass in relation to cyclone activity. The influence of cyclones deserves more attention here.

Response [3]:

(i) Thank you so much for your careful reading and your constructive comments. We apologize for our original statement not being clear. The idea that current strengthening is driving the decline in mangroves is a theory, representing one possible cause of mangrove decline. In the revised version, we have added literature on the impact of ocean currents, as shown below (see in lines 286-290):

“Boundary-current regions, such as the East Australia Current, are hotspots of global warming, sea-level rise, storms, and marine heatwaves⁴¹. If mangroves in global hotspot regions are low in protection capacity and also declining, then they will be unable to sustain increasing climate change forces such as more frequent and intense storms, rising sea-levels and widespread sea surface density variations forcing currents^{42,43}.”

Additionally, we have revised the wording in the abstract and included relevant clarifications to indicate that we are only discussing potential possibilities.

lines 12-14:

“In contrast, our results highlight alarming, widespread declining low MCPI particularly along coasts exposed to deep-water, possibly in concert with human destruction, cyclones, and intensifying oceanic boundary-currents.”

lines 312-315:

“Our modelling does not distinguish between these destruction modes and changes may in some cases be confounded by data-related issues. All discussions here are potential possibilities, intended to provide references, modelling, analytical techniques, caveats for mangrove protection policies, and suggestions on further research of data-related issues.”

Reference:

- 41. Kelly, P., Clementson, L. & Lyne, V. Decadal and seasonal changes in temperature, salinity, nitrate, and chlorophyll in inshore and offshore waters along southeast Australia. *Journal of Geophysical Research: Oceans* **120**, 4226–4244 (2015).
- 42. Van der Stocken, T., Vanschoenwinkel, B., Carroll, D., Cavanaugh, K. C. & Koedam, N. Mangrove dispersal disrupted by projected changes in global seawater density. *Nature Climate Change* **12**, 685–691 (2022).

- 43. Hilmi, E. *et al.* Mangrove Landscaping As An Adaptation Pattern To Reduce The Impact of Climate Change in Segara Anakan Lagoon, Cilacap Regency Indonesia. *Baghdad Science Journal* **21**, 0338–0338 (2024).

(ii) Thank you for your comments regarding cyclones. In the revised version, we have included additional discussion on cyclones. We calculated the trends and recurrence of global tropical cyclones events. For cyclone tracks exceeding 34 knots (including tropical storms and hurricanes, as defined by the National Hurricane Center) from 2005 to 2019, 160 km buffers were generated to estimate their impact range (Amaral et al., 2023). Based on this, we obtained the number of times each mangrove grid was affected by cyclones per year. By overlaying the layers for 14 years, we derived the tropical storm recurrence (unit: times). Through linear regression of the annual storm impact frequencies, we obtained the trend of tropical storm changes (unit: times/year).

Supplementary Fig. 18 | The distribution of tropical cyclones trends (a) and recurrence (b). The tropical cyclones trends were derived through linear regression of the annual storm impact frequency, measured in times per year; the recurrence represents the total storm frequency from 2005 to 2019, measured in times. The dashed lines represent areas not affected by tropical storm events.

Supplementary Fig. 9 | Difference of height factor between 2007 and 2019. Note large height reductions in numbered boundary current regions of: (1) East Australian Current; (2) Agulhas Current; (3) Equatorial/Benguela Current; (4) Start of Gulf Stream Current; (5) California Current; (6) Kuroshio Current; and (7) Brazil Current. “Warm” currents (low to high latitude) are in dark-red, and “cold” ones (high to low latitude) in dark-blue.

We have added the corresponding content in the manuscript (see in lines 299-303):

“Cyclones may also contribute to the decline of mangroves. The comparison of tropical cyclone distribution (Supplementary Fig. 18) with mangrove height changes (Supplementary Fig. 9) from 2005 to 2019 shows some overlapping hotspots. These areas, such as the Philippines and southern China, frequently experience storm events that coincide with declines in mangrove height. Severe storm events can result in extensive mangrove mortality, with recovery taking several years or more^{38,44}.”

Reference:

- Amaral, C., Poulter, B., Lagomasino, D., Fatoyinbo, T., Taillie, P., Lizcano, G., ... & Roman-Cuesta, R. M. (2023). Drivers of mangrove vulnerability and resilience to tropical cyclones in the North Atlantic Basin. *Science of The Total Environment*, 898, 165413.
- 38. Krauss, K. W. & Osland, M. J. Tropical cyclones and the organization of mangrove forests: a review. *Annals of Botany* **125**, 213–234 (2020).
- 44. Peereman, J., Hogan, J. A. & Lin, T. Disturbance frequency, intensity and forest structure modulate cyclone-induced changes in mangrove forest canopy cover. *Global Ecol. Biogeogr.* **31**, 37–50 (2022).

Minor comments:

Line 11: “rapid global 800% growth in Cluster 8 of low height, width, and MCPI” The reader does not know about clusters yet, so avoid cluster language in favor of “mangrove forests with characteristically low height, width, and MCPI increased by 800%”

Response [4]:

Thank you so much for your careful reading and your constructive suggestion.

We made the corresponding revisions according to your suggestion (see in lines 9-10):

“Cluster-Exchange-Network-Analysis from 2007 to 2019 highlighted an 800% increase in mangrove forests with characteristically low height, width, and MCPI.”

Fig. 1. Nice addition of the violin plots! This helps to show how the variables are changing over time.

Response [5]:

Thank you for your recognition of the violin plot! We are very pleased to hear that this figure is helpful. We greatly appreciate your valuable suggestions from the previous round.

Fig. 3 I suggest making the y axis on all graphs the same to enable comparison of trends across

areas. Currently it is hard to make any comparisons among graphs

Response [6]:

Thank you for your thoughtful comment and recommendation. Your consideration is very reasonable, but the width range of mangroves varies widely across different continents. Unifying the y-axis range for all the graphs would compress the vertical distribution of data points in each subplot, making it difficult to compare trends (see in Response Fig. 4):

Response Fig. 4 | Results of trend regression analysis for Average cross-shore Mangrove Width (AMW) based on data from 1996 to 2020.

We have improved the graphs based on your suggestion by not unifying the y-axis range but maintaining a consistent scaling ratio, so that each y-axis unit distance represents 20 m and all y-axis spans are 120 m. This way, the unit length of the y-axis is the same for all graphs, allowing for an intuitive comparison of the change trends. Steeper lines indicate faster rates of change. Below are the updated graphs (see in line 176-180):

Fig. 1 Results of trend regression analysis for Average cross-shore Mangrove Width (AMW) based on data from 1996 to 2020. (a) and (b) represent the spatial patterns of annual mangrove width change rates extracted using $1^\circ \times 1^\circ$ grids and national boundaries, respectively. The unit is meters per year. (c) depicts global average trend of mangrove width. (d), (e), (f), and (g) represent width change trends in the Americas, Africa, Asia, and Oceania, respectively.

Line 204-205: “severest decreases were located at the southwest corner of Florida (-534.3)”
 You end year for comparison is 2019. Hurricane Irma hit the southwest coast of FL in 2017 and caused extensive mangrove damage. While a lot of the mangroves have recovered at this point, this is probably driving the decline at the 2019 timepoint (only 2 years post-hurricane).

Response [7]:

Thank you so much for your constructive suggestion and guidance.

Upon verification, we found that the areas where mangrove height significantly decreased in 2019 correspond to the areas identified by Lagomasino et al. (2021) as having low resilience after Hurricane Irma. This suggests that the decline in MCPI in these areas is indeed likely influenced by the impact of Irma in 2017, with no recovery even after two years. We compared satellite images from different times on Google Earth, as shown in Supplementary Fig. 17.

We made the corresponding revisions and added relevant discussions according to your suggestion (see in line 303-308):

" For instance, in southwestern Florida, MCPI decreased significantly from reductions in width,

height, and NDVI, likely due to the impact of Hurricane Irma⁴⁵ (Supplementary Fig. 17). Low-lying topographic conditions, hydrologic isolation caused by natural or artificial coastal barriers, and storm surges all contributed to poor drainage of mangroves, resulting in extensive diebacks post-storm⁴⁶. These mangroves demonstrated low resilience and had not recovered by 2019, as indicated by the detected decline in ACH (very low or zero)."

Supplementary Fig. 17 | Spot check for mangrove destruction for the southwestern corner of **Florida, USA**. (a) shows the distribution of mangroves in 2019; (b), (c), and (d) are satellite images from December 2004, December 2017, and November 2019, respectively, obtained from *Google Earth Pro*. The white ovals highlight areas where mangrove loss occurred following Hurricane Irma's strike in September 2017 and had not fully recovered by 2019. The yellow oval areas are an enlargement of the rightmost white oval area to facilitate viewing.

Reference:

- 45. Lagomasino, D. *et al.* Storm surge and ponding explain mangrove dieback in southwest Florida following Hurricane Irma. *Nat Commun* **12**, 4003 (2021).
- 46. Radabaugh, K. R. *et al.* Mangrove Damage, Delayed Mortality, and Early Recovery Following Hurricane Irma at Two Landfall Sites in Southwest Florida, USA. *Estuaries and Coasts* **43**, 1104–1118 (2020).

Table 3 caption “Exchange of clusters is reported later.” Rather than saying “later,” refer the reader to the specific section of the text that explains clusters

Response [8]:

Thank you so much for your careful reading and advice. We made the corresponding revisions according to your suggestion. We changed the original “Exchange of clusters is reported later.”

into “Please refer to the 'Cluster Exchange Results' section below for details on ‘Cluster change’.” (see in line 227-228)

Line 302: Perhaps recovery from mortality events (due to cyclones, drought conditions, restricted hydrology, etc.) is a reason for thinning

Response [9]:

Thank you so much for your constructive comments. We have added the corresponding content according to your suggestions.

The corresponding revisions are as follows (see in lines 267-272):

“The reductions in ACH can be attributed to two potential mechanisms: a genuine reduction in height, which may result from recovery following mortality events (caused by factors such as human influence, cyclones, drought conditions and hydrological limitations)^{37,38,39,40}; and a reduction in tree density measured differently in 2005 and 2019 because ACH is an average measure based on data resolutions and transects perpendicular to the coastline (Supplementary Fig. 16 and 17).”

Reference:

- 37. Mafi-Gholami, D., Zenner, E. K. & Jaafari, A. Mangrove regional feedback to sea level rise and drought intensity at the end of the 21st century. *Ecological Indicators* **110**, 105972 (2020).
- 38. Krauss, K. W. & Osland, M. J. Tropical cyclones and the organization of mangrove forests: a review. *Annals of Botany* **125**, 213–234 (2020).
- 39. Win, S., Towprayoon, S. & Chidthisong, A. Mangrove status, its ecosystem, and climate change in Myanmar: A study in Ayeyarwaddy Delta Coastal Zone. *IOP Conf. Ser.: Earth Environ. Sci.* **496**, 012007 (2020).
- 40. Aung, T. T., Than, M. M., Katsuhiko, O. & Yukira, M. Assessing the status of three mangrove species restored by the local community in the cyclone-affected area of the Ayeyarwady Delta, Myanmar. *Wetlands Ecol Manage* **19**, 195–208 (2011).

Responses to Reviewer #3:

This is the second time I have looked at this manuscript and I can see that the authors have addressed the concerns I raised in my first review. The revised manuscript, with improved clarity and incorporation of biomass has improved the predictive capacity of MCPI. I particularly appreciated the additional analyses incorporated into the manuscript. I do not have further comments on the manuscript and commend the authors for the effort to address reviewer comments.

Response [1]:

Thank you very much for your positive feedback and for recognizing the improvements in our manuscript. We greatly appreciate your thorough and meticulous review, as well as your constructive comments, which have significantly enhanced the quality of our work.

Reviewer #3 (Remarks on code availability):

I have tried to access the code below, but access is restricted.

Response [2]:

We apologize for our oversight and have checked our code-sharing link. Since the manuscript has not yet been officially published, we set the access mode to "Restricted." However, it can be accessed through a specially generated dedicated link that is visible to reviewers. We have placed the dedicated link in the "Code availability" section of the manuscript, but the NC system's "Code availability" module seems to have a character length limit, which prevented us from providing the full dedicated link.

You can access it via the dedicated link:

https://zenodo.org/records/11178528?token=eyJhbGciOiJIUzUxMiJ9.eyJpZCI6IjJkOGQ0YmYzLTBhZWYtNDA3Zi04YTNhLTJmZmUyMjI0ODg2NiIsImRhdGEiOiJkNTE1NGZlMDVjNjE4MzFhZmJiMGYwMzk0ZWYwZTM3NSJ9.Oq3EKykUQjBKz_3Vf4GTyDFNpDXxPJHc74Cp08ccRfoDRB0QNVzTsTl0sNzqDZmXO9MjUy9hbPEfVZw5KR1qQ

or via a shortened URL:

<http://t.cn/A68xaz6C>

Responses to Reviewers

We are grateful to all Reviewers for the helpful and constructive feedback. We have addressed every comment carefully and revised the manuscript accordingly. Below please find our point-by-point responses to each review comment. *The review comments are highlighted in blue*, while our responses are in **black**.

Responses to Reviewer #1:

The authors have provided an extensive explanation behind the points raised and appropriately revised the manuscript. I am, therefore, content that the article goes forward for publication.

Response [1]:

Thank you very much for your positive feedback and for recognizing the improvements in our manuscript. We greatly appreciate your thorough and meticulous review, as well as your constructive comments, which have significantly enhanced the quality of our work.

The code is available, and it is relatively straightforward to follow, but there is no README explaining how to use the scripts provided and the code could have more comments explaining what the blocks are doing so the user could more easily follow and use the code.

Response [2]:

Thank you very much for your suggestions and guidance. We apologize for the oversight and have now added a README file with a brief introduction on how to use the scripts. Additionally, we've included more comments to explain the code blocks.

Responses to Reviewer #2:

The authors have once again provided meticulous responses to the reviewer requests. I appreciate the addition of the cyclone impacts on mangroves and additional analyses and feel the theories are more accurately portrayed in this revision. My requested edits have been sufficiently addressed.

Response [1]:

Thank you very much for your positive feedback and for recognizing the improvements in our manuscript. We greatly appreciate your thorough and meticulous review, as well as your constructive comments, which have significantly enhanced the quality of our work.